# Nuclei-specific hypothalamus networks predict a dimensional marker of stress in humans

Daria E. A. Jensen [1,2,3,4] ✉, Klaus P. Ebmeier [3], Sana Suri[3,5], Matthew F. S. Rushworth [1,2] & Miriam C. Klein-Flügge [1,2,3] ✉

The hypothalamus is part of the hypothalamic-pituitary-adrenal axis which activates stress responses through release of cortisol. It is a small but heterogeneous structure comprising multiple nuclei. In vivo human neuroimaging has rarely succeeded in recording signals from individual hypothalamus nuclei. Here we use human resting-state fMRI (n = 498) with high spatial resolution to examine relationships between the functional connectivity of specific hypothalamic nuclei and a dimensional marker of prolonged stress. First, we demonstrate that we can parcellate the human hypothalamus into seven nuclei in vivo. Using the functional connectivity between these nuclei and other subcortical structures including the amygdala, we significantly predict stress scores out-of-sample. Predictions use 0.0015% of all possible brain edges, are specific to stress, and improve when using nucleus-specific compared to whole-hypothalamus connectivity. Thus, stress relates to connectivity changes in precise and functionally meaningful subcortical networks, which may be exploited in future studies using interventions in stress disorders.

Mental disorders present a huge global burden. Currently, treatments are typically chosen based on broad symptom-based diagnoses, rather than a mechanistic understanding of an individual's behavioural and brain changes[1,2]. A growing body of work has therefore focused on analysing the brain changes that accompany, predict, or result from mental illness. Recently, in addition, this type of work has increasingly shifted towards a dimensional or transdiagnostic perspective[3,4]. In line with this, we have shown that a dimensional approach that focuses on interpretable functions such as sleep problems or negative emotions, as opposed to broad classifications or classes of symptoms, aids the characterisation of the underlying brain networks[5].

Prolonged periods of stress are associated with several common psychiatric disorders including depression and anxiety[6,7]. Yet, the brain changes that accompany stress are insufficiently understood. Stress can be defined as an actual or anticipated disruption of homeostasis or an anticipated threat to well-being[8]. Physiological and behavioural responses to acute stress are adaptive, spatially and temporally specific, and regulated by a distributed network of regions including the prefrontal cortex, amygdala, and hippocampus[9–12]. When these circuits trigger autonomic and endocrine stress responses, brainstem centres as well as the hypothalamic-pituitary-adrenal (HPA) are activated[8,9,13]. Despite the various contributions made by this diverse set of regions, in this study, we focus on the hypothalamus. As

[1]Department of Experimental Psychology, University of Oxford, Tinsley Building, Mansfield Road, Oxford OX1 3TA, UK. [2]Wellcome Centre for Integrative Neuroimaging (WIN), Centre for Functional MRI of the Brain (FMRIB, University of Oxford, Nuffield Department of Clinical Neurosciences, Level 6, West Wing, John Radcliffe Hospital, Oxford OX3 9DU, UK. [3]Department of Psychiatry, University of Oxford, Warneford Hospital, Warneford Lane, Oxford OX3 7JX, UK. [4]Clinic of Cognitive Neurology, University Medical Center Leipzig and Max Planck Institute for Human Cognitive and Brain Sciences, Stephanstrasse 1a, 04103 Leipzig, Germany. [5]Wellcome Centre for Integrative Neuroimaging (WIN), Oxford Centre for Human Brain Activity (OHBA), University of Oxford, Warneford Hospital, Warneford Lane, Oxford OX3 7JX, UK. ✉e-mail: daria.jensen@psych.ox.ac.uk; miriam.klein-flugge@psy.ox.ac.uk

part of the HPA axis, the hypothalamus is one of the key brain regions mediating the balance of hormones in response to acute stress[14–16]. For instance, the hypothalamus can directly activate the anterior pituitary and, via the adrenal cortex, trigger the release of cortisol. However, when stress turns into a prolonged state, it becomes maladaptive[17–20] and is frequently associated with compromised well-being. Currently, it remains unclear how human hypothalamus networks are affected by stress, in particular prolonged periods of stress. This might partly be because it is challenging to study the hypothalamus in vivo in humans.

The human hypothalamus is a small pea-sized structure located deep inside the brain[21], which makes it prone to signal drop-out in functional magnetic resonance imaging (fMRI) and difficult to study at voxel sizes commonly used in fMRI (~ 30 voxels at 3 mm³). In addition, the hypothalamus is not homogenous but contains an intricate sub-structure of more than 10 nuclei[22]. Thus, fMRI studies conducted in humans in vivo have typically ignored its component nuclei or ignored the hypothalamus altogether. Research in humans related to stress has tended to focus on larger, often cortical, regions with good signal-to-noise[10,23,24]. This stands in stark contrast to animal work on stress which has focused on deep subcortical and brainstem circuits[9,14,25–28]. The absence of comparable work in humans currently impedes progress in psychiatry because prolonged periods of stress are likely to affect the same subcortical circuits in humans.

In general, while it is established that whole-brain connectivity can predict health outcomes, typically, such predictions rely on large brain networks[23,29–31]. This means that they can reach impressive prediction accuracies, but only provide limited guidance for a mechanistic understanding of the nature of a condition and for targeting interventions.

Here, we use a large sample of healthy young adults (n = 498) and resting-state (rs)-fMRI data acquired with high spatial resolution and signal-to-noise to examine whether hypothalamus functional connectivity relates to a dimensional marker of prolonged stress. We examine whether subdividing the hypothalamus into component nuclei improves the identification of stress correlates. We focus on connections in a small network between the hypothalamus and subcortical regions including the extended amygdala[32,33], nucleus accumbens, and nuclei of the major brainstem neurotransmitter systems. We find that stress scores can be predicted in an independent cohort based on the functional connectivity of the hypothalamus. Predictions are improved by considering individual hypothalamic nuclei, as opposed to the whole hypothalamus, and are functionally specific to stress. Our study provides evidence that functional connectivity in precise subcortical brain circuits relates to subclinical variability in stress.

## Results

### In vivo parcellation of the human hypothalamus
The hypothalamus contains more than ten nuclei and has been characterised *post-mortem* and, more recently, using careful processing of high-resolution structural images[22,34] (Supplementary Table 2). Tracer work in non-human primates[35–39] suggests that individual hypothalamus nuclei differ not only in their local structure but also in their coupling with other brain regions. To date, however, in vivo, human neuroimaging approaches focusing on differences in connectivity have only distinguished a small number of hypothalamus subdivisions[40]. Therefore, our first aim was to build on our recent experience with subcortical parcellation of the amygdala[5] to derive a parcellation of the human hypothalamus that reflects its detailed anatomical organisation but which can be performed in vivo.

We measured functional connectivity using the Human Connectome Project's (HCP's) high-quality high-resolution rs-fMRI data. As in our previous work, we performed additional pre-processing steps to remove physiological confound signals that particularly affect subcortical regions close to major vessels and pulsating fluid-filled spaces,

such as the hypothalamus (see Methods and refs. 5, 26, 41). In an initial step, we then characterised the connectivity of all hypothalamus voxels to all other brain ordinates (i.e., all vertices on the cortical surface and all voxels in the subcortex) by computing the average functional connectivity of n = 200 young-adult HCP participants scanned at 3 Tesla (3 T, 2 mm isotropic voxel resolution; Fig. 1A; see Methods and Supplementary Fig. 2B for participant selection). Hypothalamus functional connectivity reflected connections expected from tracer work, such as strong coupling with perigenual and subgenual anterior cingulate cortex (pgACC, sgACC), insula, anterior temporal cortex, ventral striatum, central amygdala and brainstem[35–39]. This suggests that we succeeded in obtaining meaningful estimates of hypothalamus resting-state coupling. The average hypothalamus functional connectivity identified in the first 200 participants was replicated in two additional HCP datasets (3 T: n = 200; 7 T: n = 98; Supplementary Fig. 1A, B).

Having established that our data provide reliable markers of resting-state coupling, we next sought to identify putative subnuclei within the hypothalamus. We performed hierarchical clustering using our previously established procedure[5]. We computed a similarity matrix that summarises, for all pairs of hypothalamus voxels, the similarity of their absolute functional connectivity pattern with all other brain ordinates. Thus, if two hypothalamus voxels share a similar connectivity profile to the rest of the brain, their similarity is large, and they are more likely to be assigned to the same cluster. Clustering of the similarity matrix was performed using the group average hypothalamus connectome and thus blind to any variability present across participants relevant for later analyses. Cluster solutions with increasing numbers of clusters from 2 to 14 were evaluated with reference to previous parcellations in terms of their anatomical plausibility and symmetry (Supplementary Fig. 2A; see Methods). Using these criteria, we chose a detailed and anatomically plausible parcellation that contained seven nuclei per hemisphere (Fig. 1B, C). It identified subdivisions between anterior and posterior as well as dorsal and ventral parts of the hypothalamus. However, it did not distinguish between medial and lateral hypothalamus nuclei, possibly because our 2 mm resolution was insufficient for separating clusters in the medial-lateral axis. Note that the clustering algorithm was not constrained to induce hemispheric symmetry, any aspect of cluster size, or spatial contiguity of the voxels assigned to a given cluster. Nevertheless, the clusters we obtained differed in size and were spatially contiguous and relatively symmetrical between hemispheres. Thus, these anatomically realistic features emerged naturally. Once again, at this point in the analysis, we checked to ensure that it was possible to replicate our results. We closely replicated the hypothalamus parcellation derived from the first 200 participants in two additional datasets (3 T: n = 200; 7 T: n = 98; Supplementary Fig. 3) but used the original parcellation throughout the manuscript.

To aid comparison with prior work and the reporting of our results, we assigned each cluster a putative nucleus label (Fig. 1D). This process was largely guided by comparing the location of our clusters with the nuclei described in Mai et al.[22], but also incorporated other previous parcellations[34,42–44]. Our own and Mai et al.'s parcellations are shown side by side in Supplementary Fig. 9. The most dorsal anterior nucleus closely resembled what prior animal and human work commonly refer to as paraventricular nucleus, sometimes denoted as Pa, PVH, or PA[22,34,42–44], which we denote as PV throughout. The cluster ventral to PV most resembled the medial preoptic hypothalamic nucleus (MPO) across multiple previous parcellations[22,34,42–44] and the most ventral anterior cluster likely contained the supraoptic and suprachiasmatic nuclei elsewhere referred to as SChC, SChD, SCh, or SO and referred to as SO/SC here[22,34]. Moving more posteriorly, the cluster posterior to PV was labelled dorsomedial nucleus (DM) and the cluster ventral to DM ventromedial nucleus (VM), both consistent with the same nuclei in

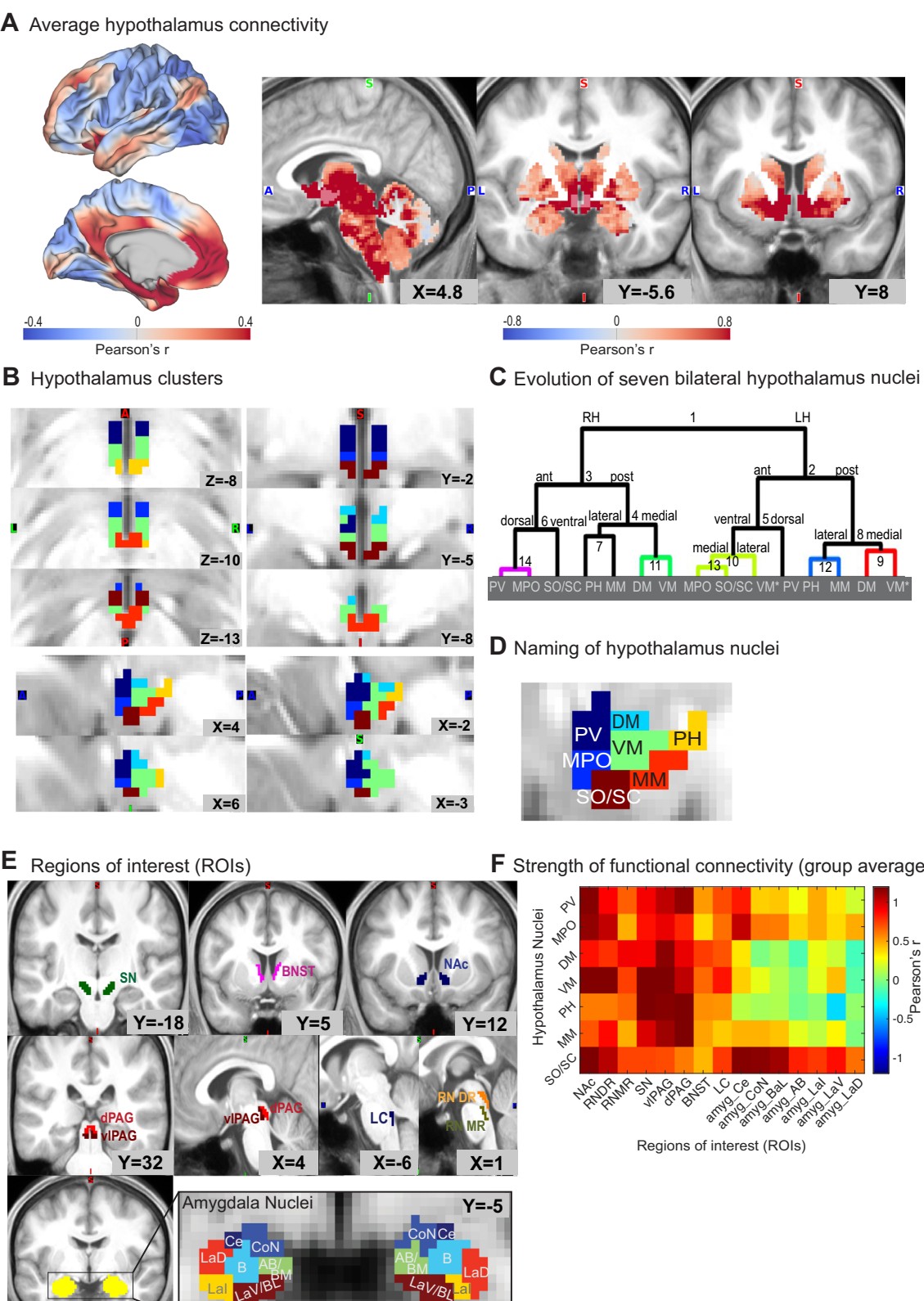

**A** Average hypothalamus connectivity

**B** Hypothalamus clusters

**C** Evolution of seven bilateral hypothalamus nuclei

**D** Naming of hypothalamus nuclei

**E** Regions of interest (ROIs)

**F** Strength of functional connectivity (group average)

previous parcellations[22,34,42–44]. The cluster posterior and ventral to VM was most consistent with the location of the mammillary body in Mai et al.[22] (MM). Finally, the most posterior superior hypothalamus division was labelled posterior hypothalamic nucleus (PH), again consistent with the same nucleus in other work[22,34,44]. See Supplementary Table 3 for a side-by-side comparison between our labels and those of previous hypothalamus parcellations.

## Nuclei-specific hypothalamus functional connectivity

Having established seven subdivisions within the hypothalamus, our next step was to characterise the functional connectivity of hypothalamus nuclei with regions of interest (ROIs) determined a priori (Fig. 1E, F). We focused on a small and exclusively subcortical network that is known to strongly connect with the hypothalamus, to regulate key neurotransmitter systems, and play a role in stress responses[45–48]. This

**Fig. 1 | Connectivity-based parcellation of the human hypothalamus. A** Group average hypothalamus-to-whole-brain functional connectivity extracted from resting-state fMRI data ($n = 200$ 3 T HCP participants; colour scale: Pearson's correlation coefficient; positive/negative functional connectivity: red/blue). The Hypothalamus outline is shown in semi-transparent colour; strong red values within the hypothalamus indicate strong autocorrelation of activity. **B** Parcellation of the human hypothalamus into seven nuclei ($n = 200$ 3 T HCP participants): top left−horizontal, top right−coronal, bottom−sagittal for right and left hemispheres. The parcellation used hierarchical clustering of the similarity between hypothalamus voxels in terms of their whole-brain functional connectivity. Clusters show high symmetry across hemispheres and good agreement with prior high-resolution and post-mortem work in humans. This parcellation was used throughout the study but was also replicated in two independent datasets ($n = 200$ 3 T and n = 98 7 T HCP participants; Supplementary Fig. 3). **C** Dendrogram of the hierarchical clustering shown in (b) shows the evolution of clusters up to depth 15. Intermediate parcellation steps are shown in Supplementary Fig. 2. **D** Putative naming of hypothalamus nuclei used throughout this study (illustrated for the right hemisphere at $X = 4$): PV paraventricular, MPO medial preoptic, DM dorsomedial, VM ventromedial, SO/SC supraoptic and suprachiasmatic, PH posterior, MM mammillary bodies. Number of voxels across both hemispheres: PV = 23, MPO = 10, DM = 9, VM = 28, PH = 10, MM = 15, SO/SC = 14 (hypothalamus total: 109). **E** Subcortical regions of interest (ROIs): substantia nigra (SN), bed nucleus of the stria terminalis (BNST), nucleus accumbens (NAc), dorsal and ventrolateral periaqueductal grey (dPAG/vlPAG), locus coeruleus (LC), dorsal and median raphe (RNDR, RNMR). For amygdala, the main analyses used individual nuclei (Figs. 3 and 4); control analyses used the whole amygdala (Fig. 5). **F** Group average functional connectivity between hypothalamus nuclei and a priori ROIs for $n = 200$ 3 T participants (replicated in two further datasets: $n = 200$ 3 T or $n = 98$ 7 T; Supplementary Fig. 1; colour bar: Pearson's $r$). Source data for 1b-e are provided as a Source Data file. A anterior, P posterior, S superior, I inferior, L left, R right, amygdala nuclei: Ce central, CoN cortical, B basal, AB/BM auxiliary basal/basomedial, LaI lateral intermediate, LaD lateral dorsal, LaV/BL lateral ventral/basolateral.

included the extended amygdala−seven nuclei per hemisphere within the amygdala[5] plus the bed nucleus of the stria terminalis, BNST[49,50])−as well as the locus coeruleus (LC), associated with noradrenaline, dorsal and median raphe (RNDR, RNMR) associated with serotonin, substantia nigra (SN) and nucleus accumbens (NAc), associated with dopamine, and the ventrolateral and dorsal periaqueductal grey (vlPAG, dPAG) associated with rest and digest and fight and flight responses (Fig. 1F; see Methods for details on ROI definition). See supplementary information for analyses that additionally included subdivisions of the hippocampus, given its clear role in stress[51,52].

Characterisation of the group average functional connectivity between these 15 ROIs and the seven hypothalamus nuclei (Fig. 1F) revealed that brainstem nuclei generally showed positive coupling with hypothalamus nuclei which was often strongest with more posterior hypothalamus (VM, MM, PH). By contrast, the functional connectivity of the nucleus accumbens and amygdala was most pronounced with anterior portions of the hypothalamus (PV, MPO and SO/SC), especially the most ventral anterior portion (SO/SC). Within the amygdala, the strongest coupling with the hypothalamus was with the central amygdala. Overall, these patterns closely resemble findings from tracer work in macaques[35–39,53]. Once again, at this point, we sought to replicate the results of our analyses in the additional datasets. We were able to replicate the functional connectivity patterns between hypothalamic nuclei and a priori ROIs found in the first 200 participants in two additional datasets (correlation between first connectivity matrix and: (1) $n = 200$ 3 T replication dataset: two-tailed Pearson's $r(103) = 0.893$, $p = 1.748 \times 10^{-37}$, CI = [0.85, 0.93]; (2) $n = 98$ 7 T replication dataset: two-tailed Pearson's $r(103) = 0.841$, $p = 3.064 \times 10^{-29}$, CI = [0.77, 0.89]; Supplementary Fig. 1C).

**Extracting a dimensional marker of stress**
Next, we turned to the behaviour of interest. Our key goal was to extract a dimensional score that robustly captures an individual's stress level. Rather than capturing the ability to respond to immediate stress, we aimed to capture participants' experience of stress over a prolonged time (i.e., questionnaire scores related to stress experiences over the last week, month or in general; see Methods). Because the young-adult HCP database does not include individuals with a diagnosis of a stress or anxiety disorder, our goal was to characterise stress on a continuum in the subclinical range. We, therefore, extracted several available questionnaire markers that captured relevant constructs, such as perceived stress, self-efficacy, and fear (Fig. 2A) and ran a factor analysis to combine them into one factor (confirmed by using a Scree test, see Methods; Fig. 2B, C). The factor analyses on the first and second cohorts of $n = 200$ 3 T participants were virtually identical (see Methods). Thus, for consistency, both cohorts were pooled, and identical factor weights were used for all individuals. For each participant, we weighted their individual questionnaire scores with the

factor loadings and summed them to obtain a single marker of stress. Figure 2D shows the distribution of the obtained stress scores for all $n = 400$ individuals in the 3 T cohort.

Again, we sought to replicate our results at this stage of the analysis. The factor analysis was replicated in a second, larger cohort. Because it focuses on behavioural rather than fMRI data, it can be derived based on all HCP participants and there is no need to restrict it to the subset of data with the highest quality physiological recordings (required for pre-processing the fMRI data). When the analysis was repeated on the full set of $n = 1206$ 3 T HCP participants minus the previously included $n = 400$ participants (total of $n = 806$), the resulting stress factor was highly similar (two-tailed Pearson's correlation between factor loadings $n = 400$ vs. $n = 806$ 3 T participants: $r(5) = 0.998$; $p = 1.162 \times 10^{-07}$; CI = [0.99, 1]; Supplementary Fig. 4B). Note, however, that stress scores obtained for the 7 T cohort, using the same factor weights, resulted in a noticeably narrower range, with less extreme stress levels compared to the 3 T cohorts (two-sample F test of equal variances between 3 T and 7 T cohorts: $F(397,96) = 1.876$; $p < 0.001$; CI = [1.35, 2.54]; 3 T: min: −7.10, max: 7.44; 7 T: min: −4.87, max: 4.12). This could be because individuals with heightened stress levels may prefer not to enter a high-field MR scanner (Supplementary Fig. 4C). This lack of variability in stress scores in the 7 T cohort may limit or even preclude an examination of how interindividual variation in hypothalamic connectivity relates to interindividual variability in stress in this cohort.

**Relating stress and hypothalamus nuclei connectivity**
In the next step, we examined whether hypothalamus nuclei connectivity carries information about individuals' stress levels. While so far, we considered the group average connectivity, in this step, we were interested in interindividual (i.e., between-subject) variability in hypothalamus connectivity and its potential relationship with interindividual variability in stress scores. We generated a new split of all $n = 400$ 3 T participants into a *train* and a *test* group with comparable means and distributions of stress scores (Supplementary Fig. 4C and overview in Supplementary Fig. 2B). This split was generated only once with the aim of ensuring a comparable range of stress scores across the two groups and solely based on behaviour (see Methods). It therefore did not bias any analyses examining relationships with functional connectivity.

To improve the reliability of the functional connectivity estimates obtained from individual participants, we rejected three participants with outlier connectivity values (see Methods and ref. 5). All subsequent analyses thus rely on a total of $n = 398$ 3 T participants (split into train and test groups: $n = 198$ and $n = 200$) and $n = 97$ 7 T participants.

Next, we established whether relationships between nuclei-specific hypothalamus functional connectivity and stress were

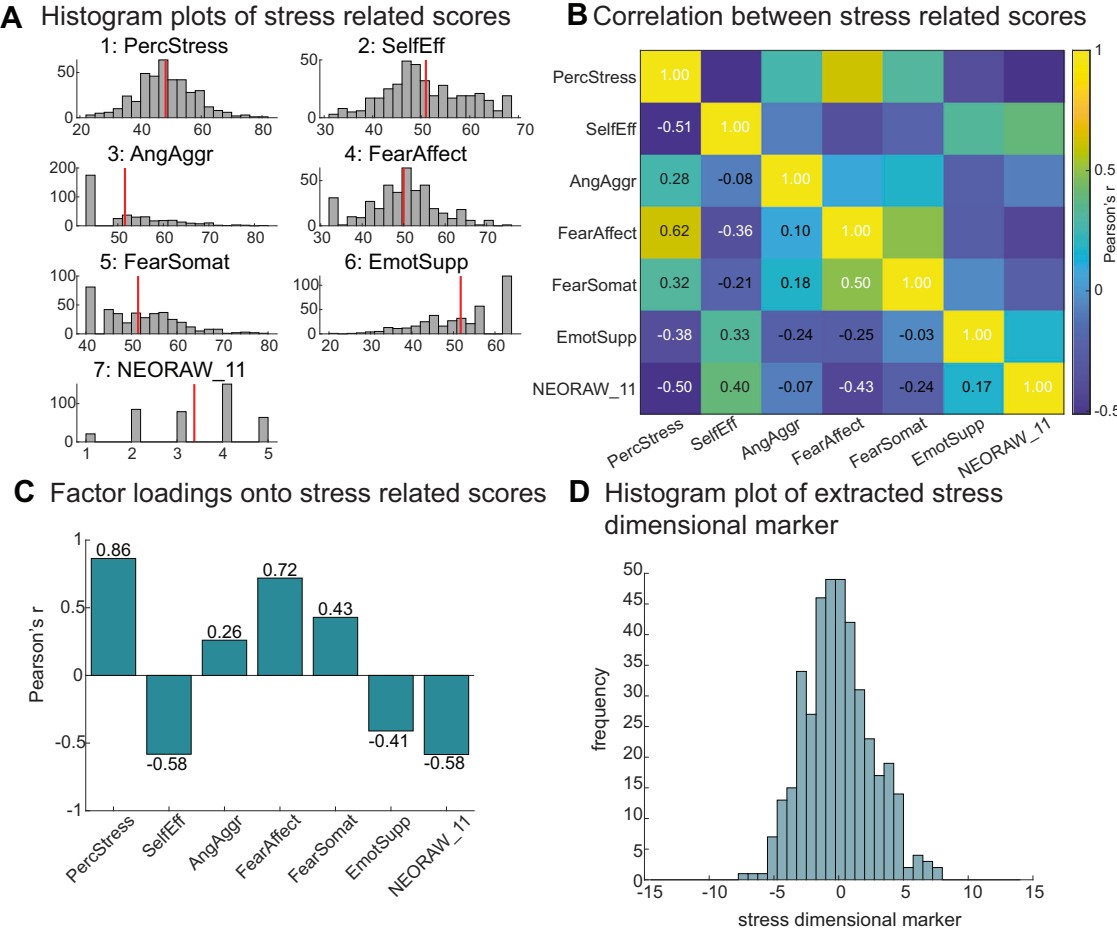

**Fig. 2 | Extracting a dimensional marker of prolonged stress. A** Histograms show distributions for seven stress-related scores in $n = 400$ 3 T HCP participants that were entered into a factor analysis as shown in panel **C**. **B** Correlations between the seven stress-related scores across all $n = 400$ participants (colour bar denotes Pearson's $r$). **C** Factor analysis was used to extract a one-dimensional marker of stress: shown are the loadings onto the seven stress-related scores. The highest loadings are with perceived stress (PercStress), fear (FearAffect), self-efficacy (SelfEff, negative loading) and the ability to cope with stress (NEORAW_11, negative loading: When I'm under a great deal of stress, sometimes I feel like I'm going to pieces). Intermediate loadings were with anger aggression (AngAggr, positive loading), somatic fear (FearSomat, positive loading), and emotional support (EmotSupp, negative loading). The factor analysis was replicated in an independent dataset (Supplementary Fig. 4). **D** Distribution of the derived dimensional marker of stress generated from the factor analysis for all $n = 400$ 3 T participants; the $n = 98$ 7 T participants had reduced stress variance (see Supplementary Fig. 4).

replicated between the two independent train and test datasets. We estimated robust regression coefficients for each of the $7 \times 15$ connections or edges between hypothalamus nuclei and ROIs, separately for the test and train groups, to quantify the relationship between functional connectivity and stress across participants (Fig. 3B). If hypothalamus connectivity carries no information about stress, then the correlation between regression coefficients obtained in the train and test datasets should be zero. We formally tested this using a non-parametric permutation null distribution ($n = 10,000$ stress score shuffles, functional connectivity unchanged). Indeed, by chance, the across-dataset replication of the pattern of regression coefficients was centred on zero (Fig. 3A). However, the similarity between train and test group regression coefficients in the actual data was significantly greater than chance (one-tailed Pearson's correlation testing for a positive relationship based on the non-parametric permutation null distribution: $r(396) = 0.305$; $p = 0.002$, CI = [0.12, 0.47]; Fig. 3A). This shows that relationships between hypothalamus nuclei connectivity and stress were similar across the 3T-train and 3T-test cohorts, two cohorts with comparable stress levels (Fig. 3A). This was not the case, however, when comparing the 3 T with the 7 T cohort which had considerably smaller variability in stress scores (Supplementary Figs. 4A, C and 6A, B).

For a first insight into the anatomical circuits that particularly contribute to the similarity across datasets, we derived a measure of contribution for each edge. We computed the difference in correlation coefficient between the patterns of regression coefficients in the train and test group when the edge was included versus excluded for computing the correlation (rDiff; Fig. 3B, third panel[5]), This highlighted, for example, the importance of functional connectivity of PV with NAc and central amygdala, of MPO with BNST, and of SO/SC with RNMR, auxiliary basal and cortical amygdala nuclei (CoN and AB; Fig. 3B, third panel).

To further demonstrate the consistency of hypothalamus connectivity relationships with stress, we repeated the above analyses, but this time comparing regression coefficients extracted from two halves of each participant's resting-state data (the first half of each session: run 1 and 3, versus the second half of each session: run 2 and 4). Again, robust regression coefficients were computed for each edge but based on the first and second half of resting-state data. The consistency in the pattern of robust regression coefficients was greater across experimental halves than expected by chance (one-tailed Pearson's correlation testing for a positive relationship based on the non-parametric permutation null distribution: $r(396) = .0368$; $p < 0.001$, CI = [0.19, 0.52]; Fig. 3A, B).

## Predicting stress using hypothalamus nuclei coupling

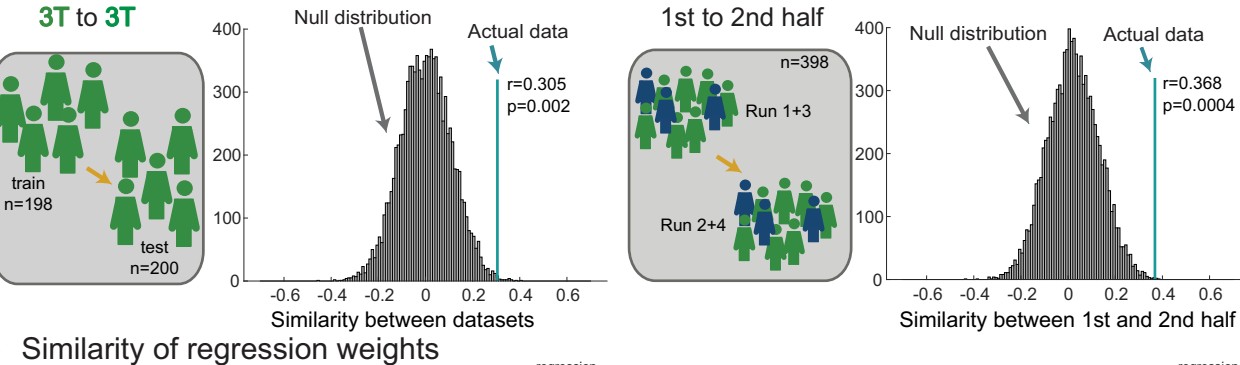

**A** (1) Across-subject (3T to 3T)          (2) Within-subject (3T, n=398)

## B Similarity of regression weights

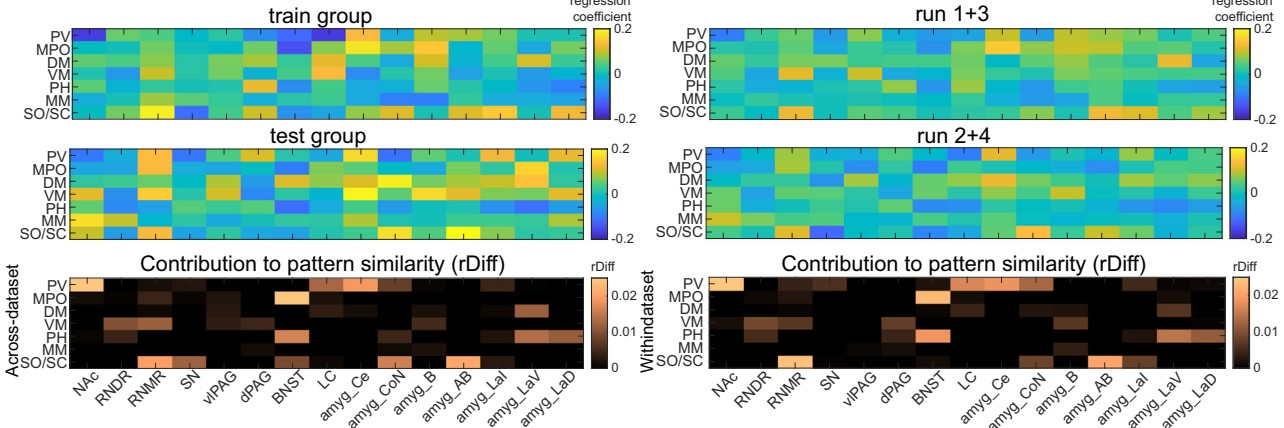

## C Out-of-sample prediction

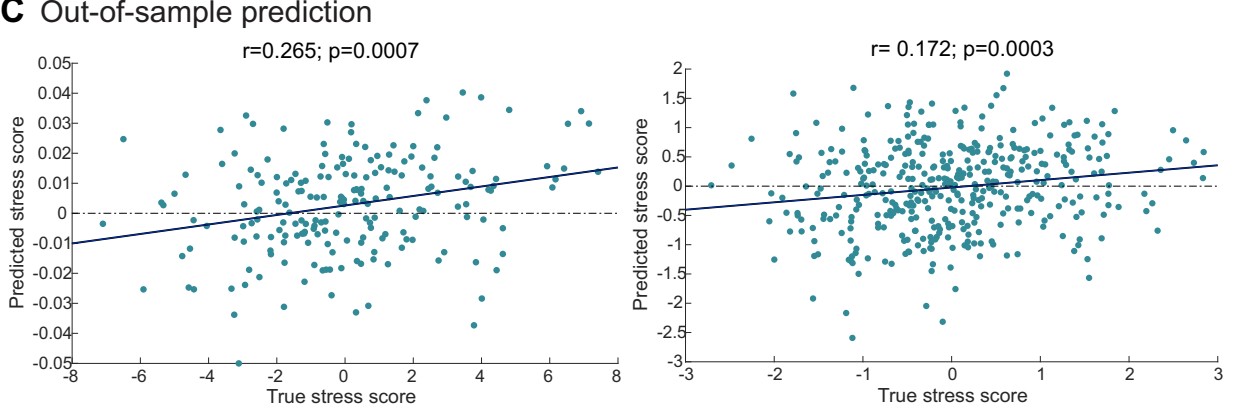

**Fig. 3 | Relating stress to interindividual variation in hypothalamus nuclei connectivity. A** Relationships between stress scores and hypothalamus connectivity were robust and replicable. We used robust regressions to characterise relationships between the functional connectivity in each edge (7 hypothalamus nuclei x 15 ROIs) and stress and tested whether the resulting patterns of regression coefficients (shown in panel B) were more similar than expected by chance. This was true (1) across two 3 T datasets (left: $n = 198$ train and $n = 200$ test participants; one-sided $p$-values from a non-parametric test using permutation null distributions) and (2) within subjects when comparing patterns extracted from the 1st and 2nd half of each individual's data (right: $n = 398$ combined 3 T participants, run 1 + 3 versus run 2 + 4; one-sided $p$-values from a non-parametric test using permutation null distributions). Null distributions were generated using 10,000 iterations of shuffled stress scores. Note that the split into train and test groups was performed to achieve comparable stress distributions (Supplementary Fig. 4C). **B** Visualisation of robust regression coefficients capturing relationships with stress (1) across test

and train 3 T datasets (left: top and middle row) and (2) across run 1 + 3 vs. run 2 + 4 (right: top and middle row). This illustrates the similarity of the patterns statistically evaluated in panel A. We extracted the contribution of each edge to this similarity by calculating the difference in correlation coefficient between the patterns obtained when excluding this edge versus including all edges (rDiff, bottom row). Visual inspection of rDiff values highlights strong similarities between the two replications (left vs right) and thus anatomical specificity. **C** Regression coefficients estimated from the training group were applied to the functional connectivity of the test group (left) to calculate predicted stress scores. This showed significant out-of-sample predictions of true stress scores (independent one-sided correlations to assess the positive relationship between predicted and true stress scores). The same was true when repeating the procedure using the first and second half of data (first half of each session: run 1 + 3 against the second half of each session: run 2 + 4). Source data for 3**A**, **B** are provided as a Source Data file. Abbreviations as in Fig. 1.

### Out-of-sample stress prediction using hypothalamus coupling

The similarity in regression coefficients across datasets and experimental halves suggests that hypothalamus nuclei connectivity contains variance related to stress. Next, we asked whether it would be sufficient to predict individual stress scores. We used all 105 (7 nuclei x 15 ROIs) robust regression coefficients estimated in the training group ($n = 198$ 3 T participants) and multiplied them with the functional connectivity values measured for all 105 edges in the *test* group ($n = 200$ 3 T participants, same split as above). This produced predicted stress scores for the test group which had been generated out-of-sample. We found a significant relationship between predicted and true stress scores in the test group (one-tailed correlation testing for a positive relationship between predicted and true stress scores: $r(196) = 0.265$; $p < 0.001$, CI $= [0.13, 0.39]$; Fig. 3C). This prediction was robust to slight changes in the weights derived from the training group (Supplementary Results). However, when we repeated the same procedure using all $n = 398$ 3T-participants as the train group to generate out-of-sample predictions for the $n = 97$ 7T-participants, the prediction was not significant, possibly due to the lack of variance in stress scores in the 7 T cohort ($r(95) = 0.118$, $p = 0.124$, CI $= [−0.08, 0.31]$; Supplementary Fig. 6C).

Finally, we also generated predicted stress scores using regression coefficients estimated on the first half of resting-state data applied to connectivity values estimated from the second half of data, across all $n = 398$ 3 T participants. Again, stress predictions were significant (Fig. 3C, right panel: $r(396) = 0.172$, $p < 0.001$, CI $= [0.08, 0.27]$).

### Characterising hypothalamus networks predictive of stress

The network between hypothalamus nuclei and a priori subcortical ROIs included here contains less than 2% of all brain ordinates (1.38%: 1262 of 91,282) and 0.0015% of all edges (105 edges of 91,282 × 91,282). Nevertheless, we were able to generate significant out-of-sample stress predictions. Figure 3B already suggested that specific edges of the network may be especially relevant for stress. Thus, in our next step, we tested whether an even smaller network might improve stress predictions.

To carry out this test, we iteratively added one edge at a time, from 1 to 105, based on their order of importance (absolute regression coefficient) in the training group. As before, we then applied training weights to the functional connectivity of the test group to compute out-of-sample stress predictions, only this time predictions were based on a smaller number of edges (Fig. 4A). Neighbouring predictions only differ by the inclusion of one additional edge, so rather than establishing statistical significance at each step (amounting to a total of 105 tests), we used this analysis to identify the smallest significant network and the overall best prediction, as established in prior work (see Methods and ref. 5).

Surprisingly, a network that included a single edge between the most ventral and anterior part of the hypothalamus (SO/SC) and the median raphe (RNMR) was sufficient to significantly predict stress levels ($r = 0.130$, Fig. 4A and Supplementary Fig. 5B). A significant prediction with $n = 1$ edge was unlikely to occur by chance (one-sided likelihood of this $n$-value given permutation null distribution: $p = 0.053$, trend-wise significant). The top prediction was achieved using n = 22 edges ($r = 0.272$; Fig. 4A; Supplementary Fig. 5B), and this peak prediction was higher than expected by chance (one-sided likelihood of this $r$-value given permutation null distribution: $p = 0.002$). We noticed several interesting features of the resulting network (Fig. 4B–D). Anterior ventral hypothalamus (SO/SC) coupling with median raphe (RNMR) and several amygdala nuclei (LaI, LaD, AB, CoN) positively predicted stress scores. In addition, coupling between more dorsal anterior hypothalamus nuclei (MPO and PV) and BNST and central/basal amygdala was important. For BNST, coupling with MPO/PV related negatively to stress, while for the central and basal

amygdala, relationships were positive. For PV, connectivity with NAc and LC also negatively contributed to predicting stress levels. Finally, hypothalamus connectivity with dPAG and RNMR tended to correlate positively with stress. Here, coupling with more mid-to-posterior hypothalamus nuclei VM and PH seemed most important. We note that functional connectivity was on average positive in the hypothalamus network considered (Fig. 1F and Supplementary Fig. 5A). This means that positive regression coefficients can be interpreted as stronger and negative regression coefficients as weaker functional connectivity. Thus, in brief, weaker functional connectivity between anterior hypothalamus nuclei and NAc and BNST, but stronger functional connectivity between anterior hypothalamus and amygdala related to stress, while relevant functional connectivity with brainstem structures dPAG and RNMR was with more mid/posterior hypothalamus nuclei and also positively related to stress (Fig. 4C, D). For predictions of stress scores in the 7 T cohort, using identical procedures, see Supplementary Results. Similarly, for predictions including hippocampus-subdivisions as additional ROIs, see Supplementary Results and Supplementary Fig. 8.

### Nuclei-specific versus whole hypothalamus predictions

To show that parcelling the hypothalamus improved stress predictions, we also repeated the regression with edges from ROIs to the entire hypothalamus (reducing it from 105 to 15 edges). All methods were otherwise identical. Out-of-sample predictions were still significant but slightly worse than those obtained with nuclei-specific hypothalamus predictions (prediction using whole network: $r = 0.244$ compared to $r = 0.265$, both $p < 0.001$, null distributions account for the number of predictors, see Methods, Fig. 5A, B left; peak prediction with increasing numbers of connections: $r = 0.252$ compared to $r = 0.272$; Fig. 5C left). The associated network resembled the nuclei-specific networks in Figs. 3 and 4 closely with connections to RNMR and the amygdala being the most important (Fig. 5D—left column). We repeated this procedure with the amygdala nuclei merged into one ROI and with and without nuclei resolution in the hypothalamus ($7 × 9 = 63$ versus 9 edges, respectively; Fig. 5 right). When losing the nuclei composition of both the hypothalamus and amygdala, prediction accuracies considerably dropped, but continued to be significant (whole-network prediction: $r = 0.161$, $p = 0.011$; peak prediction: $r = 0.202$; Fig. 5A–C, right). Hypothalamus with RNDR coupling remained important, but amygdala-to-hypothalamus connectivity lost its importance for predicting stress when both structures were each treated as homogenous wholes (Fig. 5D, right).

### Behavioural specificity of predictions for stress

In a second control analysis, we examined the behavioural specificity of hypothalamus predictions for stress. We included four behavioural dimensions extracted in prior work[5] which capture interindividual variation in (a) social and life satisfaction, (b) negative emotions, (c) sleep problems and (d) anger/rejection. These dimensions are not fully orthogonal to stress: the first two correlate strongly with stress: HCP participants with high stress scores also tend to have low life satisfaction ($r = −0.799$) and high negative emotion scores ($r = 0.903$; Fig. 6D). Anger/rejection is somewhat correlated with stress ($r = 0.627$), but sleep problems less so ($r = 0.296$).

Nevertheless, when repeating predictions for these four alternative dimensions, using identical methods, we found that hypothalamus connectivity predicted stress scores better than the four alternative dimensions. The behaviours most related to stress, life satisfaction, negative emotions, and anger/rejection, also reached significance, with negative emotions coming closest to stress (whole-network prediction: $r = 0.188$ for negative emotions, $r = 0.265$ for stress; peak prediction: $r = 0.244$ for negative emotions (including 44 edges), $r = 0.272$ for stress; Fig. 6A–C).

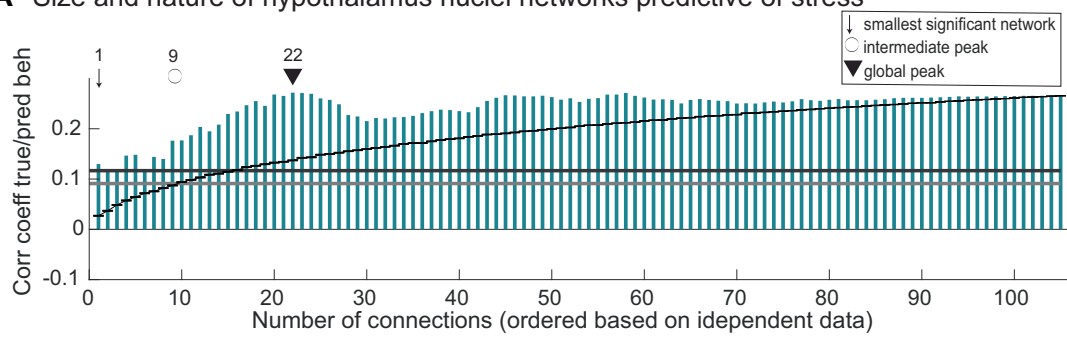

**A** Size and nature of hypothalamus nuclei networks predictive of stress

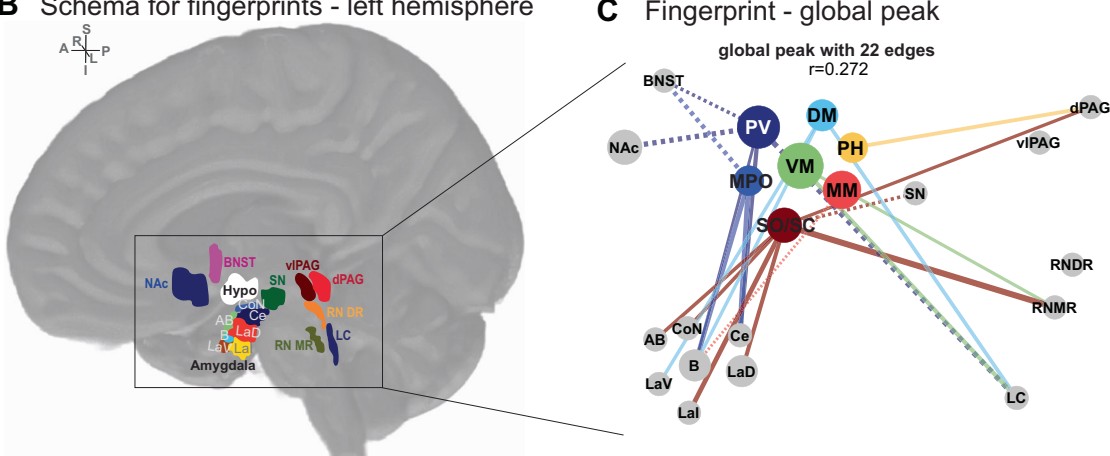

**B** Schema for fingerprints - left hemisphere

**C** Fingerprint - global peak

**D** Fingerprints - smallest significant and intermediate peaks

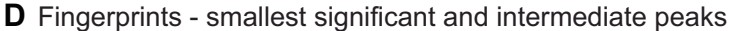

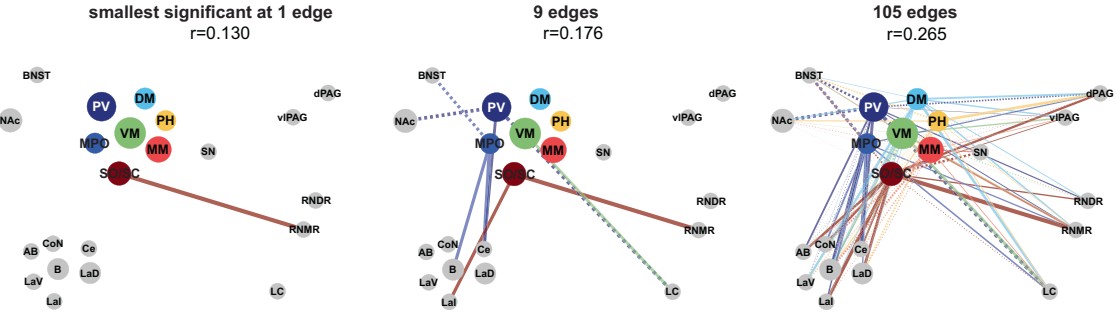

**Fig. 4 | Anatomical features of hypothalamus networks predictive of stress.**
**A** Prediction of stress scores obtained using subsets containing between 1 and 105 edges. Edges were included in order of importance (absolute robust regression coefficients) estimated from the training group ($n = 198$). As in Fig. 3, weights were applied to test participants' functional connectivity values to predict stress scores, but this time with increasing numbers of edges. Prediction accuracies (coloured bars) are shown as the correlation between true and predicted stress scores in the test participants ($n = 200$), but were only statistically evaluated at the peak (black triangle: 'global peak') and to derive the smallest number of edges that reached a significant out-of-sample prediction (black arrow, 'smallest significant network'; see Methods for generation of null distributions using permutation; for precise one-sided $p$-values in each case, see Results); black curve indicates performance using a given number of edges (on $x$) but included in random order ($n = 10,000$ shuffles; error bars denote s.e.m.); black line at $r = 0.1167$ indicates the threshold for significance at $P < .05$ purely for visualisation (grey line: $P < 0.1$, $r = 0.0911$). A subset

of all 105 hypothalamus-to-ROI edges was sufficient to achieve a significant out-of-sample prediction. A significant prediction could be achieved using only the first edge (SO/SC to RNMR). The best prediction was achieved using 22 edges ($r = .272$). **B** Schematic for the arrangement of connectivity fingerprints. **C** Fingerprint highlights anatomical edges associated with the global peak (22 edges). Fingerprints show hypothalamus nuclei in the centre (colour-coded); ROIs are positioned roughly according to their anatomical location. Line width denotes the size of the absolute regression coefficient in the training dataset; line style denotes its sign (continuous, positive; dashed, negative). **D** Further fingerprints are shown for the smallest significant network (1 edge) and for one arbitrarily chosen intermediate step at 9 edges, and using all 105 edges, for visualisation. The mean baseline connectivity for all 105 edges as well as the corresponding scatterplots are shown in Supplementary Fig. 5. Source data for 4**A**, **C**, **D** are provided as a Source Data file. Abbreviations are as in Fig. 1.

## Discussion

Chronic stress frequently precedes the onset of mental ill-health[54,55]. Thus, there is an urgent need for an improved understanding of the biological underpinnings of prolonged stress. We investigated whether functional connectivity in a small subcortical network centred on

the hypothalamus, a key region of the HPA axis, is associated with a dimensional marker of prolonged stress. We first parcellated the human hypothalamus into seven anatomically plausible nuclei. Inter-individual variation in stress was then related to functional connectivity extracted from high-quality fMRI resting-state data in nearly

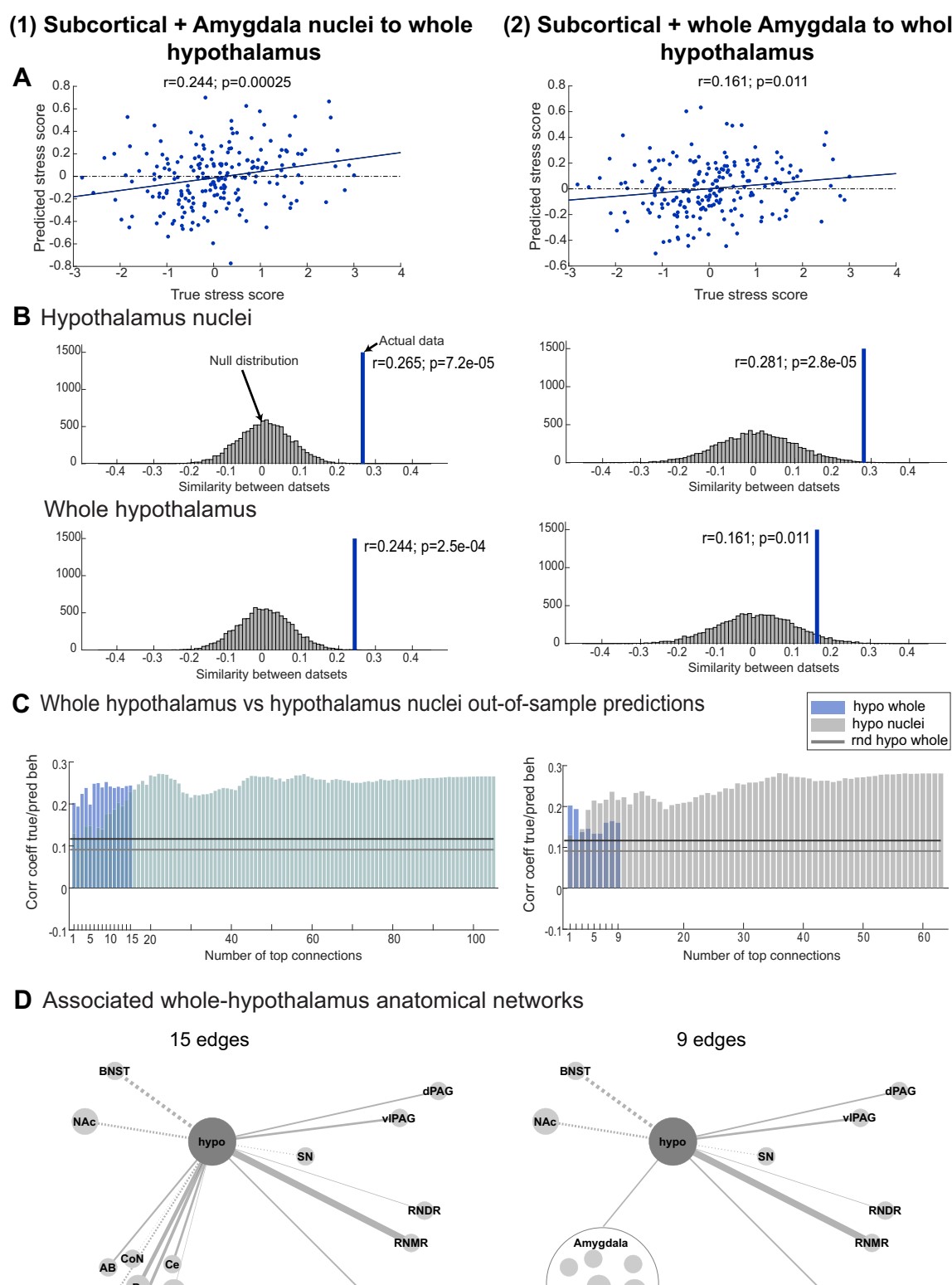

**(1) Subcortical + Amygdala nuclei to whole hypothalamus**

**(2) Subcortical + whole Amygdala to whole hypothalamus**

**A**

**B** Hypothalamus nuclei

Whole hypothalamus

**C** Whole hypothalamus vs hypothalamus nuclei out-of-sample predictions

**D** Associated whole-hypothalamus anatomical networks

15 edges                9 edges

500 participants. The functional coupling strength of connections between individual hypothalamus nuclei and subcortical and brainstem ROIs was robustly related to markers of stress, achieving predictions in held-out data that were specific to stress and which were enhanced when using nuclei as opposed to whole hypothalamus coupling.

The hypothalamus is a small deep structure prone to signal dropout. Nevertheless, we found that with sufficient spatial resolution and careful pre-processing, it was possible to identify reliable signals in the hypothalamus. For example, the average whole-hypothalamus connectivity closely resembled monosynaptic connections identified in macaque tracer work such as coupling with pgACC and sgACC, insula,

**Fig. 5 | Parcellating the hypothalamus improves predictions of stress.** Sensitivity of whole-hypothalamus as opposed to nuclei-specific hypothalamus connectivity was established using two control analyses: (1) using the original 15 ROIs (left column) and (2) using the original ROIs but replacing 7 amygdala nuclei by one whole amygdala mask (9 ROIs, right column). **A** Out-of-sample predictions of test group participants using weights estimated on the training group (as in Figs. 3C and 4A, n = 198 and n = 200) were significant based on whole-hypothalamus connectivity both when using amygdala nuclei or whole amygdala connectivity (one-sided correlations to assess positive relationships). **B** However, regression coefficients were reduced in both cases (bottom row) compared to the equivalent prediction using hypothalamus nuclei-specific functional connectivity (top row), even when correcting for the difference in the number of predictors (one-sided p-values from non-parametric test using permutation null distributions, see Methods). **C** As in Fig. 4A we generated predictions using increasing numbers of edges (coloured bars): left: whole hypothalamus x ROIs (1 × 15) vs. nuclei version (7 × 15 = 105); right:

whole hypothalamus + whole amygdala ROIs (1 × 9) vs. nuclei version (7 × 9 = 63). Peak predictions were higher in both cases when using hypothalamus nuclei and highest overall when subdividing both hypothalamus and amygdala into their component nuclei (left: 22 edges: r = 0.272; right: 36 edges, r = 0.282). **D** Fingerprints associated with the predictions using all 15 (left) and 9 (right) whole-hypothalamus edges; on the right, the amygdala was also treated as a single structure. The highest contributing edge was hypothalamus coupling with RNMR in both cases. Source data for 5B–D are provided as a Source Data file. rnd randomised, NAc nucleus accumbens, BNST bed nucleus of the stria terminalis, SN substantia nigra, dPAG dorsal periaqueductal grey, vlPAG ventrolateral PAG, RNDR dorsal raphe nuclei, RNMR median raphe nuclei, LC locus coeruleus, Ce central amygdala nucleus, CoN cortical amygdala nuclei, B basal amygdala nucleus, AB/BM auxiliary basal or basomedial amygdala nucleus, LaI lateral intermediate amygdala nuclei, LaD lateral dorsal amygdala nuclei, LaV/BL lateral ventral portion containing portions of basolateral amygdala nucleus.

anterior temporal cortex, NAc, central and superficial (CoN) amygdala and brainstem[35–39] (Fig. 1).

In line with this, our parcellation of the hypothalamus into individual nuclei largely agreed with boundaries identified in careful anatomical and post-mortem work in human and non-human primates and extended simpler parcellations based on human rs-fMRI data[22,34,40,56–58] (Fig. 1 and Supplementary Table 2). We showed the robustness of our findings by replicating the average hypothalamus connectivity and its parcellation in two independent datasets (Supplementary Fig. 1). Our parcellation was driven by the whole-brain connectivity pattern of each hypothalamus voxel. More precisely, we used the absolute connectivity between each hypothalamus voxel and the rest of the brain. When using signed connectivity values, correlation coefficients will be driven by negative versus positive connectivity differences of the studied region with the rest of the brain. By contrast, when using absolute connectivity values (or just the positive or just the negative half of all connectivity values, ignoring the other half), it will be driven by strong versus weak connectivity differences. In our hands, the parcellation was anatomically most plausible when using absolute connectivity values (see Supplementary Fig. 7). Future work may explore other parcellation approaches, for example using functional gradients previously explored in the striatum[59]. For a detailed discussion of our choice of nuclei labels and their relationship with previous parcellations, see Supplementary Discussion and Supplementary Table 3.

We found that hypothalamus nuclei coupling was robustly associated with stress. Both chronic and immediate stress has been associated with the HPA axis in animal models[14–16,60,61], but detailed investigations in humans have been lacking, although a general role of the hypothalamus in mood disorders has been highlighted[62,63]. Here, we build on such prior work, but extend it in several ways: first, by using a dimensional approach in a large healthy cohort, instead of broad disease classifications. Second, by improving the anatomical scale and considering individual hypothalamus nuclei, thus matching the circuit resolution typically considered in related animal work. And third, by providing statistically robust and replicable predictions of stress scores in independent cohorts. Using this approach, we were able to identify functionally meaningful hypothalamus networks.

Several features of the network we identified are noteworthy. First, the paraventricular nucleus was a key nucleus: PV's coupling with the nucleus accumbens, BNST and amygdala strongly contributed to stress predictions. PV is an important autonomic centre[64,65] and rat PV neurons show increased excitability to chronic stress[66]. In addition, amygdala projections to the anterior hypothalamus[35,67,68] are important for mobilising stress responses[45,69,70]: corticotropin-releasing central amygdala neurons terminate in PV[46] and suggestions that rat central amygdala to PV projections mediate stress date back several decades[71]. Yet so far, it has been difficult to examine the insights derived from this body of work in humans. Here we find that functional

connectivity between basal and central amygdala nuclei and PV - as well as adjacent anterior MPO—predicts stress in humans. This aligns well with the rodent literature and improves the anatomical detail and spatial resolution at which these circuits are typically studied in humans.

When considering hypothalamus connectivity with brainstem nuclei, we discovered an interesting dichotomy: coupling between hypothalamus and dPAG as well as median raphe tended to relate positively to stress, while coupling between hypothalamus and vlPAG as well as dorsal raphe tended to relate more weakly to stress. Median raphe generally inhibits, while dorsal raphe generally facilitates stress behaviours[47]. The stronger coupling between the median raphe and hypothalamus that we observed as a function of stress may reflect a process of down-regulation of serotonin stress responses. The connection between SO/SC and median raphe particularly stood out as one of the strongest predictors of stress. However, the opposite effect was observed in PAG. DPAG and vlPAG typically activate versus inhibit active coping strategies, respectively[72]. However, coupling between dPAG and the hypothalamus was stronger in people with higher stress scores, suggesting prolonged stress may mean an over-usage of active coping mechanisms mediated by this circuit. Here, important edges centred around posterior hypothalamus nucleus PH, consistent with animal work highlighting these pathways[47,48,73,74]. Given our analyses relied on healthy volunteers, our interpretation of the underlying circuit changes should be confirmed in studies using longitudinal designs or direct interventions in clinical populations to establish causality.

The hippocampus is affected by stress, in particular early life adversity, which can have lasting effects on its structure and function[75,76]. It is therefore an important subcortical region not included a priori here. We did not want to treat the hippocampus as a homogenous region; however, since starting this work, Tian and colleagues[59] published a detailed hippocampus parcellation, which we included a posteriori in our analyses for comparison (Supplementary Information and Supplementary Fig. 8). Surprisingly, the inclusion of the hippocampus subregions as additional ROIs did not improve stress predictions. While these new analyses do not question the importance of the hippocampus in stress predictions per se, they do suggest functional connectivity between specific hippocampus-subdivisions to hypothalamus-nuclei does not carry additional predictive variance for stress. We note, however, that the marker of stress used in this study was derived from subjective questionnaire items such as perceived stress and fear. As part of the HCP dataset, we did not have access to a true chronic stress item (e.g., burnout) or a validated stress marker such as the cortisol response. Coupling between the hippocampus and regions other than the hypothalamus, or as a function of objective stress markers, should be considered in future work.

We found that stress predictions were better when considering nuclei as opposed to whole hypothalamus connectivity, and they were behaviourally specific. In our healthy cohort, we captured

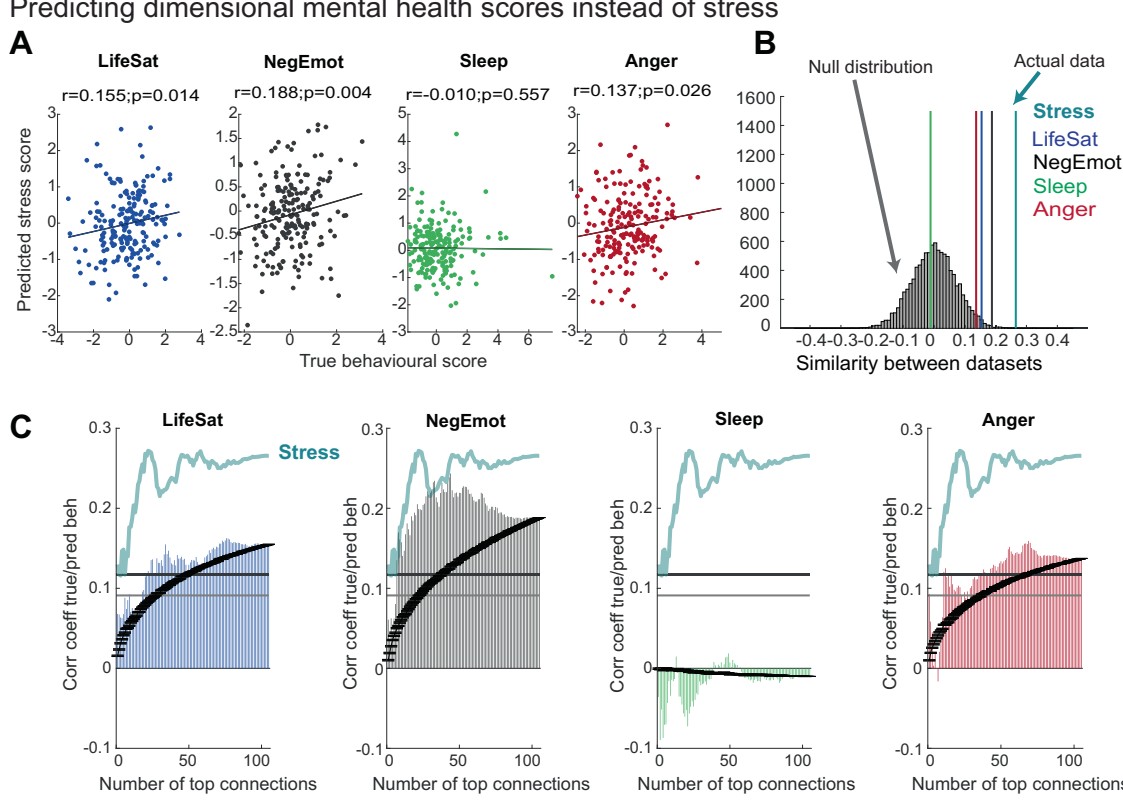

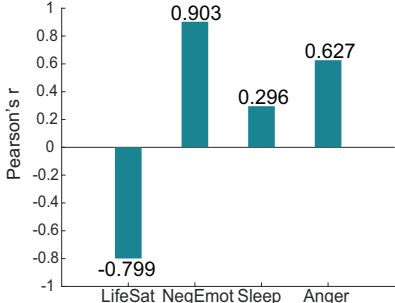

**Fig. 6 | Behavioural specificity of hypothalamus connectivity predictions for stress. A** Out-of-sample predictions were achieved using nuclei-specific hypothalamus connectivity (all 105 edges as in Fig. 3C) for four alternative mental health dimensions: life satisfaction (LifeSat), negative emotions (NegEmot), Sleep and Anger (see ref. 5 for details). Significant predictions were achieved for Lifesat, NegEmot, and Anger, but not Sleep (one-sided correlations to assess positive relationships). **B** However, in all cases, prediction accuracies were lower than those achieved for stress, showing hypothalamus connectivity is particularly meaningful for stress. **C** Predictions achieved with increasing numbers of connections (as in Figs. 4A and 5C) show overall greater predictions for stress (turquoise line) compared to the four alternative dimensional scores when considering the peak and the smallest predictive network. **D** Correlations between dimensional control scores and stress scores show some shared variance, in particular for life satisfaction (negative) and negative emotions. Source data for 6B, C are provided as a Source Data file.

interindividual variability in stress along a continuous subclinical dimension which reflected longer-term stress rather than an immediate short-lived hormonal stress reaction. High stress scores correlated with negative emotions and anger/rejection, and inversely with social and life satisfaction. As expected, these alternative mental health dimensions could also be predicted from hypothalamus coupling. However, predictions were consistently strongest for stress (Fig. 6). Given our work solely relied on healthy participant's data, future longitudinal work should examine whether changes in hypothalamus coupling may be a precursor to mental illness and whether targeted interventions such as those using transcranial ultrasonic stimulation (TUS[77,78]) might be able to rebalance the hypothalamus networks identified here[79–81].

Direct insight into process abnormalities in stress disorders such as PTSD cannot be gained from our work with healthy participants. An important question for future work is therefore whether brain changes in stress disorders such as PTSD are categorically different from healthy controls, or just vary further along the continuum examined in a healthy population here. In addition, it would be interesting to examine whether the same networks studied at rest here would be affected during an external or internal stress challenge, as well as after prolonged exposure to stress.

Importantly, from the work presented here, we can only conclude that predictions of stress can be achieved when based on hypothalamus functional connectivity. We cannot draw conclusions about the contributions of other brain hubs not investigated here. Previous work

has shown more widespread whole-brain as well as specific amygdala/hippocampus contributions in the context of acute stress[82–85]. This study was designed around the hypothalamus, a region that has been relatively neglected in human neuroimaging, to examine whether this small subcortical circuit carries relevance for predicting stress. However, similar prediction may be possible using other circuits. Indeed, it is likely that other regions known to play a role in the context of stress, both in the subcortex (e.g., amygdala, hippocampus) and cortex (e.g., ACC) contribute to stress predictions and could improve prediction accuracies. Related to this, the signal-to-noise ratio can vary considerably across regions, and this would need to be considered when making any comparison between the predictive information about stress in cortically- and subcortically-centred networks.

We note several limitations of our work. Our parcellation achieved good subdivisions of the hypothalamus in the anterior-to-posterior and dorsal-to-ventral axis. However, at a higher spatial resolution, it might be possible to identify lateral and medial subdivisions of the hypothalamus[86] which was not possible here. The lateral nucleus is particularly important in the context of feeding behaviour and for acquiring and expressing cue-reward associations. It thus has an important role in learning[87]. It has been identified in previous human work at higher resolution, including post-mortem studies[22,34,42]. Its importance in stress and its functional connectivity fingerprint should be considered in future investigations when data acquired at higher spatial resolutions and with better signal-to-noise become available. In our main parcellation (Fig. 1), the lateral hypothalamus was likely part of the paraventricular nucleus. A lateral nucleus appeared at a resolution of 1.6 mm at 7 T (Supplementary Fig. 3a). However, the 7 T data suffered from other problems such as a lack of variance in stress scores and missing physiological recordings required for data clean-up. We also note that while our work shows the importance of focusing on functionally relevant circuits, future work might additionally consider functionally relevant states, instead of the resting state, potentially including internal and external stress challenges or longitudinal measures of stress, which might further aid predictions[88]. In general, all predictions here are based on cross-sectional data and longitudinal work will be vital in helping establish causality and in clarifying the direction of relationships between brain and stress markers. Finally, in normal population data, we are likely to underestimate effect sizes compared to clinical samples, so replication of our results in the latter will be important.

In summary, we examined whether a subcortical brain network is associated with stress scores in a large healthy human population. Rather than identifying an extensive network for maximal prediction accuracy, we focused on a small network centred on the hypothalamus which involved less than 2% of brain ordinates and 0.0015% of brain edges. We characterised hypothalamus connectivity at the scale of its individual subnuclei. Using this nucleus-specific functional connectivity, we were able to significantly predict stress scores in an independent cohort. Predictions were behaviourally specific, improved by nucleus level resolution (as opposed to whole hypothalamus level resolution) and were consistent with a large body of work in animal models. Future interventional studies should explore whether inducing changes in precise hypothalamus networks may alter an individual's experience of stress. Exploring this in both healthy and clinical populations could have implications for treating a range of mental health disorders whose onset is often preceded by a prolonged experience of stress.

## Methods
### Participants
Data and ethics were provided by the Human Connectome Project (https://www.humanconnectome.org/), WU-Minn Consortium (Principal Investigators: David Van Essen and Kamil Ugurbil; 1U54MH091657), funded by the 16 NIH Institutes and Centres that support the NIH

Blueprint for Neuroscience Research; and by the McDonnell Centre for Systems Neuroscience at Washington University. All participants gave informed consent and were reimbursed for their time ($450 for 3 T MRI + interview, $400 for 7 T MRI) and travel. HCP participants were scanned at the McDonnell Centre for Systems Neuroscience at Washington University, University of Minnesota (WU-Minn), USA, on a Siemens Skyra 3 Tesla scanner and on a 7 T Siemens Magnetom scanner (for details see refs. 89, 90).

We included three cohorts from the full young-adult HCP dataset (Supplementary Fig. 2B): an initial dataset of $n = 200$ 3 T participants (D1), a replication dataset of the same size ($n = 200$, 3 T; R1), and a third dataset containing all non-overlapping $n = 98$ participants scanned at 7 T (R2). These datasets are identical to those included in our prior work where we explain the participant inclusion criteria in depth[5]. For completeness, we will reproduce the resulting demographics and a brief description of our inclusion criteria here: D1: mean age 29 ± 0.26; age range 22–36; 108 females; R1: mean age 28 ± 0.28; age range 22–36; 99 females; R2: mean age 29 ± 0.33; age range 23–36; 59 females. In brief, we included participants with four complete resting-state runs and good-quality recordings of both cardiac and respiratory activity. These were not available for all individuals from the larger dataset of $n = 1206$ HCP participants, but they were important because one key aspect of our additional data pre-processing was to correct rs-fMRI data for physiological noise[5,26,41]. Physiological noise recordings were not available in the 7 T data, but the higher field strength improved the tSNR in subcortical regions, so all 7 T participants not already included in the 3 T cohort were included in our third (second replication) dataset. Finally, 3 T participants were selected to achieve a widespread in their mental well-being scores (for more details, see ref. 5).

### Data and data pre-processing
Four runs of rs-fMRI data were acquired in each participant, lasting 14.4 min per run at 3 T and 16 min per run at 7 T. For the 3 T data, two of these runs had a right-left encoding direction and two had a left-right encoding direction (TR of 720 ms, TE of 33 ms, isotropic resolution of 2 mm, 72 slices, multiband factor of 8 resulting in 1200 time points). For the 7 T data, two of these runs had an anterior-posterior encoding direction and two had a posterior-anterior encoding direction (TR of 1 s, TE of 22.2 ms, isotropic resolution of 1.6 mm, 85 slices, multiband factor of 5 resulting in 900 time points, in-plane acceleration factor of 2).

We used all four runs per participant and downloaded the minimally pre-processed data[91]. Among other steps, the minimal pre-processing pipeline included distortion correction, temporal filtering, projection onto a surface reconstruction obtained from the T1-weighted image while keeping subcortical voxels in volume space (cifti format), and minimal smoothing. These steps were achieved using tools from FMRIB's software library (FSL, version 6.0), FreeSurfer (version 5.2) and Connectome Workbench (version 1.3.2). A multi-modal areal-feature-based surface registration (MSMall) method was used to achieve registration across participants (for more details, see refs. 91, 92). Unless otherwise stated, the 2 mm subsampled 7 T data was used to match the resolution of the 3 T data.

In addition to minimal pre-processing, we corrected rs-fMRI data for physiological noise. This was a key aspect of our pre-processing as physiological noise particularly affects subcortical and brainstem regions and thus, the hypothalamus and its connected regions. This step and its resulting benefits in terms of the achieved temporal signal-to-noise (tSNR) ratio are explained in more depth in our prior work[5], but we summarise the key aspects again here: recorded raw physiological traces were processed using the Physiological Noise Modelling (PNM) toolbox[93,94] to generate 33 physiological regressors. The physiological regressors were created based on voxel x,y,z = [0,0,0] given slices were acquired interleaved and with a very short TR (720 ms). The variance explained by these regressors was then removed from the rs-

fMRI data. As described before[5], we also corrected for motion and noise detected using independent component analysis (ICA). The resulting rs-fMRI data were demeaned and the variance of the noise in the data was normalised as described in ref. [92], i.e., the normalisation factor was estimated on unstructured noise after the strongest PCA components containing structured signal/noise were removed.

Once data clean-up was complete, we concatenated the four resting-state runs from each participant. We obtained time courses of all brain-ordinates of all 400 3 T participants (D1: *n* = 200; R1: *n* = 200; 91,282 × 4800 from the four combined runs of 1200 time points each) and all 98 7 T participants (R2: *n* = 98; 91,282 × 3600 from the four combined runs of 900-time points each). For the creation of each of the three group connectomes (D1, R1, and R2), we applied additional smoothing to the surface, but not to subcortical structures (sigma = 5 mm). Analyses on individual time series did not involve any further smoothing.

To obtain a group average connectome for each dataset, the individual time series of all participants included in D1, R1, or R2, respectively, were combined using the following procedure: first, we generated the group average time series using the algorithm MIGP, which creates an average time series using a group-level principal component analysis[95]. Second, we created a dense connectome (i.e., including the connectivity from each brain-ordinate to each other brain-ordinate) by generating correlations of the group average time series (using a Pearson's correlation between each pair of brain ordinates followed by a transformation to Fisher's *z*). Ringing artefacts were corrected using Wishart RollOff[92]. This produced three group dense connectomes containing *n* = 200 (D1, R1) or *n* = 98 (R2) participants.

## Group dense connectome and hypothalamus similarity matrix

The average dense group connectome (91,282 × 91,282 brain-ordinates) was restricted to the connectivity between all hypothalamus voxels and the rest of the brain (109 voxels x 91,282 brain-ordinates) using a hypothalamus mask extracted from Zhou[96]. Out of the 91,282 brain vertices, 31,870 were subcortical voxels and 59,412 were cortical vertices. Figure 1A shows the group mean functional connectivity for the entire hypothalamus for D1, whereby positive functional connectivity values are shown in red-yellow colours and negative functional connectivity values in blue-green colours (see Supplementary Fig. 1A, B for the corresponding plots in R1 and R2). For illustration purposes, we corrected Fig. 1A and Supplementary Fig. 1A, B for global absolute signal strength[5]: raw connectivity values in each brain-ordinate were divided by the mean absolute connectivity of this brain-ordinate to the whole brain as a proxy for its signal strength. Next, absolute values of the restricted hypothalamus-to-all dense connectome were converted into a similarity matrix that summarised the similarity between any pair of hypothalamus voxels in terms of their absolute functional connectivity to all other brain-ordinates (calculated using Pearson's correlation coefficient in FSLnets function nets_netmats, https://fsl.fmrib.ox.ac.uk/fsl/fslwiki/FSLNets). The self-connectivity of the hypothalamus was included for generating this similarity matrix because it might be informative for parcellating the hypothalamus into finer subdivisions. Using absolute values meant that the clustering was driven by similarities and differences in high versus low functional connectivity, rather than positive versus negative functional connectivity values. Note that almost identical clusters were obtained when using only positive connectivity values to parcellate the hypothalamus. By contrast, a parcellation using signed values was less anatomically plausible, lacking symmetry and producing unequal numbers of nuclei across hemispheres. For computing absolute values, the negative 46.65% of the connectivity matrix was flipped in its sign. Overall, the hypothalamus parcellation was thus driven by similarities and differences between hypothalamus voxels' absolute connectivity to the rest of the brain.

## Parcellation of the hypothalamus

Hierarchical clustering (function nets_hierarchy.m part of FSLnets) was then performed on the similarity matrix. As hierarchical clustering is an agglomerative approach related to similarity values, voxels with similar connectivity profiles should end up in the same cluster, while voxels with different connectivity profiles should end up in separate clusters. We chose the most anatomically plausible parcellation by comparing each higher-order clustering step with the previous clustering step (see Supplementary Fig. 2) and by comparing the location and size of the clusters obtained at increasing levels of depths to known anatomical subdivisions of the hypothalamus. In other words, while the clustering itself was fully automated, the depth of clustering at which we decided to stop was subjective and based on anatomical considerations. We expected reasonable symmetry across hemispheres and focused on identifying divisions between anterior-posterior and medial-lateral clusters within the hypothalamus. We therefore stopped at the clustering depth where these criteria started to break down. To evaluate the quality of the clustering, we compared clusters with existing post-mortem images[58], and MRI work[34,40,42,43,57,94,97] as well as with atlases[22,42,98] to identify similarities in their anatomy. Supplementary Fig. 9 and Supplementary Table 3 summarise similarities and differences between our and other existing parcellations.

To replicate the parcellation, we repeated the same procedure on the second 3 T dataset, R1, and on the n = 98 7 T participants in R2, for the latter using the improved resolution of 1.6 mm to examine whether we could additionally achieve a medial-lateral division of the hypothalamus (Supplementary Fig. 1C). We quantified the similarity between the two 2mm-parcellations obtained for D1 and R1 as described by Klein-Flügge and colleagues[5] using the mean distance between cluster centroids and the percentage of voxel overlap with the same label. The overlap between the original parcellation in D1 and the 3 T replication in R1 was 60.55% (meanDist = 0.7715, which is unlikely by chance: non-parametric tests using permutation null distribution *p* = 0.00001 and *p* = 0.0004, respectively, see Supplementary Fig. 3B, C). Throughout the manuscript, we used the parcellation derived from D1 for all cohorts. This meant that the same hypothalamus clusters were used across all participants and datasets. Also, because this parcellation was derived from the group mean, the choice of parcellation could not bias any key analyses focusing on individual differences. See Supplementary Fig. 2B for an overview of the data used for each step of analysis.

## Naming of clusters

The naming of our clusters is based on the nomenclature of the clusters in the Atlas of the Human Brain by Mai and colleagues[22] (http://atlas.thehumanbrain.info/) and the very fine-grained MRI-based parcellation by Neudorfer and colleagues[34]. We compared the similarities between our labelling and the labels of other parcellations from existing post-mortem images[58], and MRI work[34,40,42–44,57,94,97,99] as well as with atlases[22] and summarised similarities in Supplementary Table 3. All clusters are shown with their corresponding labels in Fig. 1D. Overall, we identified a good correspondence between the position and labelling of our nuclei and those reported in refs. 22, 34, 42–44, 57, 58, 99.

## ROI selection for nuclei-specific hypothalamus connectivity

This study focused on the connectivity of the hypothalamus with subcortical regions. All ROIs were chosen a priori based on the literature in rodents[35–39,100,101], monkeys[35,37,38,102] and humans[103]. More specifically, we selected forebrain subcortical and brainstem ROIs which have been shown in tracer studies to be strongly connected to the hypothalamus, such as the amygdala[35–39], nucleus accumbens (Nac[104]), and PAG[100–102,105]. In addition, we included ROIs that have been linked to stress-related functions and/or form part of the major neurotransmitter systems. In the brainstem, this includes the ventral tegmental area (VTA), substantia nigra (SN), locus coeruleus (LC) and

raphe nucleus (RN). The critical forebrain subcortical structures include Nac and BNST[50,106–109].

Masks of all ROIs were extracted from published atlases or studies on specific regions (see Supplementary Table 4 and Fig. 1E). As done before[5], all masks were visually inspected to ensure they traced the structure of interest well. The following choices were made: the SN probabilistic mask[110] was binarized and included all voxels above a probability threshold of >0.25. The Nac probabilistic mask was binarized as well, but because of the large size of the ROI, we used a probability threshold of >0.75. We used binary masks of BNST, PAG, LC and RN[111–114] and subsampled them to 2 mm. After subsampling, only voxels exceeding >0.25 were included in the masks. ROIs of the raphe nuclei were adjusted to maximise anatomical plausibility using a threshold of >0.6 and >0.72 for dorsal and median raphe, respectively[5].

In a posteriori analyses, we also included five subdivisions of the hippocampus as additional ROIs: tail, body, head-l, head-m1, head-m2. The masks for these hippocampal subdivisions were taken from Tian & colleagues[59]).

### Stress-related behavioural markers

Our goal was to capture interindividual variation in stress levels experienced over a prolonged time period. We therefore inspected all behavioural and questionnaire markers acquired as part of the young-adult HCP dataset and included all variables that captured relevant elements of stress. A total of seven scores were included. Six of those measures were extracted from the NIH toolbox emotion battery (www.nihtoolbox.org[115,116]). Of these, two were from the stress and self-efficacy sub-toolbox: Perceived stress (PercStress) and Self-efficacy (SelfEff). Three measures captured negative affect: anger aggression (AngAggr), fear affect (FearAffect) and fear somatic (FearSomat). The sixth score was a measure of emotional support (EmotSupp) from the social relationship toolbox because a lack of emotional support can contribute to individuals' stress levels. The seventh score was the response to the following question from the NEO-Five-Factor Inventory (FFI): When I'm under a great deal of stress, sometimes I feel like I'm going to pieces (NEORAW_11[117]). Individuals were asked to choose on a five-point scale how strongly they agreed or disagreed with the statement (SA=Strongly Agree, A=Agree, N=Neither Agree, or Disagree, D=Disagree, SD=Strongly Disagree). All questions related to a prolonged time period, they asked how true statements were in general (SelfEff, AngAggr), in the past 7 days (FearAffect, FearSomat), or in the past month (EmotSupp, PercStress). Thus, our final list included seven variables (PercStress, SelfEff, AngAggr, FearAffect, FearSomat, EmotSupp, and NEORAW_11) all of which showed sufficient inter-individual variance in the 3 T cohorts D1 and R1 (see distributions in Fig. 2A and Supplementary Fig. 4), but reduced variance within the 7 T cohort, R2 (Supplementary Fig. 4A).

### Factor analysis for extracting a dimensional stress marker

Next, we conducted a factor analysis on the seven scores (z-scored) using Matlab's function 'factoran', with a 'promax' rotation (MATLAB version R2021a). The factor analysis on the D1 and R1 was virtually identical (similarity between seven factor loadings comparing D1 and R1: Pearson's $r(5) = 0.996$; $p = 7.90 \times 10^{-7}$; CI = [0.97, 1]). Thus, for consistency, both cohorts were pooled, and identical factor weights were used for all individuals throughout the manuscript (Fig. 2B). A Scree test based on all $n = 400$ 3 T participants suggested one factor (nFactors package in R (version 4.2.1, in RStudio version 2022.12.0 + 353) with function nScree[118]). We multiplied the loadings from the resulting factor onto each individual's original seven scores to construct a one-dimensional marker of stress per participant (Fig. 2C, D). We were able to closely replicate the factor analysis when including all non-overlapping $n = 1206 - 400 = 806$ HCP 3 T participants (Supplementary Fig. 4B). However, when the factor loadings were applied to the $n = 98$ 7 T dataset (R2), there was markedly reduced variability in stress

scores in this dataset which compromised our ability to examine stress in this cohort (see Supplementary Figs. 4C and 6). This may be due to a selection bias because more anxious participants may prefer not to enter a high-field scanner.

### Relating stress and hypothalamus nuclei connectivity

In our key analyses, the functional connectivity between specific hypothalamus nuclei and a priori ROIs were used to predict the dimensional stress score across subjects using a previously established procedure[5]. For each participant, we extracted the functional connectivity between each of the seven hypothalamus nuclei and each of the 15 a priori ROIs. This resulted in 105 functional connectivity values per participant. These connectivity values formed the matrix of potential predictors X that could be used to predict the dimensional stress score y. For simplicity, we refer to predictors in the X as edges or connections, although functional connectivity is a proxy for and does not perfectly capture anatomical connectivity[119].

Upon inspection of the stress scores y in our original split of the $n = 400$ 3 T participants into D1 and R1, we noticed that stress scores were not comparable (trend-wise difference in the group means: $t(394) = 0.991$; $p = 0.066$; CI = [−0.03,1]). We thus generated a new split of all $n = 400$ 3 T participants into a train and a test group ($n = 200$ each) which had comparable means and variances of stress scores (Supplementary Fig. 4C). This split was generated once and solely based on the stress marker y.

To improve the reliability of hypothalamus functional connectivity estimates in X, we rejected outlier participants if more than 10% of their functional connectivity values across all edges deviated more than 3.5 standard deviations from the mean across participants[5]. Using this procedure, two outliers were identified in the 3 T train cohort, leaving $n = 198$ and $n = 200$ participants in the train and test groups, respectively, and one outlier was excluded from the 7 T R2 cohort, leaving $n = 97$ participants in R2. Thus, from this point onwards, all key analyses looking at interindividual variation were performed on $n = 398$ 3 T participants (Figs. 3–5) and supplementary analyses were performed on $n = 97$ 7 T participants (Supplementary Fig. 6 and Fig. 2B).

Next, confound variables (whole brain volume, motion, and intracranial volume) were regressed out of the connectivity matrix X as described in refs. 5, 120. Additional confounds included the sex, age in years, body-mass index (BMI), race, ethnicity, and years of education completed (see Supplementary Table 1). A total of 11 confounds were thus regressed out of X. This was done to ensure that our results would not be driven by remaining differences between participants (e.g., differences in the quality of motion correction). Nevertheless, we confirmed that key results held without inclusion of these confounds, especially given BMI's potential relationship with stress. In general, across analyses, z-scoring was applied to regressors in the functional connectivity matrix X of the training group before running regression analyses. To achieve unbiased predictions in the test group, the z-scoring from the training group was applied to the test group using the mean and standard deviation from the training group.

In the first step, to establish whether hypothalamus connectivity captured meaningful variance related to stress, we estimated robust linear regression weights (function robustfit in MATLAB), separately for all functional connectivity edges in X (105) to capture their relationship with the stress scores in y. This first step was performed to train robust regression weights for individual edges, but not used to evaluate predictions. Following a procedure established previously[5], we then tested whether these regression weights were (1) similar between 3 T datasets ('across-subject'; Fig. 3A, B, left) and (2) similar between halves of fMRI data of each participant ('within-subject'; Fig. 3A, B, right). In other words, we fit robust regressions for each edge separately for train and test participants (Fig. 3A, B, left), or separately for functional connectivity estimates extracted from just the first or

second half of resting-state data (run 1 + 3 corresponding to the first half of each session, versus run 2 + 4 corresponding to the second half of each session, using test+train; Fig. 3A, B, right). The resulting regression weights are shown in Fig. 3B. Next, we computed Pearson's correlation coefficient between the overall pattern of regression weights, i.e., establishing whether across or within subjects, when estimated on separate parts of the data, these weights were robust and replicable and thus likely meaningful. We evaluated whether these patterns were more similar than expected by chance with a permutation null distribution generated by shuffling the stress vector y $n = 10,000$ times (while keeping the functional connectivity values unchanged) and recomputing the correlation coefficient between the overall pattern of regression coefficients for each shuffle (Fig. 3A). Finally, to visualise the contribution of each edge, we computed rDiff, the difference in the correlation coefficient between the patterns of robust regression coefficients obtained when a single edge was left out of X, compared to when all edges were included in X (which was repeated once for each of the 105 edges; Fig. 3B, bottom row).

### Out-of-sample stress prediction using hypothalamus coupling
Having tested the robustness of hypothalamus connectivity relationships with stress, we moved on to our key analysis to establish if hypothalamus nuclei connectivity was sufficient to predict stress in an independent cohort. To test this, we applied the robust regression coefficients estimated independently for all 105 edges in the training group to the functional connectivity estimates from the test group, using a weighted sum of all edge's weights and all edges' individual connectivity values, to derive individual predicted stress scores for all participants. We used Pearson's correlation to establish the similarity between predicted and true stress scores in the test group (Fig. 3C, left). For consistency and replicability, the equivalent test was repeated using robust regression weights trained on run 1 + 3 and applied to the connectivity estimates from run 2 + 4, across all participants (test +train; Fig. 3C, right).

Throughout the manuscript, analyses probing whether functional connectivity could predict stress examined whether there was a *positive* relationship between two patterns of robust regression coefficients (Fig. 3A, B) or between predicted and true behavioural scores (Figs. 3–5). *p*-values for these analyses were accordingly one-tailed.

### Characterising hypothalamus networks predictive of stress
So far, the generated out-of-sample predictions used all 105 edges. To examine whether predictions of stress levels in an independent cohort were possible with fewer edges, we repeated the same procedure by iteratively including an increasing number of edges from 1 to 105 (Fig. 4A). We sorted edges in the order of their absolute robust regression weights based on the training group and then applied between 1 and 105 train weights to the connectivity of the test group to generate predicted stress scores for the test group. Figure 4A shows the goodness of fit, i.e., the correlation between predicted and true stress scores in the test group going from 1 to 105 edges. For visualisation, lines are shown to indicate trend-wise and $p < 0.05$ significance levels, which correspond to Pearson's *r*-values of $r = 0.091$ and $r = 0.117$ (given $n = 200$, one-sided). The first 30 edges that were included are spelt out in the Supplementary Methods. To avoid conducting many tests, we followed our previously established procedure[5]: the obtained correlation coefficients were used to perform only two tests to identify (a) the smallest number of edges (smallest *n*-value) that led to a significant out-of-sample prediction (i.e., where $r > 0.117$); and (b) the best possible out-of-sample prediction (maximum *r*-value). To establish significance, we generated two permutation null distributions. We repeated the above procedure of including 1 to 105 edges $n = 10,000$ times but based on shuffled behavioural scores. In each iteration, we determined the size n of the

smallest network that reached significance (if no significant predictions were achieved for a given iteration, we used a conservative score of $n = 106$, i.e., the maximum number of edges plus 1) and the maximum Pearson's *r* achieved. One *p*-value was then extracted from each of these two null distributions.

To illustrate the quality of the prediction and the contributing edges, scatterplots and fingerprints were generated for the smallest network that reached significance and for the global peak (Fig. 4B–D and Supplementary Fig. 5B) and fingerprints were included for a few arbitrarily chosen intermediate steps for visualisation of the important edges (Fig. 4D). Fingerprints show the size of the train regression coefficient for each edge as the width of the line, its sign for predicting stress as the line style (continuous = positive; dashed=negative) and the colour reflects the hypothalamus nucleus. Supplementary Fig. 5A in addition shows whether functional connectivity was on average positive, negative, or close to zero for each edge.

Equivalent analyses using all edges or increasing the number of edges (Figs. 3 and 4) were performed with $n = 398$ 3 T participants as the training group and $n = 97$ 7 T participants as the test group and are reported in Supplementary Fig. 6.

### Nuclei-specific versus whole hypothalamus predictions
To examine whether parcellating the hypothalamus (and amygdala) led to improved prediction accuracies, we repeated the regression procedure, once with edges from the 15 ROIs to the *entire* hypothalamus instead of its individual nuclei ($n = 15$ edges instead of $n = 7 \times 15 = 105$ edges; Fig. 5A–D, left), and once with edges from the same ROIs but with the amygdala included as a whole instead of its individual nuclei (15-6 = 9 ROIs) both for the whole and the parcellated hypothalamus ($n = 9$ edges or $n = 7 \times 9 = 63$ edges, respectively; Fig. 5A–D, right). In other words, compared to previous analyses, no change was done to any ROIs except the hypothalamus and/or amygdala. Relationships with stress were established between the hypothalamus and the same set of ROIs as before (e.g., NAc, LC, etc), just changing the spatial precision with which the hypothalamus (and amygdala) were represented. As before, robust regression coefficients obtained from the training group were applied to the test group's functional connectivity to predict the test group's stress scores. We note that typically, a larger number of predictors should perform better. However, here, the edges captured the same information in both cases, namely functional connectivity with all hypothalamus and all amygdala voxels. It is conceivable that (a) BOLD measurements in smaller sets of voxels are noisier and as a result, predictions generated from the functional connectivity of hypothalamic nuclei, as opposed to the hypothalamus as a whole, might be worse, or that (b) hypothalamus subdivisions into nuclei are meaningful and increase the prediction accuracy for stress scores. Nevertheless, to make predictions from whole and nuclei-specific hypothalamus functional connectivity more comparable, we generated null distributions for all four cases (amygdala whole/nuclei x hypothalamus whole/nuclei). We shuffled the order of participants' stress scores in both test and train datasets and repeated the procedure of predicting test stress scores from train weights 10,000 times. The *p*-values extracted from the respective null distributions account for the number of predictors, allowing a direct comparison (Fig. 5B). Thus, this test established if parcellating the hypothalamus and amygdala into nuclei helped us improve the accuracy of our predictions.

### Behavioural specificity of predictions for stress
A second control analysis was performed to assess the behavioural specificity of our predictions for stress as opposed to other subclinical mental health dimensions. Building on our prior work[5], we compared the predictions of individuals' stress scores with predictions achieved for four related mental health scores capturing 'social and life satisfaction', 'negative emotions', 'sleep problems', and 'anger and

rejection'. These were extracted using factor analysis from a larger set of 33 questionnaire scores as described in ref. 5. The four alternative dimensional scores were not fully orthogonal to stress (Fig. 6D). Nevertheless, we hypothesised that functional connectivity in hypothalamus nuclei networks may better predict stress scores compared to related mental health scores. Predictions for these four alternative scores were generated using all $n = 105$ edges (Fig. 6A) or increasing number of edges (Fig. 6C) using identical procedures as before.

## Reporting summary

Further information on research design is available in the Nature Portfolio Reporting Summary linked to this article.

## Data availability

The neuroimaging data used in this study are available from the Human Connectome Project (www.humanconnectome.org). Users must apply for access and agree to the HCP data use terms (for details see https://www.humanconnectome.org/study/hcp-young-adult/data-use-terms). Here, we used both Open Access and Restricted data. The masks of all ROIs used in this study as well as all individual hypothalamus nuclei generated here and further Source data to figures are provided in the Source Data file and are available in the OSF repository http://osf.io/bq3fd. Source data are provided in this paper.

## Code availability

Code that allows HCP users to replicate analyses, including plotting all figures presented in the manuscript is provided in the OSF repository http://osf.io/bq3fd. Intermediate analysis outputs can be made available to registered HCP users. Please see the README file in the Scripts folder for further details.

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

## Acknowledgements

This research was funded in part by Sir Henry Wellcome and Henry Dale Fellowships awarded to MKF (103184/Z/13/Z and 223263/Z/21/Z), an MRC grant and the Wellcome Senior Investigator Award awarded to MFSR (MR/P024955/1 and 221794/Z/20/Z), and a grant from the HDH Wills 1965 Charitable Trust (REGISTERED CHARITY NO. 1117747, PI: KPE). This research was funded in whole, or in part, by the Wellcome Trust [203139/Z/16/Z and 203139/A/16/Z]. For the purpose of Open Access, the author has applied a CC BY public copyright licence to any Author Accepted Manuscript (AAM) version arising from this submission.

## Author contributions

D.E.A.J. and M.C.K.F. designed the study. D.E.A.J., M.F.S.R. and M.C.K.F. conceived analyses, inspired by our prior work using similar methods to investigate amygdala circuits. D.E.A.J. and M.C.K.F. wrote analysis code. M.F.S.R. gave analysis advice, and all authors (D.E.A.J., M.C.K.F., M.F.S.R., S.S., K.P.E.) edited the manuscript.

## Competing interests

The authors declare no competing interests.
