## [Peer Review File · Nature Communications]

Nuclei-specific hypothalamus networks predict a dimensional marker of stress in humansReviewer #1 (Remarks to the Author):

REVIEW on NCOMMS-23-23766-T: Nuclei-specific hypothalamus networks predict a dimensional marker of stress in humans

SUMMARY: The authors use three high quality resting state fMRI samples from population cohorts to build functional connectivity (FC) based parcellation of the hypothalamus after state-of-the-art preprocessing. They secondly use these subregions along with a selection of additional subcortical regions to define a set of 15 regions with 105 edges by which they aim to predict an aggregated subjective stress sum score in two independent cohorts. In addition the authors demonstrate that the prediction quality is higher with hypothalamic and amygdala subnucleus information compared with the information from undivided regions.

GENERAL COMMENT: Conceptually this report combines (i) the presentation of a hypothalamus parcellation based on functional connectivity (of resting state fMRI population cohort samples) with (ii) a first subclinical validation of the resulting parcellation based on individual stress scores. There is undoubtedly considerable methodological rigor visible but there exist potentially correctable inconsistencies in the approach (for example lack of hippocampal information for the prediction network), and the translational propulsion seems to strong given that the amount of predicted behavioural variance is very low when analyzed conservatively. Overall, I think the undoubted major value of this brain mapping work lies in the delivery of a data based hypothalamus parcellation.

INTRODUCTION

Q1. The authors should elaborate on why the hippocampus and its subregions were not included in an analysis on stress response regulation. This fact seems skipped and remains undiscussed. There is also multiple sources of subregion definitions of the hippocampus out, including one based on 2x2x2 mm³ rsfMRI data (Tian et al.).

Q2. The stress response regulation system in humans should be introduced more broadly comprehensively and not with too much of a keyhole perspective built on the hypothalamus (e. g. work by Hermans et al.).

Q3. A definition of stress and stress regulation in a neurobiological sense should be pointed out.

METHODS

Q4. Atlas based region selection: More recently, for subcortical regions, an FC based atlas has been developed, matching the here presented FC concept. Why were the regions chosen from heterogeneous atlases with complex thresholding techniques? The same question accounts for the hippocampus that is clearly missing.

Q5. HTH parcellation based on cortical connectivity: The >91000 voxels contained all cortical and subcortical voxels EXCEPT the hypothalamus itself, is this correct? Figure 1A (right side) seems to show also the hypothalamus itself along the third ventricle. Were the hypothalamic voxels themselves also contained? The contours of the HTH should be marked in the figure to lay open which voxels are auto-correlational.

Q6. The 91.282 data points: How many were surface vertices and how many were subcortical voxels?

Q7. The 33 physiological regressors of the PNM toolbox, were these whole brain/intracranial volume based, or customised to the ROI selection in some way?

Q8. "Variance of the noise normalised" (line 491ff) step should be explained in one more sentence: The data have been denoised by multiple regression before, so what is considered "noise" after that? Do the authors want to express that the variance of the residuals (the remaining signal after regression) was normalized? If so, the term "residuals" seems better suited.

Q9. The 5 mm smoothing refer to the surface, correct?

Q10. The authors mention "three connectomes" - they still refer to D1, R1 and R2? Maybe the

syntax "D1+R1" is misleading and should be changed.

Q11. Major: "group average time course": This needs a clearer presentation: rsfMRI data are not time locked and cannot be easily averaged across subjects. The 2nd step ('cifti-correlation') should be revised in that the maths behind the step and the tool are kept separate. No technical term "cifti-correlation" is actually needed if the procedure is simply a Pearson correlation followed by a transformation to a Fisher's z score.

Q12. By "dense connectome" the authors refer to a voxel-by-voxel connector, correct? If so, also here the phrasing should be revised and simply described that all surface/voxel time series (in contrary to regional time courses) were used.

Q13. Figure 1A: Could the multi-colour bar possibly be changed to a hot/cool color principle? It is counterintuitive to color very negative values light green after dark blue.

Q14. Decision to use absolute FC values and not preserve the signature: This needs some explanation as it is a bit counter-intuitive. Usually one would consider a complete flip of the sign (say 0.5 and -0.5) an indication for a different FC, and also the whole brain hypothalamus FC maps show that anterior and posterior parts of the brain are differently connected, even to the total HTH. I understand that global signal issues shift the total FC histogram and that the zero position is thus dependent on the preprocessing, but is it really correct to bundle together voxels with a flipped signature of their FC to a target? Did the more recently published Tian atlas for subcortical nuclei parcellation pursue this strategy? What was the average percentage of negative FC values flipped?

Q15. Supplementary Table 3 is very good and informative. Could the comparison between the FC based parcellation and the competing 6 other ones be summarised in a few sentences in the result section? The 1:1 correspondence to Mai et al. is of striking, but it is difficult to follow to what degree these final steps were heuristic/human and not automated. Is there a way to quantify the overlap with these 6 systems despite them having a different N of subregions? Was there a level in the hierarchical tree with more than 7 regions per hemisphere that was rejected, and how was the cut-off decision made?

Q16. Regressing out confound variables from the regional FC matrix: Wasn't the preprocessing designed to control for these influence? Is it a good idea to correct a second time? I am not convinced that clinical variables such as BMI and education need to be corrected for. BMI, btw, can be increased due to chronic stress, so there is a risk to loose true effects. Were at least some of the analyses performed w/o the second nuisance variable correction?

Q17. Major: "Robust regression coefficient": Application of regression coefficients between FC and stress cores from training to test group: It is not entirely clear, what kind of basic regression analysis was actually performed to connect the 105 edges with the single stress score: (a) Massive univariate (=separate) or (b) some truly multivariate analysis or (c) a multiple linear regression with all FC values predicting one stress value? It seems that an SVM or some other type of supervised machine learning would also be a good candidate. The authors should explain this more comprehensively.

Q18. In the chapter "Predictions using nuclei versus whole HTH connectivity", how were the other subcortical nuclei dealt with (for example, the NAcc etc.) - were they kept in?

Q19. Specificity of the FC matrix predicting the stress scores: I would like to suggest a different null model as FC is so powerful that often very indirectly, 'networks' predict a behavioural phenotype. Could the authors define 15 cortical regions (e. g. cingulate cortex or prefrontal, each of them with the same ROI size) and estimate the correlational predictive power in the exactly same way? I am not fully convinced of the claimed hypothalamic specificity.

Q20. In the same direction, in the plot of the rising number of edges used, would the proportion of edges occupied by the HTH be marked in color within each bar?

RESULTS

Q21. The paragraphs of the result section should more directly match the paragraphs of the methods section.

Q22. There are several doublings between the methods and the result sections: One of many examples is the topic of the new split of the test and train samples due to different stress scores.

Q23. The used softwares and tools could be aggregated into one separate methods section.

Q24. It is surprising that the seven moderately correlated self reported stress items result in only one common factor. What were the threshold criteria for this solution? Is it still only one resulting factor when D1 and R1 are pooled?

Q25. The authors report that the application of the factor loadings to the 7 T sample resulted in a lower range of the (aggregated) stress score. Was this also reflected in lower stress scores at the level of the original 7 questionnaires? If so, the translation to the 7 T still seems valid, as lower stress score range is (correctly) found. Please discuss.

DISCUSSION

Q26. The cohorts were not 'transdiagnostic' in the clinical sense. If diagnoses were made, they should be reported. It seems that stress levels in the population with not a lot of consideration of clinical significance are reported. This is an important point and no claims for clinical translation should be made at this level. Correlations can be totally different in disease as here compensatory processes play in.

Q27. Competing methods such as functional gradients to parcel out regions should be discussed (Tian paper). In turn, the principle driving the parcellation here (similarity of regional connectivity of a HTH voxel).

Q28. As mentioned before, the use of the absolute FC values that mirror anti-correlation to the positive side should be re-visited in the discussion.

Q29. As also mentioned before, not including hippocampal information in the prediction network will be difficult to accept for many stress researchers. It might be discussed that at least the connectivity between the HTH and the hippocampus was considered for the parcellation.

Q30. Literature connecting the resting state to explicit HPA axis markers could be cited.

GENERAL/STYLE

Q31. Computer syntax style in reporting regions (e. g. "RN_DR" with an underscore) should be rather avoided in the main text.

Q32. p-values and r-values should be reported with uniform precision in terms of the number of digits after the comma.

Reviewer #2 (Remarks to the Author):

I think this is a very interesting study of the hypothalamic nuclei network. The hypothalamus network related to stress, in particular, is quite new and exciting, and the analysis procedures seem sound overall. I would like to raise some points of details regarding interpretation of the results, which I hope may improve this manuscript.

I think the parcellation results need further consideration. The supraoptic, suprachiasmatic and medial preoptic nuclei are located in the anterior part of the hypothalamus, but these nuclei were included in the "PHs (posterior superior hypothalamic nucleus)" (Table S3). It seems to me that these nuclei are included in the AH or VMA of this study. It also seems possible that the PHs of this study corresponds to the posterior hypothalamic nucleus and that the PH of this study corresponds to the mammillary body. It would be useful to classify the nuclei more quantitatively by referring to the coordinates of the nuclei reported in previous studies, and please make sure entire parcellation results appear more interpretable.

Please correct the Z coordinates in Fig. 1B.

It would be helpful to provide a table listing the 22 important connections to clarify specific networks of interest.

Reviewer #3 (Remarks to the Author):

Review of Jensen et al: Nuclei-specific hypothalamus networks predict a dimensional marker of 2 stress in humans

The study focuses on the role of the hypothalamus in stress responses. The hypothalamus is a complex structure composed of multiple nuclei. The study used resting-state fMRI on a large sample of healthy young adults, the Human Connectome Project (n=498) to examine the functional connectivity of specific hypothalamic nuclei and its relationship with prolonged stress.

The key findings are as follows: (1) the investigators were able to parcellate the human hypothalamus into seven nuclei in vivo; (2) the functional connectivity between these nuclei and other subcortical structures, including the amygdala, was used to predict stress scores out-of-sample; (3) the authors suggest that stress is related to connectivity changes in precise and functionally meaningful subcortical networks.

The strengths of the manuscripts are:

- (1) The use of the HCP large sample size which speaks to the statistical power and the generalizability of the results.
- (2) The investigators used high-quality fMRI resting-state data and parcellated the human hypothalamus into seven anatomically plausible nuclei, which is a level of detail allows for a more nuanced understanding of the brain's structure and function.
- (3) The authors replicated the average hypothalamus connectivity and its parcellation in two independent datasets, which strengthens the validity of the findings.
- (4) The study found that stress predictions were better when considering nuclei as opposed to whole hypothalamus connectivity, and they were behaviorally specific.

There are several weaknesses that should be considered:

- (1) This manuscript is based on cross-sectional HCP data, which limits the ability to draw conclusions about causality or the direction of relationships between variables.
- (2) The authors acknowledged that at a higher spatial resolution, it might be possible to identify lateral and medial subdivisions of the hypothalamus, which was not possible in this study.
- (3) The finding is based on resting state fMRI, which provides some idea about how brain structures are connected at rest but not necessarily how they interact to respond to an external or internal challenge (stress). Therefore, the findings need to be considered with caution when interpreting the connectivity predictions.
- (4) The HCP data are based on healthy volunteers, thus there is relatively little insights that can be gained into process abnormalities that may occur in the context of stress disorders (such as PTSD).
- (5) The author imply that these findings may have some implication for interventions but since (a) this is based on healthy volunteers and (b) there was no intervention involved, this is highly speculative. Future research is needed to explore whether inducing changes in precise

hypothalamus networks may alter an individual's experience of stress.

Taken together, this is an interesting manuscript that provides some conceptual advance in understanding the role of the hypothalamus, but has limitations in terms of the causal conclusions and the applicability to individuals with stress disorders.

REVIEWER COMMENTS

Reviewer #1 (Remarks to the Author):

REVIEW on NCOMMS-23-23766-T: Nuclei-specific hypothalamus networks predict a dimensional marker of stress in humans

SUMMARY: The authors use three high quality resting state fMRI samples from population cohorts to build functional connectivity (FC) based parcellation of the hypothalamus after state-of-the-art preprocessing. They secondly use these subregions along with a selection of additional subcortical regions to define a set of 15 regions with 105 edges by which they aim to predict an aggregated subjective stress sum score in two independent cohorts. In addition the authors demonstrate that the prediction quality is higher with hypothalamic and amygdala subnucleus information compared with the information from undivided regions.

GENERAL COMMENT: Conceptually this report combines (i) the presentation of a hypothalamus parcellation based on functional connectivity (of resting state fMRI population cohort samples) with (ii) a first subclinical validation of the resulting parcellation based on individual stress scores. There is undoubtedly considerable methodological rigor visible but there exist potentially correctable inconsistencies in the approach (for example lack of hippocampal information for the prediction network), and the translational propulsion seems to strong given that the amount of predicted behavioural variance is very low when analyzed conservatively. Overall, I think the undoubted major value of this brain mapping work lies in the delivery of a data based hypothalamus parcellation.

We would like to thank the reviewer for their positive feedback and thoughtful evaluation of our work. We found their comments to be extremely helpful and have responded to them one by one below. Thank you very much for taking the time to give us such a detailed evaluation of our work.

INTRODUCTION

Q1. The authors should elaborate on why the hippocampus and its subregions were not included in an analysis on stress response regulation. This fact seems skipped and remains undiscussed. There is also multiple sources of subregion definitions of the hippocampus out, including one based on 2x2x2 mm³ rsfMRI data (Tian et al.).

Thank you for this comment. We indeed considered including the hippocampus several times at different stages of the project. We are very aware of its importance in the context of stress. At the same time, we felt that it would not be helpful to include the hippocampus as a single structure (treating it as one homogenous region). We also thought it would take quite some time to parcellate the hippocampus carefully in each person and we wanted to keep the focus of our manuscript on the hypothalamus which has received a lot of interest in the context of stress rodents, but comparatively little in human work.

However, the reviewer is of course right in pointing out that a very good parcellation at the same resolution has since been published by Tian and co-authors. We have now followed the reviewer's advice and reran our analyses using an additional five subdivisions of the hippocampus as shown in part of Tian et al.'s Figure 4 pasted below: tail, body, head-I, head-m1, head-m2

We found that the inclusion of these additional 5 x 7 edges (5 hippocampus ROIs x 7 hypothalamus nuclei) did not improve the overall prediction of stress scores when using all predictors ($r=0.157$ instead of $r=0.265$). Please see both scatterplots pasted for comparison below.

105 edges (15 ROIs x 7 hypothalamus nuclei)	140 edges (20 ROIs including 5 hippocampus sub-regions x 7 hypothalamus nuclei)
$r=0.265$; $p=0.0007$ Manuscript Figure 3C	$r=0.157$; $p=0.131$ Supplementary Material Figure S8C
Strongest 20 predictors: SO/SC-RN-MR, MPO-amyg-Ce, PV-LC, PV-NAc, SO/SC-amyg-Lal, MPO-amyg-B, MPO-BNST, VM-LC, PV-amyg-Ce, SO/SC-amyg-LaD, PH-dPAG, DM-LC, PV-BNST, SO/SC-amyg-CoN, SO/SC-amyg-AB, SO/SC-SN, DM-amyg-LaV, DM-amyg-B, SO/SC-dPAG, VM-RN-MR, MM-amyg-B (see Supplement Result, p.3).	Strongest 20 predictors: SO/SC-RN-MR, MPO-amyg-Ce, PV-LC, PV-NAc, VM-HCthead, SO/SC-amyg-Lal, MPO-amyg-B, MPO-BNST, VM-LC, PV-amyg-Ce, PV-HCm1head, SO/SC-amyg-LaD, PH-dPAG, DM-LC, PV-BNST, SO/SC-amyg-CoN, SO/SC-amyg-AB, SO/SC-SN, SO/SC-HCtail, DM-amyg-LaV.

Even when edges were included iteratively, adding the hippocampus connectivity with the hypothalamus only added noise to the prediction and the peak was not superior (updated peak prediction at 18 connections including hippocampus-to-hypothalamus: $r=0.222$ vs original peak prediction at 22 connections without hippocampus included: $r=0.272$ – compare the top plot below with Fig. 4 in the manuscript, pasted underneath the new figure for comparison below).

Here, among the strongest 20 predictors identified in the training set (see table above), we only identified three edges between hypothalamus and hippocampus (VM to HChead (5th), PV to HCm1head (11th), and VMa to HCtail (19th)).

Supplementary Material Figure S8D: Anatomical features of hypothalamus networks predictive of stress. Prediction of stress scores obtained using subsets containing between 1 and 140 edges for all hypothalamus nuclei to all 20 ROIs including five hippocampus subregions. Edges were included in order of importance (absolute robust regression coefficients) estimated from the train group ($n=198$). Prediction accuracies (turquoise bars) are shown as the correlation between true and predicted stress scores in the test participants ($n=200$), but were only statistically evaluated at the peak (red arrow: 'global peak') and to derive the smallest number of edges that reached a significant out-of-sample prediction (black arrow, 'smallest significant network', here a single connection); black curve indicates performance using a given number of edges (on x) but included in random order ($n = 10,000$ shuffles; error bars denote s.e.m.); black line at $r = .1167$ indicates threshold for significance at $P < .05$ purely for visualisation (grey line: $P < 0.1$, $r = .0911$). A subset of all 140 hypothalamus-to-ROI edges was sufficient to achieve a significant out-of-sample prediction. A significant prediction could be achieved using only the first edge (SO/SC to RNMR; $r = .131$). The best prediction was achieved using 18 edges ($r = .222$).

Manuscript Figure 4A: Anatomical features of hypothalamus networks predictive of stress. Prediction of stress scores obtained using subsets containing between 1 and 105 edges. Edges were included in order of importance (absolute robust regression coefficients) estimated from the train group ($n=198$). As in Fig 3, weights were applied to test participants' functional connectivity values to predict stress scores, but this time with increasing numbers of edges. Prediction accuracies (coloured bars) are shown as the correlation between true and predicted stress scores in the test participants ($n=200$), but were only statistically evaluated at

the peak (red arrow: 'global peak') and to derive the smallest number of edges that reached a significant out-of-sample prediction (black arrow, 'smallest significant network'; see Methods for generation of null distributions using permutation; for precise P values in each case, see Results); black curve indicates performance using a given number of edges (on x) but included in random order (n = 10,000 shuffles; error bars denote s.e.m.); black line at $r = .1167$ indicates threshold for significance at $P < .05$ purely for visualisation (grey line: $P < 0.1$, $r = .0911$). A subset of all 105 hypothalamus-to-ROI edges was sufficient to achieve a significant out-of-sample prediction. A significant prediction could be achieved using only the first edge (SO/SC to RNMR). The best prediction was achieved using 22 edges ($r = .272$).

Thus, to keep the number of predictors low and the focus of our main manuscript on the hypothalamus, we have decided to include this new analysis in the Supplementary Material p. 2 and 4 (new Supplementary Fig S8, pasted below).

We now refer to it in the Results & Methods as follows (please note that page numbers throughout relate to the 'tracked changes' version of the manuscript):

Methods (p.24):

In a posteriori analyses, we also included five subdivisions of the hippocampus as additional ROIs: tail, body, head-l, head-m1, head-m2. The masks for these hippocampal subdivisions were taken from Tian & colleagues (Tian et al., 2020).

Results (p.11):

Similarly, for predictions including hippocampus subdivisions as additional ROIs, see Supplementary Results and Supplementary Figure S8.

Supplementary Methods (p. 2):

Inclusion of additional hippocampus ROIs

Our a priori ROIs did not include the hippocampus. In an additional analysis, based on a reviewer suggestion, we included five subdivisions of the hippocampus as five additional ROIs. These were based on a parcellation by Tian et al. (Tian et al., 2020) and included: tail, body, head-l, head-m1, and head-m2. We computed the functional connectivity of these hippocampal subdivisions with each of the seven hypothalamus nuclei, leading to $5 \times 7 = 35$ additional edges. We then repeated the same analyses as before: (1) prediction of stress scores in the test group using all edges with weights estimated from the training group; (2) prediction of stress scores in the test group using iteratively more edges, sorted based on their importance in the training group, to find the peak and smallest significant network.

Supplementary Results (p. 4):

Finally, we repeated our key analyses, corresponding to Figures 3C and 4A using 35 additional edges between five hippocampal subdivisions and our seven hypothalamus nuclei. We found that the inclusion of these additional edges did not improve the overall prediction of stress scores when using all predictors ($r = 0.157$ instead of $r = 0.265$). Similarly, when edges were included iteratively, adding the hippocampus to the set of ROIs only added noise to the predictions: the

peak was not superior (updated peak prediction at 18 connections including hippocampus-to-hypothalamus: $r=0.222$ versus original peak prediction at 22 connections without hippocampus included: $r=0.272$). Here, among the strongest 20 predictors identified in the training set, we only identified three edges between hypothalamus and hippocampus (VM to HC1head (5th), PV to HCm1head (11th), and VMa to HCtail (19th)). **Supplementary Figure S8** shows the results of these additional analyses.

We note that these new analyses do not question the importance of the hippocampus itself in the context of stress. They just show that it is not a specific hippocampus-subdivision to hypothalamus-nucleus functional connectivity that carries predictive variance for stress. It may well be that coupling between the hippocampus and regions other than the hypothalamus are important. We have therefore also added a few sentences to the Discussion to further emphasize the importance of the hippocampus.

Discussion (p.16):

The hippocampus shows sign of degeneration following stress (Levone et al., 2014; McEwen et al., 2016) and is therefore an important subcortical region not included *a priori* here. We did not want to treat the hippocampus as a homogenous region; however, since starting this work, Tian and colleagues (Tian et al., 2020) published a detailed hippocampus parcellation, which we included *a posteriori* into our analyses for comparison (**Supplementary Information** and **Supplementary Figure S8**). Surprisingly, inclusion of the hippocampus subregions as additional ROIs did not improve stress predictions. While these new analyses do not question the importance of the hippocampus in stress predictions *per se*, they do suggest that functional connectivity between specific hippocampus-subdivisions to hypothalamus-nuclei does not carry additional predictive variance for stress. Coupling between the hippocampus and regions other than the hypothalamus should be considered in future work.

We have now included this new supplementary figure in the manuscript (Supplementary Fig S8):

Predicting stress using hypothalamus nuclei coupling with hippocampal sub-groups

A (1) Across-subject (3T to 3T)

(2) Across datasets 3Tvs. 7T

B Similarity of regression weights

C Out-of-sample prediction

D Size and nature of hypothalamus nuclei networks predictive of stress scores (3T to 3T)

Supplementary Figure S8: Predicting stress scores using hypothalamus nuclei coupling with five additional hippocampal subregions. We replicated stress predictions using hypothalamus nuclei connectivity with our a priori ROIs and an additional five hippocampal subregions from Tian et al. (2020). Thus, instead of the original 105 edges (7 hypothalamus nuclei x 15 ROIs), we now included 140 edges between 7 hypothalamus nuclei and 20 ROIs.

Q2. The stress response regulation system in humans should be introduced more broadly comprehensively and not with too much of a keyhole perspective built on the hypothalamus (e. g. work by Hermans et al.).

Thank you for this comment. It was partly because of the short format of this journal that we kept the introduction quite focussed on the hypothalamus. If the editors are happy for us to expand our introduction even more, we would be happy to do this. For now, we have included one paragraph where we sign-post some of the relevant literature in brief. We are happy to amend this if you have any further suggestions.

Introduction (p.2):

Prolonged periods of stress are associated with several common psychiatric disorders including depression and anxiety (Agid et al., 2000; Schmidt et al., 2008). Yet, the brain changes that accompany stress are insufficiently understood. Stress can be defined as an “actual or anticipated disruption of homeostasis or an anticipated threat to well-being” (Ulrich-Lai & Herman, 2009). Physiological and behavioural responses to acute stress are adaptive, spatially and temporally specific, and regulated by a distributed network of regions including prefrontal cortex, amygdala, and hippocampus (Hermans et al., 2011, 2014; McEwen et al., 2015; McEwen & Gianaros, 2010). When these circuits trigger autonomic and endocrine stress responses, brainstem centres as well as the hypothalamic-pituitary-adrenal (HPA) are activated (McEwen et al., 2015; Myers et al., 2012; Ulrich-Lai & Herman, 2009). Despite the various contributions made by this diverse set of regions, in this study, we focus on the hypothalamus. As part of the HPA axis, the hypothalamus is one of the key brain regions mediating the balance of hormones in response to acute stress (Bao et al., 2008; Herman et al., 2008; Zhang et al., 2021). For instance, the hypothalamus can directly activate the anterior pituitary and, via the adrenal cortex, trigger the release of cortisol. However, when stress turns into a prolonged state, it becomes maladaptive (Arnsten, 2009; de Kloet et al., 2005; Roozendaal et al., 2009; Schwabe et al., 2022) and is frequently associated with compromised well-being. Currently, it remains unclear how human hypothalamus networks are affected by stress, in particular prolonged periods of stress. This might partly be because it is challenging to study the hypothalamus *in vivo* in humans.

Q3. A definition of stress and stress regulation in a neurobiological sense should be pointed out.

We have now included a definition of stress and stress regulation (pasted in response to Q2 above) and we hope that this is what the reviewer had in mind. There are multiple possible definitions of stress, and we are trying to make clear further down in the paragraph that we are particularly interested in the effects of prolonged stress in this study.

METHODS

Q4. Atlas based region selection: More recently, for subcortical regions, an FC based atlas has been developed, matching the here presented FC concept. Why were the regions chosen from heterogeneous atlases with complex thresholding techniques? The same question accounts for the hippocampus that is clearly missing.

We are not sure which atlas the reviewer is referring to. To the best of our knowledge, there are currently no FC-based atlases for small brainstem nuclei (e.g., for the LC or Raphe). If the reviewer is referring to the work by Tian et al., we agree that it is fantastic, but it does not include all the regions of interest for our study, in particular, it does not include any in the brainstem, and only two in the amygdala. Our ROIs were chosen *a priori* based on their strong coupling with the hypothalamus, their association with neurotransmitter systems, and their importance for stress-related responses and mental well-being. We then used the best mask we could find for each ROI.

Please note that the same masks were used in our previous work on the amygdala (Klein-Flügge et al., Nat Hum Beh, 2022) and that all masks have been published here (<https://osf.io/egm2r/>). The thresholds used are not arbitrary but were carefully selected to match the size and shape of the relevant nucleus/area based on anatomical knowledge and atlases. Importantly, this was done prior to and independent of the relevant analyses related to stress.

In case helpful, we have pasted the more detailed description of this procedure from our Nat Hum Beh paper referenced above below for the reviewer to inspect. We would be happy to include this additional detail in the Supplementary Methods if the reviewer found this helpful.

- We included the substantia nigra (SN), which contains many dopaminergic neurons and the nucleus accumbens (NAc), an area receiving strong dopaminergic innervation (Averbeck & Costa, 2017). DA has been implicated in mental health disorders; for example, Parkinson's disease, which is characterized by a loss of DA neurons in SN, leads to depression in a large percentage of patients (~35%; (Aarsland et al., 2011). But DA also plays a key role in reward-learning and sleep regulation. Striatal dysfunction has, for instance, been associated with sleep disturbances and a subset of NAc core neurons was found to regulate slow-wave sleep (Brown et al., 2012; Oishi et al., 2017). The SN mask was taken from the NITRC Atlas of the basal ganglia (Keuken et al., 2014) and contained 134 voxels. The NAc was taken from the Harvard Subcortical Atlas and contained 188 voxels.
- The bed nucleus of the stria terminalis (BNST) was included because of its role in mediating the long-term effects of anxiety and responses to stress (Lebow & Chen, 2016). It is also sometimes considered part of the extended amygdala. The BNST mask was obtained from (Folloni et al., 2019) and contained 45 voxels.
- Two regions with opposing functionality within the periaqueductal grey (PAG) were included because of their importance in regulating autonomic arousal: ventrolateral PAG (vlPAG) which mediates rest- and digest-related behaviour and dorsal PAG (dPAG) which mediates fight and flight responses. The masks for these regions were taken from (Faull et al., 2016). The dPAG was the summation of Faull et

al.'s dorsomedial (dm) and dorsolateral (dl) aspects of PAG; vIPAG contained 43 voxels and dPAG 45 voxels.

- The role of serotonin and of selective serotonin reuptake inhibitors (SSRI) in the pathology and treatment of mental health disorders is well known. The raphe nuclei are the most important source of serotonin in the brain. Masks for dorsal and median raphe nuclei were taken from the Harvard Ascending Arousal Network Atlas (Edlow et al., 2012). The dorsal raphe nucleus (RNDR) contained 23 voxels, and the median raphe nucleus (RNMR) contained 8 voxels.
- Finally, locus coeruleus (LC), the main site of noradrenaline production was defined based on (Betts et al., 2017) and contained 20 voxels. Probabilistic masks were binarized first, including all voxels with probability $>.25$, in other words, voxels that had a larger than 25% chance of being within the given region (NAc, SN).
- Binary files and all masks we received in binary format (BNST, PAG, LC, RN) were subsampled to 2mm. While most regions were then simply binarized again using any voxels $>.25$ in subsampled space, we carefully and visually inspected all subcortical ROIs individually at 2mm to try and achieve the best tracing of their shape, volume and position when compared with a higher-resolution versions from published atlases. However, this meant that we did not apply the exact same threshold for each ROI; we deviated from our standardized procedure in three cases to improve the masks: nucleus accumbens, dorsal and median raphe. In these three cases, slightly different thresholds provided more anatomically plausible ROIs. NAc thresholded at $.25$ would have yielded an unusually large ROI, so a threshold of $>.35$ was applied in the second step; for the raphe nuclei, thresholds were adjusted manually to maximise anatomical plausibility ($>.6$ and $>.69$ for dorsal and median, respectively).
- Similarly, in some cases, a few voxels were added or removed manually from the ROI to achieve better tracing of the original higher resolution mask. Importantly, all of these mask optimization steps were performed based on anatomical criteria and prior to any subsequent analyses. All ROIs are published as part of the OSF repository <https://doi.org/10.17605/OSF.IO/EGM2R>

We believe we have addressed the point about the hippocampus above in response to Q1.

Q5. HTH parcellation based on cortical connectivity: The >91000 voxels contained all cortical and subcortical voxels EXCEPT the hypothalamus itself, is this correct? Figure 1A (right side) seems to show also the hypothalamus itself along the third ventricle. Were the hypothalamic voxels themselves also contained? The contours of the HTH should be marked in the figure to lay open which voxels are auto-correlational.

For the purpose of producing the visual map in Figure 1A, the hypothalamus was included and of course, it is expected that it would show a high value due to auto-correlations. We have now highlighted the hypothalamus in the figure as suggested by the reviewer (using a transparent overlay of the hypothalamus ROI which we felt worked best), and commented on this in the legend, as follows:

Manuscript Figure 1A: Connectivity-based parcellation of the human hypothalamus. Group average hypothalamus-to-whole-brain functional connectivity extracted from resting-state fMRI data of $n=200$ 3T HCP participants. Colour scale indicates Pearson's correlation coefficient. Positive versus negative functional connectivity is shown in red versus blue colours, respectively. Hypothalamus outline is shown in semi-transparent colour; strong red values within the hypothalamus indicate strong autocorrelation of activity.

For the parcellation, similarly, hierarchical clustering was run on the connectivity of the hypothalamus to all brain vertices, including itself. In other words, this included the hypothalamus self-connectivity. This was an intentional choice as we believe the intrinsic connectivity within the hypothalamus might be informative for the parcellation. However, for multiple reasons, it is unlikely to have had a major effect on the parcellation:

(1) In one case the similarity matrix is based on the hypothalamus x all brain-ordinates (109 x 91,282 edges); in the other case, without the hypothalamus self-connectivity, it would be based on 109 x 91,173 edges. The difference is very small (109 voxels are 0.0012% of all 91,282 brain ordinates).

(2) Because hypothalamus voxels tend to be connected strongly to other hypothalamus voxels, these 109 x 109 edges do not add a lot of variance to the similarity matrix: the correlation between the similarity matrices with/without inclusion of the hypothalamus self-connectivity is $r=0.92$, so it is unlikely to have led to major differences in the parcellation.

Nevertheless, we tried parcellating the hypothalamus again, this time excluding the intrinsic hypothalamus functional connectivity. While not much changed, overall, we identify a worse separation between VMa (now called SO/SC based on R2's suggestion) in the left hemisphere in step 4. In general, more splits occurred on the right hemisphere, leading to interhemispheric asymmetry. Given that we know that the hypothalamus is broadly symmetrical, this may indicate that some useful connectivity information was lost when excluding intrinsic hypothalamus connectivity. And at later clustering steps, showing step 10 below as an example, the parcellation in the left hemisphere appears overall less anatomically plausible compared to what is known from published atlases.

Original parcellation using hypothalamus functional connectivity to all other brain ordinates	Parcellation using hypothalamus functional connectivity to all other brain ordinates except the hypothalamus itself
Step 4:  Step 10: 	Step 4:  Step 10: 
Taken together, we would prefer to include the intrinsic hypothalamus connectivity for the reasons outlined above. We have made this choice more explicit in the Methods of the manuscript (p.21):

Note that the self-connectivity of the hypothalamus was included for generating this similarity matrix because it might be informative for parcellating the hypothalamus into subdivisions.

Q6. The 91.282 data points: How many were surface vertices and how many were subcortical voxels?

Out of the 91,282 brain vertices, 31,870 were subcortical voxels including 17,853 voxels of the cerebellum (without cerebellum = 14,017) and 59,412 were cortical vertices. We have now included this information in the manuscript.

Methods (p.21):

“Out of the 91,282 brain vertices, 31,870 were subcortical voxels and 59,412 were cortical vertices.”

Q7. The 33 physiological regressors of the PNM toolbox, were these whole brain/intracranial volume based, or customised to the ROI selection in some way?

The 33 physiological regressors were independent of our ROI selection. They were created based on voxel $x,y,z=[0,0,0]$ which is a representative choice of slice given slices were acquired interleaved and with a very short TR (0.72). This was discussed with the person who wrote the

analysis code for the PNM toolbox (Mark Jenkinson). This detail has been added to the manuscript as follows:

Methods (p.20):

“The physiological regressors were created based on voxel $x,y,z=[0,0,0]$ given slices were acquired interleaved and with a very short TR (720ms).”

Q8. “Variance of the noise normalised” (line 491ff) step should be explained in one more sentence: The data have been denoised by multiple regression before, so what is considered “noise” after that? Do the authors want to express that the variance of the residuals (the remaining signal after regression) was normalized? If so, the term “residuals” seems better suited

We followed a procedure adopted by (Smith et al., 2015) which is why we cite this publication). We have now included an additional sentence to explain it. Basically, this step is removing the strongest PCA components, i.e., structured signal/noise, so that the remaining, unstructured noise can be used to estimate the normalisation factor. We will also publish all scripts to allow other researchers to replicate our steps in case of any remaining uncertainties (which has already been done here <https://osf.io/egm2r/> for our previous work which relies on very similar procedures and where the noise normalisation was identical). We used the following code snippet on each run before concatenating runs together:

```
grot=demean(double(dtserie.cdata)');  
[uu,sss,vv]=ss_svds(grot,30);  
vv(abs(vv)<2.3*std(vv(:)))=0;  
stddevs=max(std(grot-uu*sss*vv'),0.001);  
grot=grot./repmat(stddevs,size(grot,1),1);
```

We have added the following text to the manuscript (Methods, p.20):

“The resulting rs-fMRI data were demeaned and the variance of the noise in the data was normalised as described in (Glasser et al., 2016), i.e., the normalisation factor was estimated on unstructured noise after the strongest PCA components containing structured signal/noise were removed.”

Q9. The 5 mm smoothing refer to the surface, correct?

Yes, that's correct and has now been clarified in the text. Subcortex was not subjected to any further smoothing.

Methods (p.20):

“For the creation of each of the three group connectomes, we applied additional smoothing **to the surface, but not to subcortical structures** (sigma=5mm). Analyses on individual time series did not involve any further smoothing.”

Q10. The authors mention “three connectomes” - they still refer to D1, R1 and R2? Maybe the syntax “D1+R1” is misleading and should be changed.

We have included a Supplementary Figure (S2B) to provide an overview of the data sets used across all analyses. We have pasted it here for information and now refer to it in the main Methods as follows:

In Methods (p.19):

“We included three cohorts from the full young-adult HCP data set (**Supplementary Figure S2B**): an initial dataset of n=200 3T participants (D1), a replication dataset of the same size (n=200, 3T; R1), and a third dataset containing all non-overlapping n=98 participants scanned at 7T (R2).”

Overview of data used

Parcellation of the hypothalamus

n=200	initial D1 (3T)	group connectivity and parcellation	Fig. 1, S2, S7, S9
n=200	replication R1 (3T)	replication group connectivity	Fig. S1
		replication parcellation	Fig. S3
n=98	replication R2 (7T)	replication group connectivity	Fig. S1
		replication parcellation	Fig. S3

Behaviour

n=400	dataset D1+R1 (3T)	behavioural factor analysis	Fig. 2
n=806	remaining dataset (3T)	replication behavioural factor analysis	Fig. S4
n=98	replication R2 (7T)	behavioural histogram plots	Fig. S4

Outlier rejection

Prediction (train → test)

n=200 (3T)	→	n=198 (3T)	Fig. 3-6, S5, S8
n=99 vs. n=99 (3T, robust prediction)	→	n=198 (3T)	Fig. S5
n=97 (7T)	→	n=398 (3T)	Fig. S6, S8

Supplementary Figure S2B

Q11. Major: “group average time course”: This needs a clearer presentation: rsfMRI data are not time locked and cannot be easily averaged across subjects. The 2nd step (‘cifti-correlation’) should be revised in that the maths behind the step and the tool are kept

separate. No technical term “cifti-correlation” is actually needed if the procedure is simply a Pearson correlation followed by a transformation to a Fisher’s z score.

Apologies, we can see that the wording was confusing. Because it is computationally difficult to **concatenate** all participant’s time-courses to compute an average connectome, a group-PCA based algorithm was used to retain the average and iteratively include more and more participant’s data (based on PCA). This approach was published in Neuroimage a few years ago Smith et al., 2014 and is suitable for large datasets and mathematically equivalent to the long concatenation followed by correlation (as shown in the original publication).

Here is a screenshot from the original parcellation explaining the algorithm:

Appendix A. MIGP MATLAB code

Below, $Y\{i\}$ is the $t \times v$ matrix for subject i , and should already have been processed to remove the mean of every column (timeseries), and, in the case of MELODIC, will also by default have had the (subject-specific) temporal variance normalisation procedure applied (Beckmann and Smith, 2004). The final output dimensionality is n .

```
r=randperm(s);           % used to randomise the order subjects are processed in
W=Y(r(1));              % copy first (randomly chosen) subject Y (t x v matrix) into W
for i=2:s                % main loop over all other subjects
    W=[W; Y(r(i))];      % concatenate W with the next subject
    [U,D]=eigs(W*W',t+2-1); % efficiently get the top "temporal" eigenvectors of W ...
    W=U'*W;              % ... and multiply these into W to get weighted spatial eigenvectors
end
W=W(1:n,:);            % output just the required number of strongest spatial eigenvectors
```

We have now expanded our explanation of this and removed the technical terms as suggested.

Methods (p.20)

“To obtain a group average connectome for each data set, the individual time series of all participants included in D1, R1, or R2, respectively, were ~~averaged~~ **combined** using the following procedure: first, we generated the group average time series using the algorithm ‘MIGP’, which **creates an average time series** using a group-level principal component analysis(Smith et al., 2014). Second, we created a dense connectome (**i.e., including the connectivity from each brain-ordinate to each other brain-ordinate**) by generating correlations of the ~~rows in the~~ **group** average time series ~~cifti file using the function cifti-correlation~~ **(using a Pearson’s correlation between each pair of brain ordinates followed by a transformation to Fisher’s z).**”

Q12. By “dense connectome” the authors refer to a voxel-by-voxel connector, correct? If

so, also here the phrasing should be revised and simply described that all surface/voxel time series (in contrary to regional time courses) were used.

Yes, that's exactly right. In our case, we don't refer to voxels because on the cortical surface we have vertices (subcortically, we still have voxels). In HCP data, Glasser and colleagues have coined the term 'brain ordinate'. But as the reviewer says, it is just the voxel-to-voxel (or brain-ordinate to brain-ordinate) connection (Pearson's r between the two time-series). We have changed the wording to make this clear in the paragraph highlighted in Q11 just above.

Q13. Figure 1A: Could the multi-colour bar possibly be changed to a hot/cool color principle? It is counterintuitive to color very negative values light green after dark blue.

Thanks for the suggestion – we have changed this as suggested. The updated figure looks as follows:

Manuscript Figure 1A

Q14. Decision to use absolute FC values and not preserve the signature: This needs some explanation as it is a bit counter-intuitive. Usually one would consider a complete flip of the sign (say 0.5 and -0.5) an indication for a different FC, and also the whole brain hypothalamus FC maps show that anterior and posterior parts of the brain are differently connected, even to the total HTH. I understand that global signal issues shift the total FC histogram and that the zero position is thus dependent on the preprocessing, but is it really correct to bundle together voxels with a flipped signature of their FC to a target? Did the more recently published Tian atlas for subcortical nuclei parcellation pursue this strategy? What was the average percentage of negative FC values flipped?

This is a very important question and one we have spent a lot of time thinking about too. First, to clarify, we only used the **absolute** connectivity to produce the hypothalamus parcellation. All other analyses, and thus importantly those related to stress, use the signed connectivity which we agree is very important.

For the parcellation, there is already a lot of anatomical knowledge from post-mortem work, so we know the ground truth (or something close to it) in terms of where to expect boundaries between nuclei. In particular, we know that nuclei should be relatively symmetrical across hemispheres (i.e., involving the same number in the left and right hemisphere) and should be spatially connected. We initially also thought that using the signed connectivity should give the best match to known subdivisions. However, we found when we tried this that the resulting parcels were not symmetrical and less anatomically plausible (see below).

When we thought about this a bit more, we realised that the relevant question for determining the correct approach might be to think about whether for subdividing a region into meaningful subunits, it is more important to focus on **strong vs weak** connectivity differences with its connected regions, or on **negative versus positive** connectivity differences with its connected regions. This is because the similarity matrix that gets fed into the hierarchical clustering algorithm simply looks at the similarity (Pearson's correlation coefficient) across hypothalamus voxels in terms of their connectivity to all other brain ordinates. Thus, if it uses the **signed connectivity**, the correlation coefficient will be driven by **negative versus positive** connectivity differences of the studied region with the rest of the brain. By contrast, when using absolute connectivity values (or just the positive or just the negative half of all connectivity values, ignoring the other half), it will be driven by **strong vs weak** connectivity differences.

Once we had realised this, we were not too surprised to find that the resulting clusters were more interpretable and closer to the ground truth we know from post-mortem and animal work when using the **absolute connectivity (or just positive connectivity)**. We thought this made sense because any regions that connect weakly to the hypothalamus would hardly influence the resulting parcellation when using signed connectivity values (the correlation would be driven by strongly positive and strongly negative functional connectivity values, but very little by weak connectivity values with either sign).

Indeed, to pursue this further, we performed separate parcellations on **just** positive or **just** negative connectivity which looked more plausible than the signed connectivity (using just the positive connectivity turned out to provide a parcellation very close to the one based on absolute connectivity that we used throughout the manuscript).

Below, we hope to illustrate this to the reviewer by showing parcellation steps 1-4 for these four different variants (plus the dendrogram until step 17): (1) absolute; (2) signed; (3) signed with just positive, (4) signed with just negative. This shows that the hemispheres only split in step 3 in the signed version and there are many more subdivisions in the left compared to right hemisphere (leading to an asymmetric parcellation). When only using the negative values, the posterior areas become disjointed/disconnected. Using only the positive signed functional connectivity results in a very similar parcellation to the one we used, especially in the left hemisphere, but the symmetry is still better in our parcellation using absolute connectivity values.

We have added some of this detail to Supplementary Figure S7.

A Signed versus non/signed parcellation of the hypothalamus in n=200 participants

Supplementary Figure S7: Comparing hypothalamus parcellations from signed and absolute connectivity in n=200 participants. Left: for the main parcellation presented in the manuscript (“non-signed” or absolute; left column), hierarchical clustering was performed on the similarity between hypothalamus voxels in terms of their profile of absolute functional connectivity to the rest

of the brain. When using absolute connectivity values, correlation coefficients in the similarity matrix will be driven by strong versus weak connectivity differences. Here we show how hypothalamus subdivisions evolved at different hierarchical clustering depths up until step 14. By contrast, for the “signed” parcellation (middle column), clustering was performed on the similarity matrix derived from both positive and negative functional connectivity. This means that correlation coefficients in the similarity matrix will be driven by negative versus positive connectivity differences of the hypothalamus with the rest of the brain. In the “signed positive” column, the parcellation was performed on a similarity matrix computed using only positive functional coupling values, i.e., excluding any regions with negative functional coupling with the hypothalamus.

The parcellation by Tian et al., (2020) used a slightly different approach (using PCA and a sparse graph). However, we have used the procedure described here in our own peer-reviewed work previously (Klein-Flügge et al., 2022) and are more familiar with it. Given that we invested a lot of time and thought into it, we would prefer to continue using it here.

We have added a few additional sentences explaining this rationale to the Methods and Discussion.

Methods (p.21)

“Using absolute values meant that the clustering was driven by similarities and differences in high versus low functional connectivity, rather than positive versus negative functional connectivity values. Note that almost identical clusters were obtained when using only positive connectivity values to parcellate the hypothalamus. By contrast, a parcellation using signed values was less anatomically plausible, lacking symmetry and producing unequal numbers of nuclei across hemispheres. For computing absolute values, the negative 46.65% of the connectivity matrix was flipped in its sign. Overall, the hypothalamus parcellation was thus driven by similarities and differences between hypothalamus voxels’ absolute connectivity to the rest of the brain.”

Discussion (p.14)

“In line with this, our parcellation of the hypothalamus into individual nuclei largely agreed with boundaries identified in careful anatomical and post-mortem work in human and non-human primates and extended simpler parcellations based on human rs-fMRI data (Kullmann et al., 2014; Mai et al., 2015; Makris et al., 2013; Neudorfer et al., 2020; Osada et al., 2017; Schönknecht et al., 2013) (**Fig 1, Supplementary Table S2**). We showed the robustness of our findings by replicating the average hypothalamus connectivity and its parcellation in two independent datasets (**Supplementary Fig S1**). Our parcellation was driven by the whole-brain connectivity pattern of each hypothalamus voxel. More precisely, we used the absolute connectivity between each hypothalamus voxel and the rest of the brain. When using signed connectivity values, correlation coefficients will be driven by negative versus positive connectivity differences of the studied region with the rest of the brain. By contrast, when using absolute connectivity values (or just the positive or just the negative half of all connectivity values, ignoring the other half), it will be driven by strong versus weak connectivity differences.

In our hands, the parcellation was anatomically most plausible when using absolute connectivity values (see **Supplementary Fig S7**). Future work may explore other parcellation approaches, for example using functional gradients previously explored in the striatum (Tian et al., 2020).”

Q15. Supplementary Table 3 is very good and informative. Could the comparison between the FC based parcellation and the competing 6 other ones be summarised in a few sentences in the result section? The 1:1 correspondence to Mai et al. is of striking, but it is difficult to follow to what degree these final steps were heuristic/human and not automated. Is there a way to quantify the overlap with these 6 systems despite them having a different N of subregions? Was there a level in the hierarchical tree with more than 7 regions per hemisphere that was rejected, and how was the cut-off decision made?

Thank you – we have now included some additional information in the results, as suggested (see below).

The parcellation was automated and used a hierarchical clustering algorithm, as described in the manuscript. The only step that was not automated was the decision about the depth of clustering. This decision about when **not** to continue further subdividing clusters was taken by carefully comparing more and less parsimonious parcellations (i.e., at shallower and deeper levels of the hierarchical clustering tree) in terms of their anatomical plausibility. We have now explained this further in the methods. The reason for stopping at the reported depth was because the next steps started to look asymmetric across hemispheres. We have now pasted images of two deeper clustering steps below to allow the reviewer to follow our rationale.

To explain those plots: our parcellation (clustering depth 14) is shown on top. In step 15 in the row below, the VM cluster (orange in row 14) splits into two blue clusters, but only in the left hemisphere. Each of those two subdivisions of VM only contains 5 voxels now. At depth 16, VM splits in the right hemisphere too (green-yellow in row 15 becomes two shades of blue in row 16), but those subdivisions are not matched across the left and right hemisphere. Thus, we did not consider this an improvement or anatomically plausible. It might be because the voxel numbers are getting quite low so that the signal may not be reliable enough to yield any further subdivisions.

Results (p.5):

To aid comparison with prior work and the reporting of our results, we assigned each cluster a putative nucleus label (**Fig 1D**). This process was largely guided by comparing the location of our clusters with the nuclei described in Mai et al. (Mai et al., 2016), but also incorporated other previous parcellations (Baroncini et al., 2012; Lechan & Toni, 2000; Neudorfer et al., 2020; Ogawa et al., 2020). Our own and Mai et al.'s parcellations are shown side by side in **Supplementary Figure S9**. The most dorsal anterior nucleus closely resembled what prior animal and human work commonly refer to as paraventricular nucleus, sometimes denoted as Pa, PVH, or PA (Baroncini et al., 2012; Lechan & Toni, 2000; Mai et al., 2016; Neudorfer et al., 2020; Ogawa et al., 2020), which we denote as PV throughout. The cluster ventral to PV most resembled the medial preoptic hypothalamic nucleus (MPO) across multiple previous parcellations (Baroncini et al., 2012; Lechan & Toni, 2000; Mai et al., 2016; Neudorfer et al., 2020; Ogawa et al., 2020) and the most ventral anterior cluster likely contained the supraoptic and suprachiasmatic nuclei elsewhere referred to as SChC, SChD, SCh, or SO and referred to as SO/SC here (Mai et al., 2016; Neudorfer et al., 2020). Moving more posteriorly, the cluster posterior to PV was labelled dorsomedial nucleus (DM) and the cluster ventral to DM ventromedial nucleus (VM), both

consistent with the same nuclei in previous parcellations (Baroncini et al., 2012; Lechan & Toni, 2000; Mai et al., 2016; Neudorfer et al., 2020; Ogawa et al., 2020) . The cluster posterior and ventral to VM was most consistent with the location of the mammillary body in Mai et al. (MM; Mai et al., 2016). Finally, the most posterior superior hypothalamus division was labelled posterior hypothalamic nucleus (PH), again consistent with the same nucleus in other work (Mai et al., 2016; Neudorfer et al., 2020; Ogawa et al., 2020). See **Supplementary Table S3** for a side-by-side comparison between our labels and those of previous hypothalamus parcellations.

Methods (p.22):

We chose the most anatomically plausible parcellation by comparing each higher order clustering step with the previous clustering step (see **Supplementary Fig S2**) and by comparing the location and size of the clusters obtained at increasing levels of depths to known anatomical subdivisions of the hypothalamus. In other words, while the clustering itself was fully automated, the depth of clustering at which we decided to stop was subjective and based on anatomical considerations. We expected reasonable symmetry across hemispheres and focused on identifying divisions between anterior-posterior and medial-lateral clusters within the hypothalamus. We therefore stopped at the clustering depth where these criteria started to break down. To evaluate the quality of the clustering, we compared clusters with existing post-mortem images (Makris et al., 2013a), and MRI work (Baroncini et al., 2012; Bocchetta et al., 2015; Kullmann et al., 2014; Lechan & Toni, 2000; Lemaire et al., 2011; Neudorfer et al., 2020; Schonknecht et al., 2013) as well as with atlases (Baroncini et al., 2012; Mai et al., 2016; Nieuwenhuys et al., 2008) to identify similarities in their anatomy. **Supplementary Figure S9** and **Supplementary Table S3** summarize similarities and differences between our and other existing parcellations.

Supplementary Discussion (p. 5-7):

[...] **Supplementary Table S3** shows our nomenclature side by side with these authors' terminology. Overall, there was good agreement between the parcellation of Mai et al.'s (Mai et al., 2016) and our parcellation, with their Pa nucleus corresponding to our PV nucleus, their PH agreeing with our PH, VMH with VM, and DMH with DM. In Lechan & Toni (Lechan & Toni, 2000), our combined region SO/SC was referred to as the infundibular, Sch, SO and arcuate nucleus.

The comparison of our nuclei to the volumetric automatic *in vivo* MR parcellations by Billot and colleagues (Billot et al., 2020) and the manual MRI parcellation including post-mortem histological validation on n=2 subjects by Makris and colleagues (Makris et al., 2013) showed that our anterior nuclei PV and DM of the hypothalamus were grouped together into their anterior-superior nucleus, and that their anterior-inferior nucleus corresponded to our MPO. An equivalent nucleus to our PH was not present in their parcellation (compare with Billot and

colleagues (Billot et al., 2020), Fig 3). They also stated that the boundary between their anterior-inferior and anterior-superior nucleus was faint.

Ogawa and colleagues (Ogawa et al., 2020), using resting-state fMRI, defined seven comparable nuclei of the hypothalamus. Five nuclei agreed well, namely PV, DM, MPO, PH, and VM. The region they referred to as arcuate nucleus was likely located within our MM, and their anterior nucleus matched our SO/SC well.

Neudorfer and colleagues (Neudorfer et al., 2020) parcellation, which was performed on *in vivo* T1/T2-weighted structural MR images, overall agreed well with our parcellation – our PV agreed with their paraventricular and dorsal periventricular nuclei. DM, VM, PH and MPO were relatively comparable between parcellations. Neudorfer's (Neudorfer et al., 2020) SO, SCh and AN likely correspond to our combined SO/SC, and our MM might be closest to their tuberomammillary nucleus. The key difference, where our and their parcellations did not agree well, was in terms of their large lateral nucleus. The lack of a clear separation between medial and lateral nuclei in our study made the comparison between this aspect of their parcellation and ours difficult. Using anatomical landmarks, we concluded that the lateral nucleus by Neudorfer and colleagues (Neudorfer et al., 2020) might be partly included within our PV, MPO, SO/SC, DM and VM.

The parcellation of Schonknecht and colleagues (Schonknecht et al., 2013) used *in vivo* diffusion tensor imaging (DTI) as well as T1/T2-weighted structural images. This parcellation was less detailed, delineating three subdivisions in lateral, anterior, and posteromedial hypothalamus.

In summary, our parcellation and the resulting boundaries between nuclei of the hypothalamus were largely consistent with the literature – in particular for PV, DM, VM, MPO, and PH. Other nuclei differed anatomically across different parcellations (e.g., SO/SC, MM). The anterior hypothalamus consists of SO/SC, but also contains the preoptic area including the MPO. SO/SC is sometimes also referred to as the infundibular or arcuate nucleus (Lechan & Toni, 2000) or as the inferior tuberal nucleus (Billot et al., 2020). The middle area of the hypothalamus consists of the PV, DM, and VM. We suggest that our PH is part of the posterior area of the hypothalamus. However, it may contain parts of the middle area. The posterior and mammillary body of the hypothalamus can be identified consistently across the literature and are contained in our MM nucleus.

To address the final point in the reviewer's question about quantifying the overlap between different parcellations, we tried to get hold of masks of the various parcellations we compared in Table 3, but we could not get hold of these and thus were unable to quantify the overlap. This is also something R2 asked about. In response to R2, we have improved our nuclei labels which are now slightly changed throughout the manuscript. We have also now added a new

supplementary Figure (S9) to enable a detailed visual side-by-side comparison. This figure looks as follows:

Comparison between our parcellation and Mai et al. (2016)

Supplementary Figure S9: Side-by-side comparison with hypothalamus atlas. Comparison between our parcellation and hypothalamus nuclei in the Atlas of the human brain in Mai et al., (2016). The coronal sections of Mai et al., (2016; left) were overlaid with our identified nuclei (middle) which are again shown on the right. Our nuclei are indicated as follows: PV - paraventricular nucleus (dark blue), MPO – medial preoptic nucleus (middle blue), DM - dorsomedial nucleus (light blue), VM - ventromedial nucleus (green), PH - posterior hypothalamic nucleus (yellow), MM - mammillary body (red), SO/SC - supraoptic and

suprachiasmatic nucleus (dark red and brown). Atlas sections (left and middle column) from the Atlas of the human brain were used to facilitate the delineation of hypothalamic and surrounding structures: AVPe - anteroventral periventricular hypothalamic nucleus; BNST - bed nucleus of stria terminalis; DMH - dorsomedial hypothalamic nucleus; fx - fornix; HDB - horizontal limb of the diagonal band; LH - lateral hypothalamus; ML - medial mammillary nucleus - lateral part; MM - mammillary bodies; MPO - medial preoptic nucleus (includes MPO, MPOM); PA - paraventricular nucleus (includes PaAP, PaPC, PaMC, PaPo and PaD); PH - posterior hypothalamus; Sch - suprachiasmatic nucleus (includes SChD, SChC); SO - supraoptic nucleus (includes SO, SOVM, SODL); SuM - supramammillary nucleus; TM - tuberomammillary nucleus; VMH - ventromedial nucleus (includes VMH, VMHVL, VMHDM). Reproduced with permission from Mai JK, Paxinos G, Voss T (2016): Atlas of the Human Brain, 4th ed. San Diego: Elsevier Academic Press.

Q16. Regressing out confound variables from the regional FC matrix: Wasn't the preprocessing designed to control for these influence? Is it a good idea to correct a second time? I am not convinced that clinical variables such as BMI and education need to be corrected for. BMI, btw, can be increased due to chronic stress, so there is a risk to loose true effects. Were at least some of the analyses performed w/o the second nuisance variable correction?

The reviewer is correct that confounding variables such as motion and physiological noise are corrected for as much as possible in the pre-processing of individual participants. However, we know that such corrections are imperfect (e.g., motion correction performs worse in someone who had a cough and therefore may have had lots of sudden motion jumps). Therefore, when we moved to our key between-subject comparisons, we wanted to be conservative and make sure the results would not be driven by remaining differences between participants (e.g., differences in the quality of motion correction). We agree with the reviewer that this is unlikely, but we nevertheless included the average motion, as well as whole brain volume, intracranial volume, sex, age, BMI, race, ethnicity, and years of education to account for any variance in functional connectivity values between-subject that might be explained by these factors.

However, we overlooked the potential relationship between BMI and stress, and we agree that it is important to check that our results hold with/without inclusion of these confounds.

We have therefore now rerun our key analysis without inclusion of these confounds, as shown below. This has made no difference to our conclusion.

The figure below shows that, in our original analysis, we were able to predict stress in an independent test group using all edges (with weights trained in the training group) at $r=0.265$; $p=0.0007$ (left column and see Manuscript Figure 3 and 4). Without the use of confounders for BMI and education this prediction is slightly higher ($r=0.279$; $p=0.0003$; middle column). Without the inclusion of any confounders, the stress prediction is slightly worse ($r=0.256$; $p=0.0013$; right column). Most importantly, the conclusions are not affected by the precise choice of confounding variables.

Figure: Comparison between using different sets of confounders in our analysis. We compared our original analysis with confounders (left column – see Manuscript Figures 3 and 4), with equivalent analyses using minimal confounders (middle column) or no confounders (right column). This did not change any of the conclusions.

We now mention this in the manuscript:

Methods (p.26):

Next, confound variables (whole brain volume, motion, and intracranial volume) were regressed out of the connectivity matrix X as described in (Klein-Flügge et al., 2022; Smith et al., 2015). Additional confounds included the sex, age in years, body-mass index (BMI), race, ethnicity, and years of education completed. A total of 11 confounds were thus regressed out of X . **This was done to ensure that our results would not be driven by remaining differences between participants (e.g., differences in the quality of motion correction). Nevertheless, we confirmed that key results held without inclusion of these confounds, especially given BMI’s potential relationship with stress.**

Q17. Major: “Robust regression coefficient”: Application of regression coefficients between FC and stress cores from training to test group: It is not entirely clear, what kind of basic regression analysis was actually performed to connect the 105 edges with the single stress score: (a) Massive univariate (=separate) or (b) some truly multivariate analysis or (c) a multiple linear regression with all FC values predicting one stress value?

It seems that an SVM or some other type of supervised machine learning would also be a good candidate. The authors should explain this more comprehensively.

Thank you – this is clearly an important point. We spent a long time thinking about what the most sensitive analysis approach would be. Because different potential FC predictors are highly correlated, we initially considered combining them into fewer less correlated predictors (e.g., using dimensionality reduction). However, this would have made an anatomically meaningful interpretation difficult because a single predictor would have become the weighted sum of multiple edge's functional connectivity values. Because we were particularly interested in specific anatomical circuits, we decided that it was important to keep the anatomical interpretability and work with individual edges/connections in the brain. To allow us to do this, given the high correlations among regressors, we first performed separate predictions (regressions) for each edge's FC in the **training** cohort to estimate the contribution / predictive power of an individual edge (as in the reviewer's point (a)). However, this was just done to derive 'weights', not to test the predictive performance. To predict the **left-out test cohort**, we then used these weights in a single regression to predict stress scores (i.e., as in the reviewer's point (c)), using a simple weighted sum. This is the same procedure that we published in our prior work (Klein-Flügge et al., 2022).

We have now clarified this thought process further and made the details clearer in our Methods:

Methods (p.26 and p.27):

“In a first step, to establish whether hypothalamus connectivity captured meaningful variance related to stress, we estimated robust linear regression weights (function `robustfit` in MATLAB), separately for all functional connectivity edges in **X** (105) to capture their relationship with the stress scores in **y**. **This first step was performed to train robust regression weights for individual edges, but not used to evaluate predictions.**

[...]

Having tested the robustness of hypothalamus connectivity relationships with stress, we moved on to our key analysis to establish if hypothalamus nuclei connectivity was sufficient to predict stress in an independent cohort. To test this, we applied the robust regression coefficients estimated **independently** for all 105 edges in the train group to the functional connectivity estimates from the test group, **using a weighted sum of all edge's weights and all edges' individual connectivity values**, to derive individual predicted stress scores **for all participants.**”

Q18. In the chapter “Predictions using nuclei versus whole HTH connectivity”, how were the other subcortical nuclei dealt with (for example, the NAcc etc.) - were they kept in?

Apologies that this was not clear. The reviewer is right, only the hypothalamus was changed, all other ROIs were represented in exactly the same way in both analyses to make them comparable.

Like in the hypothalamus nuclei version, we used the ROI mean time series (each ROI treated with equal weight independent of size) in this control analysis.

We have now clarified this in the text:

Methods (p.29):

To examine whether parcellating the hypothalamus (and amygdala) led to improved prediction accuracies, we repeated the regression procedure, once with edges from the 15 ROIs to the *entire* hypothalamus instead of its individual nuclei ($n=15$ edges instead of $n=7 \times 15=105$ edges; **Fig 5A-D**, left), and once with edges from the same ROIs but with the amygdala included as a whole instead of its individual nuclei ($15-6=9$ ROIs) both for the whole and the parcellated hypothalamus ($n=9$ edges or $n=7 \times 9=63$ edges, respectively; **Fig 5A-D**, right). **In other words, compared to previous analyses, no change was made to any ROIs except the hypothalamus and/or amygdala. Relationships with stress were established between hypothalamus and the same set of ROIs as before (e.g., NAcc, LC, etc), just changing the spatial precision with which the hypothalamus (and amygdala) were represented.**

Q19. Specificity of the FC matrix predicting the stress scores: I would like to suggest a different null model as FC is so powerful that often very indirectly, ‘networks’ predict a behavioural phenotype. Could the authors define 15 cortical regions (e. g. cingulate cortex or prefrontal, each of them with the same ROI size) and estimate the correlational predictive power in the exactly same way? I am not fully convinced of the claimed hypothalamic specificity.

We are slightly concerned that the choice of 15 similarly sized cortical ROIs would be somewhat arbitrary and that it would be a suboptimal comparison because the fMRI signal-to-noise ratio in subcortical regions is worse than in cortex. We had strong a priori reasons to look at the hypothalamus (despite it being a region with more signal loss), but we are also not claiming that a prediction of stress scores is not possible using another brain circuit. We hope the reviewer is happy with our decision to tone down the specificity instead of conducting the suggested analysis.

We would like to emphasize that this study was designed around the hypothalamus because this region has been neglected in human neuroimaging (especially compared with the literature on stress in rodents). Thus, we believe it is an important contribution to show that a stress prediction is possible using this very small subcortical circuit. However, we now more explicitly acknowledge in a new paragraph in the Discussion that a similar prediction may also be possible using other (e.g., cortical) circuits.

Discussion (p.17):

“Importantly, from the work presented here, we cannot conclude that predictions of stress can only be achieved when based on hypothalamus functional connectivity. This study was designed around the hypothalamus, a region that has been relatively neglected in human neuroimaging, to

examine whether this small subcortical circuit carries relevance for predicting stress. However, similar prediction may be possible using other circuits. Indeed, it is likely that other regions known to play a role in the context of stress, both in subcortex (e.g., amygdala, hippocampus) and cortex (e.g., ACC) contribute to stress predictions and could improve prediction accuracies. Related to this, the signal-to-noise ratio can vary considerably across regions, and this would need to be considered when making any comparison between the predictive information about stress in cortically- and subcortically-centred networks.”

Q20. In the same direction, in the plot of the rising number of edges used, would the proportion of edges occupied by the HTH be marked in color within each bar?

We are unsure exactly what the reviewer is suggesting. All edges included in all predictions in our manuscript are with one of the nuclei of the hypothalamus, as it is used as the ‘hub’ that the other ROIs connect with. Thus, 100% of the connections at all points in the rising number of edges plot are with the hypothalamus. Apologies if we misunderstood the suggestion.

RESULTS

Q21. The paragraphs of the result section should more directly match the paragraphs of the methods section.

Thanks – we have now tried to streamline the section headings. There are some headings that only exist in the Methods because we go into more details in the Methods compared to the Results. Updated headings are shown in red (note there is a 60-character limit at Nat Comms):

Results headings	Corresponding methods headings
In vivo parcellation of the human hypothalamus	Parcellation of the hypothalamus
Nuclei-specific hypothalamus functional connectivity	ROI selection for nuclei-specific hypothalamus connectivity
Extracting a dimensional marker of stress	Factor analysis for extracting a dimensional stress marker
Relating stress and hypothalamus nuclei connectivity	Relating stress and hypothalamus nuclei connectivity
Out-of-sample stress prediction using hypothalamus coupling	Out-of-sample stress prediction using hypothalamus coupling

Characterising hypothalamus networks predictive of stress	Characterising hypothalamus networks predictive of stress
Nuclei-specific versus whole hypothalamus predictions	Nuclei-specific versus whole hypothalamus predictions
Behavioural specificity of predictions for stress	Behavioural specificity of predictions for stress

Q22. There are several doublings between the methods and the result sections: One of many examples is the topic of the new split of the test and train samples due to different stress scores.

Thanks, we have now gone through the manuscript to removed any unnecessary duplication. However, because the results section comes first at Nat Comms, we still briefly explain our methods at a few places in the results because we felt they are sometimes essential to follow the logic of how the results were obtained.

Examples of deletions from the Results:

~~Upon inspection of the stress scores in our original split of the $n=400$ 3T participants into two sets of $n=200$ participants, we noticed that stress scores were not quite comparable between the two groups (trend wise difference in their means: $t(398)=-.99$; $p=.07$; $CI=[-.03,1.0]$).~~

~~We also corrected the functional connectivity matrix for confounding variables (Klein-Flügge et al., 2022; Smith et al., 2015) and used robust regressions throughout (see Methods).~~

To carry out this test, we iteratively added one edge at a time, from 1 to 105, based on their order of importance (absolute regression coefficient) in the training group. ~~In other words, we ordered edges based on their absolute robust regression coefficients estimated in the training group.~~

Q23. The used softwares and tools could be aggregated into one separate methods section.

Thanks, we tried to do this now, but felt it would require more words and repetition. By contrast, referring to the tools in passing when describing the process is quite brief. We hope the reviewer is okay with us keeping it as it is.

Q24. It is surprising that the seven moderately correlated self reported stress items result in only one common factor. What were the threshold criteria for this solution? Is it still only one resulting factor when D1 and R1 are pooled?

Yes, the stress factor comes out robustly as one factor, both when pooling D1/R1 or when performing separate factor analyses for each dataset - please see the plots below comparing different groups of participants in different rows. In each case, we used the parallel score (Scree test) to extract the 'knee' (bend) in the explained variance, which is a standard measure.

Dataset:	Scree test	Factor loadings
D1		
R1		
R1+D1 (n=400)		

Q25. The authors report that the application of the factor loadings to the 7 T sample resulted in a lower range of the (aggregated) stress score. Was this also reflected in lower stress scores at the level of the original 7 questionnaires? If so, the translation to the 7 T still seems valid, as lower stress score range is (correctly) found. Please discuss.

Yes, we agree that this is a valid/true result and that there is no problem with the translation from individual scores to an aggregate dimensional score in the 7T cohort. However, a lack of variance makes it difficult to look at between-subject variability in stress scores (this becomes clear when imagining a hypothetical extreme case, where all participants have the exact same stress level;

then it would be impossible to study any relationships with stress). We are showing the histograms for the seven original questionnaire scores in Supplementary Figure S4 (which we have pasted below again for convenience), and we have summarized them in numerical figures in the table below. As you can see, while the means are mostly comparable, the variance is consistently smaller in the 7T sample:

Supplementary Figure S4A: Distribution and replication of factor analysis-derived dimensional stress score. Histograms of stress scores included in the factor analysis for (1) all 3T HCP participants included in this study, n=400 (before outlier rejection), (2) all remaining 3T HCP participants not included in this study (because of lacking resting-state data or physiological noise recordings); n=806 (n=1206-400), and (3) all 7T participants included in this study, n=98 (before outlier rejection). This shows lower variance in stress scores in the 7T cohort.

3T n=400	7T n=98
PercStress - Mean: 48.254, Var: 91.1599	PercStress - Mean: 47.5429, Var: 60.5936
SelfEff - Mean: 50.953, Var: 71.2623	SelfEff - Mean: 50.9153, Var: 56.9821
AngAggr - Mean: 51.5645, Var: 77.6604	AngAggr - Mean: 51.4878, Var: 69.7227
FearAffect - Mean: 49.983, Var: 71.2242	FearAffect - Mean: 49.8041, Var: 44.1173
FearSomat - Mean: 51.4555, Var: 74.5825	FearSomat - Mean: 50.7837, Var: 54.296
EmotSupp - Mean: 51.74, Var: 91.1383	EmotSupp - Mean: 50.8612, Var: 81.8838
NEORAW_11 - Mean: 3.38, Var: 1.2988	NEORAW_11 - Mean: 3.5408, Var: 0.86945

To check whether this is simply reflective of the smaller number of participants (n=98 at 7T, but n=400 at 3T), we repeatedly selected a random subset of participants of size n=98 from the total

3T pool of n=400 participants. The mean and the variance, averaged across n=100 subsamples, is shown below, compared again against the 7T data, now with the same size. Again, the variance in the 7T dataset – now balanced for size – was consistently lower.

Average mean/variances from i=100 iterations, each using a random subset of n=98 participants of the total n=400 3T participants	7T n=98
PercStress - Mean: 48.4162, Var: 91.6478	PercStress - Mean: 47.5429, Var: 60.5936
SelfEff - Mean: 50.7667, Var: 71.1442	SelfEff - Mean: 50.9153, Var: 56.9821
AngAggr - Mean: 51.7307, Var: 78.033	AngAggr - Mean: 51.4878, Var: 69.7227
FearAffect - Mean: 49.9979, Var: 70.446	FearAffect - Mean: 49.8041, Var: 44.1173
FearSomat - Mean: 51.5337, Var: 73.6416	FearSomat - Mean: 50.7837, Var: 54.296
EmotSupp - Mean: 51.5527, Var: 93.5464	EmotSupp - Mean: 50.8612, Var: 81.8838
NEORAW_11 - Mean: 3.383, Var: 1.2897	NEORAW_11 - Mean: 3.5408, Var: 0.86945

DISCUSSION

Q26. The cohorts were not ‘transdiagnostic’ in the clinical sense. If diagnoses were made, they should be reported. It seems that stress levels in the population with not a lot of consideration of clinical significance are reported. This is an important point and no claims for clinical translation should be made at this level. Correlations can be totally different in disease as here compensatory processes play in.

We completely agree – apologies for the imprecise wording. We now refer to them as **dimensional** rather than transdiagnostic scores throughout the manuscript. The HCP participants are indeed young healthy volunteers who do not currently have a diagnosis. When referring to transdiagnostic markers, we were trying to be in line with a growing literature where even in the normal population, transdiagnostic scores are used as a term. Nevertheless, we agree that it is a misleading term.

We are not making any claims for a direct clinical significance of our work and have also toned down the discussion sections that mention any link to clinical applications, making sure we are referring to future work rather than the findings reported here.

Example changes in the discussions (p.16-18):

Given our analyses relied on healthy volunteers, our interpretation of the underlying circuit changes should be confirmed in studies using longitudinal designs or direct interventions in clinical populations to establish causality.

Given our work solely relied on healthy participant's data, future longitudinal work should examine whether changes in hypothalamus coupling may be a precursor to mental illness, and whether targeted interventions such as those using transcranial ultrasonic stimulation (TUS ; Bongioanni et al., 2021; Verhagen et al., 2019) might be able to rebalance the hypothalamus networks identified here (Haber et al., 2023; Hamani et al., 2011; Huys et al., 2016).

[...]

Exploring this in both healthy and clinical populations could have implications for treating a range of mental health disorders whose onset is often preceded by a prolonged experience of stress.

Q27. Competing methods such as functional gradients to parcel out regions should be discussed (Tian paper). In turn, the principle driving the parcellation here (similarity of regional connectivity of a HTH voxel).

Given space limitations, we are now briefly mentioning alternative methods and the potential for using functional gradients in the Discussion. It is possible that functional gradients may be more meaningful for characterizing boundaries between cortical regions or for cortical loops with the striatum where it is known that a gradient-like organization is present. This is not the case in the same way for the hypothalamus or amygdala which are an agglomeration of distinct but spatially adjacent nuclei. We have also added a sentence explaining what information drives our parcellation. We note that it is not the regional connectivity but the whole-brain connectivity of each hypothalamus voxel which we have also clarified in the Methods in case it was unclear.

Discussion (p.14):

In line with this, our parcellation of the hypothalamus into individual nuclei largely agreed with boundaries identified in careful anatomical and post-mortem work in human and non-human primates and extended simpler parcellations based on human rs-fMRI data (Kullmann et al., 2014; Mai et al., 2015; Makris et al., 2013; Neudorfer et al., 2020a; Osada et al., 2017; Schönknecht et al., 2013) (Fig 1, Supplementary Table S2). We showed the robustness of our findings by replicating the average hypothalamus connectivity and its parcellation in two independent datasets (Supplementary Fig S1). Our parcellation was driven by the whole-brain connectivity pattern of each hypothalamus voxel. More precisely, we used the absolute connectivity between each hypothalamus voxel and the rest of the brain. When using signed connectivity values, correlation coefficients will be driven by negative versus positive connectivity differences of the studied region with the rest of the brain. By contrast, when using absolute connectivity values (or just the positive or just the negative half of all connectivity

values, ignoring the other half), it will be driven by strong versus weak connectivity differences. In our hands, the parcellation was anatomically most plausible when using absolute connectivity values (see **Supplementary Fig S7**). Future work may explore other parcellation approaches, for example using functional gradients previously explored in the striatum (Tian et al., 2020).

Methods (p.21):

Overall, the hypothalamus parcellation was thus driven by similarities and differences between hypothalamus voxels' absolute connectivity to the rest of the brain.

Q28. As mentioned before, the use of the absolute FC values that mirror anti-correlation to the positive side should be re-visited in the discussion.

We now mention this again in the discussion as pasted above in response to Q27. It is important to note that only the parcellation is affected by this choice, not the relationships with stress nor the related networks we describe in subsequent analyses.

Q29. As also mentioned before, not including hippocampal information in the prediction network will be difficult to accept for many stress researchers. It might be discussed that at least the connectivity between the HTH and the hippocampus was considered for the parcellation.

We have now added our new analyses to the discussion, and they are also included in the Supplement, as discussed above.

Discussion (p.16):

The hippocampus shows sign of degeneration following stress (Levone et al., 2014; McEwen et al., 2016) and is therefore an important subcortical region not included a priori here. We did not want to treat the hippocampus as a homogenous region; however, since starting this work, Tian and colleagues (Tian et al., 2020) published a detailed hippocampus parcellation, which we included a posteriori into our analyses for comparison (**Supplementary Information** and **Supplementary Figure S8**). Surprisingly, inclusion of the hippocampus subregions as additional ROIs did not improve stress predictions. While these new analyses do not question the importance of the hippocampus in stress predictions *per se*, they do suggest functional connectivity between specific hippocampus-subdivisions to hypothalamus-nuclei does not carry additional predictive variance for stress. Coupling between the hippocampus and regions other than the hypothalamus should be considered in future work.

Q30. Literature connecting the resting state to explicit HPA axis markers could be cited.

We are aware of a literature looking at large-scale resting-state network changes during acute stress, such as, for example, work showing that the salience network is upregulated while the executive control network is downregulated during acute stress (e.g., Hermans et al., 2014 now cited in the manuscript, Introduction, p.2). Is this what the reviewer has in mind? We are not aware of and could not find a similar literature related to prolonged/chronic stress but would be happy to include it if the reviewer could point us to the relevant work.

GENERAL/STYLE

Q31. Computer syntax style in reporting regions (e. g. “RN_DR” with an underscore) should be rather avoided in the main text.

We have changed this throughout – thank you.

Q32. p-values and r-values should be reported with uniform precision in terms of the number of digits after the comma.

We now always report 3 digits after the comma throughout.

Reviewer #2 (Remarks to the Author):

I think this is a very interesting study of the hypothalamic nuclei network. The hypothalamus network related to stress, in particular, is quite new and exciting, and the analysis procedures seem sound overall. I would like to raise some points of details regarding interpretation of the results, which I hope may improve this manuscript.

Thank you very much for your positive feedback and helpful suggestions.

I think the parcellation results need further consideration. The supraoptic, suprachiasmatic and medial preoptic nuclei are located in the anterior part of the hypothalamus, but these nuclei were included in the “PHs (posterior superior hypothalamic nucleus)” (Table S3). It seems to me that these nuclei are included in the AH or VMa of this study. It also seems possible that the PHs of this study corresponds to the posterior hypothalamic nucleus and that the PH of this study corresponds to the mammillary body. It would be useful to classify the nuclei more quantitatively by referring to the coordinates of the nuclei reported in previous studies, and please make sure entire parcellation results appear more interpretable.

We have now carefully reconsidered our table S3 thanks to this comment. We agree that some of our nuclei labels and descriptions of their equivalent in other parcellations needed improving. While the location of the identified subdivisions and thus the parcellation itself still holds, we have reworked our labels throughout the entire manuscript, including Table S3. In brief

- the supraoptic, suprachiasmatic and medial preoptic nuclei are now all part of more anterior hypothalamus nuclei, as follows:
- we think the supraoptic and suprachiasmatic nuclei should have always been part of our former VMa, apologies for the oversight; we have now relabelled what we used to call VMa to SO/SC which seems more in line with the majority of parcellations including those of Baroncini et al. (2012), Mai et al. (2016) and Neudorfer et al. (2020) and this nucleus contains supraoptic (SO) and suprachiasmatic (SChC and SChD)
- the nucleus we originally labelled anterior hypothalamus (AH), we have now relabelled MPO because this resembles the medial preoptic nucleus as in (Baroncini et al. (2012), Mai et al. (2016), Neudorfer et al. (2020) and Ogawa et al. (2020) as highlighted by the reviewer
- As suggested, we agree that what we used to call PHs actually most likely corresponds to the posterior hypothalamic nucleus, so we have relabelled it to PH and corrected this in the table
- What we used to call PH has now been relabelled to MM for mammillary body, again we agree with the reviewer’s assessment – thank you
- PV, DM and VM have remained unchanged.

This is our updated table S3:

Our nuclei	Mai et al., 2016	Baroncini et al., 2012; Lechan & Toni, 2000	Billot et al., 2020; Makris et al., 2013	Ogawa et al., 2020	Neudorfer et al., 2020
-------------------	--------------------------------	--	---	---------------------------	-------------------------------

PV	paraventricular nucleus (Pa + subdivisions)	paraventricular nucleus (Pa)	anterior-superior (a-sHyp)	paraventricular nucleus (PVH)	paraventricular nucleus (PA), dorsal periventricular nucleus (DP),
MPO	medial preoptic nucleus (MPO)	preoptic area (MPO)	anterior-inferior (a-iHyp)	medial preoptic nucleus (MPO)	medial preoptic nucleus (MPO), PE
SO\SC	supraoptic (SO), suprachiasmatic** (SChC and SChD)	infundibular or arcuate nucleus (Inf) supraoptic nucleus (SO)	inferior tuberal (infTub)	anterior hypothalamic nucleus (AH)	Arcuate (AN) suprachiasmatic** (SCh), supraoptic (SO),
DM	dorsomedial nucleus (DM)	dorsomedial nucleus (DMH)	anterior-superior (a-sHyp)	dorsomedial nucleus (DMH)	dorsomedial nucleus (DM)
VM	ventromedial nucleus (VM)	ventromedial nucleus (VMH)	superior tuberal (supTub)	ventromedial nucleus (VMH)	ventromedial nucleus (VM), possibly AHA
MM	Inf, mammillary body (MM), intercalated nucleus	medial and mammillary nucleus (MM and LM)	posterior (posHyp)	arcuate nucleus (ARC)	tuberomammillary, lateral (TM)
PH	posterior (PH),	lateral (LHAp), posterior hypothalamic nucleus (PH),	--	posterior hypothalamic nucleus (PH)	posterior (PH),

We have also, in part based on a suggestion by R1, included a new supplementary figure. We were initially hoping to provide a quantitative measure of overlap between the nuclei we report here and those reported in previous atlases, but this is difficult to do because most published studies did not publish the masks of their nuclei alongside their studies (and they typically do not mention the coordinates of the centre of mass either). Thus, we have opted for a detailed visual side-by-side comparison to guide the reader in making their own judgement. Please find the new figure below and in **Supplementary Fig S9**.

Comparison between our parcellation and Mai et al. (2016)

Supplementary Figure S9: Side-by-side comparison with hypothalamus atlas. Comparison between our parcellation and hypothalamus nuclei in the Atlas of the human brain in Mai et al., (2016). The coronal sections of Mai et al., (2016; left) were overlaid with our identified nuclei (middle) which are again shown on the right. Our nuclei are indicated as follows: PV -

paraventricular nucleus (dark blue), MPO – medial preoptic nucleus (middle blue), DM - dorsomedial nucleus (light blue), VM - ventromedial nucleus (green), PH - posterior hypothalamic nucleus (yellow), MM - mammillary body (red), SO/SC - supraoptic and suprachiasmatic nucleus (dark red and brown). Atlas sections (left and middle column) from the Atlas of the human brain were used to facilitate the delineation of hypothalamic and surrounding structures: AVPe - anteroventral periventricular hypothalamic nucleus; BNST - bed nucleus of stria terminalis; DMH - dorsomedial hypothalamic nucleus; fx - fornix; HDB - horizontal limb of the diagonal band; LH - lateral hypothalamus; ML - medial mammillary nucleus - lateral part; MM - mammillary bodies; MPO - medial preoptic nucleus (includes MPO, MPOM); PA - paraventricular nucleus (includes PaAP, PaPC, PaMC, PaPo and PaD); PH - posterior hypothalamus; Sch - suprachiasmatic nucleus (includes SchD, SCHC); SO - supraoptic nucleus (includes SO, SOVM, SODL); SuM - supramammillary nucleus; TM - tuberomammillary nucleus; VMH - ventromedial nucleus (includes VMH, VMHVL, VMHDM). Reproduced with permission from Mai JK, Paxinos G, Voss T (2016): Atlas of the Human Brain, 4th ed. San Diego: Elsevier Academic Press.

Finally, we have also added some more detail to our supplementary discussion:

Supplementary Discussion (p. 5-7):

[...] **Supplementary Table S3** shows our nomenclature side by side with these authors' terminology. Overall, there was good agreement between the parcellation of Mai et al.'s (Mai et al., 2016) and our parcellation, with their Pa nucleus corresponding to our PV nucleus, their PH agreeing with our PH, VMH with VM, and DMH with DM. In Lechan & Toni (Lechan & Toni, 2000), our combined region SO/SC was referred to as the infundibular, Sch, SO and arcuate nucleus.

The comparison of our nuclei to the volumetric automatic *in vivo* MR parcellations by Billot and colleagues (Billot et al., 2020) and the manual MRI parcellation including post-mortem histological validation on n=2 subjects by Makris and colleagues (Makris et al., 2013) showed that our anterior nuclei PV and DM of the hypothalamus were grouped together into their anterior-superior nucleus, and that their anterior-inferior nucleus corresponded to our MPO. An equivalent nucleus to our PH was not present in their parcellation (compare with Billot and colleagues (Billot et al., 2020), Fig 3). They also stated that the boundary between their anterior-inferior and anterior-superior nucleus was faint.

Ogawa and colleagues (Ogawa et al., 2020), using resting-state fMRI, defined seven comparable nuclei of the hypothalamus. Five nuclei agreed well, namely PV, DM, MPO, PH, and VM. The region they referred to as arcuate nucleus was likely located within our MM, and their anterior nucleus matched our SO/SC well.

Neudorfer and colleagues (Neudorfer et al., 2020) parcellation, which was performed on *in vivo* T1/T2-weighted structural MR images, overall agreed well with our parcellation – our PV agreed with their paraventricular and dorsal periventricular nuclei. DM, VM, PH and MPO were relatively comparable between parcellations. Neudorfer's (Neudorfer et al., 2020) SO, Sch and AN likely correspond to our combined SO/SC, and our MM might be closest to their tuberomammillary nucleus. The key difference, where our and their parcellations did not agree well, was in terms of their large lateral nucleus. The lack of a clear separation between medial and lateral nuclei in our study made the comparison between this aspect of their parcellation and ours difficult. Using anatomical landmarks, we concluded that the lateral nucleus by

Neudorfer and colleagues (Neudorfer et al., 2020) might be partly included within our PV, MPO, SO/SC, DM and VM.

The parcellation of Schonknecht and colleagues (Schonknecht et al., 2013) used *in vivo* diffusion tensor imaging (DTI) as well as T1/T2-weighted structural images. This parcellation was less detailed, delineating three subdivisions in lateral, anterior, and posteromedial hypothalamus.

In summary, our parcellation and the resulting boundaries between nuclei of the hypothalamus were largely consistent with the literature – in particular for PV, DM, VM, MPO, and PH. Other nuclei differed anatomically across different parcellations (e.g., SO/SC, MM). The anterior hypothalamus consists of SO/SC, but also contains the preoptic area including the MPO. SO/SC is sometimes also referred to as the infundibular or arcuate nucleus (Lechan & Toni, 2000) or as the inferior tuberal nucleus (Billot et al., 2020). The middle area of the hypothalamus consists of the PV, DM, and VM. We suggest that our PH is part of the posterior area of the hypothalamus. However, it may contain parts of the middle area. The posterior and mammillary body of the hypothalamus can be identified consistently across the literature and are contained in our MM nucleus.

Please correct the Z coordinates in Fig. 1B.

Thanks, this has been corrected.

It would be helpful to provide a table listing the 22 important connections to clarify specific networks of interest.

Thanks, we have included such a table as a new Supplementary table (S5) now. For convenience, we have pasted this table below as well. As requested, it shows the top 22 edges that contribute to predicting stress in the main analysis shown in Figure 4A (i.e., at the peak prediction accuracy of n=22):

number of edges:	edges
1	SO/SC-RN-MR
2	MPO-amyg-Ce
3	PV-LC
4	PV-NAc,
5	SO/SC-amyg-LaI
6	MPO-amyg-B
7	MPO-BNST
8	VM-LC
9	PV-amyg-Ce
10	SO/SC-amyg-LaD
11	PH-dPAG

12	DM-LC
13	PV-BNST
14	SO/SC-amyg-CoN
15	SO/SC-amyg-AB
16	SO/SC-SN
17	SO/SC-amyg-LaV
18	DM-amyg-B
19	SO/SC-dPAG
20	VM-RN-MR
21	MM-amyg-B
22	PV-amyg-B

Reviewer #3 (Remarks to the Author):

Review of Jensen et al: Nuclei-specific hypothalamus networks predict a dimensional marker of stress in humans.

The study focuses on the role of the hypothalamus in stress responses. The hypothalamus is a complex structure composed of multiple nuclei. The study used resting-state fMRI on a large sample of healthy young adults, the Human Connectome Project (n=498) to examine the functional connectivity of specific hypothalamic nuclei and its relationship with prolonged stress.

The key findings are as follows: (1) the investigators were able to parcellate the human hypothalamus into seven nuclei in vivo; (2) the functional connectivity between these nuclei and other subcortical structures, including the amygdala, was used to predict stress scores out-of-sample; (3) the authors suggest that stress is related to connectivity changes in precise and functionally meaningful subcortical networks.

The strengths of the manuscripts are:

(1) The use of the HCP large sample size which speaks to the statistical power and the generalizability of the results.

(2) The investigators used high-quality fMRI resting-state data and parcellated the human hypothalamus into seven anatomically plausible nuclei, which is a level of detail allows for a more nuanced understanding of the brain's structure and function.

(3) The authors replicated the average hypothalamus connectivity and its parcellation in two independent datasets, which strengthens the validity of the findings.

(4) The study found that stress predictions were better when considering nuclei as opposed to whole hypothalamus connectivity, and they were behaviorally specific.

Thank you very much for this comprehensive summary and positive assessment of our work.

There are several weaknesses that should be considered:

(1) This manuscript is based on cross-sectional HCP data, which limits the ability to draw conclusions about causality or the direction of relationships between variables.

We fully agree with the reviewer. We have now added further detail to the discussion highlighting this limitation of our study.

Discussion (p.18; please note that page numbers throughout relate to the 'tracked changes' version of the manuscript):

We also note that while our work shows the importance of focusing on functionally relevant circuits, future work might additionally consider functionally relevant states, instead of the resting state, **potentially including internal and external stress challenges or longitudinal measures of stress**, which might further aid predictions (Misaki et al., 2023). **In general, all predictions here are based on cross-sectional data and longitudinal work will be vital in helping establish causality and in clarifying the direction of relationships between brain and stress markers.**

(2) The authors acknowledged that at a higher spatial resolution, it might be possible to identify lateral and medial subdivisions of the hypothalamus, which was not possible in this study.

We agree with the reviewer that this would have been very nice indeed. It is a limitation we are aware of and which we have already mentioned in the manuscript. We have now added some further detail to highlight the lack of the lateral nucleus in the Discussion.

Unfortunately, we are not aware of any other dataset with a similar quality and quantity of resting-state data that would have a higher spatial resolution and allow us to explore this further. We did examine the 7T HCP data to see if it could be exploited for a finer-grained parcellation, but it is a much smaller cohort, and we did not find that the signals were as reliable in the hypothalamus as in the 3T-HCP dataset. This might be partly because physiological monitoring was unfortunately not performed in this cohort and thus, we cannot correct the resting-state data for physiological noise.

Discussion (p.17):

We note several limitations of our work. Our parcellation achieved good subdivisions of the hypothalamus in the anterior-to-posterior and dorsal-to-ventral axis. However, at a higher spatial resolution, it might be possible to identify lateral and medial subdivisions of the hypothalamus (Haynes & Haber, 2013) which was not possible here. **The lateral nucleus is particularly important in the context of feeding behaviour and for acquiring and expressing cue-reward associations. It thus has an important role in learning (Sharpe et al., 2023). It has been identified in previous human work at higher resolution, including post-mortem studies (Baroncini et al., 2012; Mai et al., 2016; Neudorfer et al., 2020). Its importance in stress and its functional connectivity fingerprint should be considered in future work when data acquired at higher spatial resolutions and with better signal-to-noise becomes available.** In our main parcellation (**Fig 1**), the lateral hypothalamus was likely part of the paraventricular nucleus. A lateral nucleus appeared at a resolution of 1.6mm at 7T (**Supplementary Fig S3A**). However, the 7T data suffered from other problems such as a lack of variance in stress scores and missing physiological recordings required for data clean-up.

(3) The finding is based on resting state fMRI, which provides some idea about how brain structures are connected at rest but not necessarily how they interact to respond to an

external or internal challenge (stress). Therefore, the findings need to be considered with caution when interpreting the connectivity predictions.

Thanks, we agree that this is an important point. For the HCP data, fMRI measurements were only performed at rest and during specific cognitive and motor tasks, but not during a stress manipulation. This would have been very interesting, but we are currently limited to investigating the network at rest. We therefore do not know if the reactivity to stressors (whether internal or external) would have recruited an overlapping circuit to the one we identify as predictive of stress. Our marker of stress is much more of a state marker that captures prolonged periods of stress, but it would be very interesting for future work to examine how much overlap there is between the networks that mediate immediate stress reactivity versus those that are affected by long-term exposure to stress. We have now included a sentence acknowledging this interesting question in the discussion as follows:

Discussion (p.17):

In addition, it would be interesting to examine whether the same networks studied at rest here would be affected during an external or internal stress challenge, as well as after prolonged exposure to stress.

(4) The HCP data are based on healthy volunteers, thus there is relatively little insights that can be gained into process abnormalities that may occur in the context of stress disorders (such as PTSD).

We fully agree. Unfortunately, we are limited by the available data, but we think that we can still learn something interesting about variation within the subclinical/healthy range. We hope that follow-up work in clinical cohorts will build on our work and look at the same brain networks in PTSD and other stress-related disorders. There is currently little work on this in humans (see e.g., Raise-Abdullahi et al., 2023). Therefore, it remains an open question whether the brain networks are categorically different in PTSD or vary further along the continuum examined here, so that our data could make some useful predictions for clinical disorders. This is an interesting question that future work should address.

We have added this consideration to the discussion as follows (p.16-17):

Given our work solely relied on healthy participant's data, future longitudinal work should examine whether changes in hypothalamus coupling may be a precursor to mental illness, and whether targeted interventions such as those using transcranial ultrasonic stimulation (TUS; Bongioanni et al., 2021; Verhagen et al., 2019) might be able to rebalance the hypothalamus networks identified here (Haber et al., 2023; Hamani et al., 2011; Huys et al., 2016).

Direct insight into process abnormalities in stress disorders such as PTSD cannot be gained from our work in healthy participants. An important question for future work is therefore whether brain changes in stress disorders such as PTSD are categorically different from healthy controls,

or just further along the continuum examined in a healthy population here. In addition, it would be interesting to examine whether the same networks studied at rest here would be affected during an external or internal stress challenge, as well as after prolonged exposure to stress.

(5) The author imply that these findings may have some implication for interventions but since (a) this is based on healthy volunteers and (b) there was no intervention involved, this is highly speculative. Future research is needed to explore whether inducing changes in precise hypothalamus networks may alter an individual's experience of stress.

We agree and this is exactly how our phrasing was meant – apologies if we overstated the clinical implications. We were meaning to point to the importance for future work in exploring this, given the limitations of our study outlined by the reviewer. We have further toned down our wording wherever we mention interventions:

Abstract (p.2):

Here, we show stress relates to connectivity changes in precise and functionally meaningful subcortical networks, which may be exploited in **future studies using interventions** in stress disorders.

Discussion (p.16 and p.18):

Given our analyses relied on healthy volunteers, our interpretation of the underlying circuit changes should be confirmed in studies using longitudinal designs or direct interventions **in clinical populations** to establish causality.

As expected, these alternative mental health dimensions could also be predicted from hypothalamus coupling. However, predictions were consistently strongest for stress (Fig 6). **Given our work solely relied on healthy participant's data**, future longitudinal work should examine whether changes in hypothalamus coupling may be a precursor to mental illness, and whether targeted interventions such as those using transcranial ultrasonic stimulation (TUS; Bongioanni et al., 2021; Verhagen et al., 2019) **might be** able to rebalance the hypothalamus networks identified here (Haber et al., 2023; Hamani et al., 2011; Huys et al., 2016).

Future interventional studies should explore whether inducing changes in precise hypothalamus networks may alter an individual's experience of stress. **Exploring this in both healthy and clinical populations could have implications** for treating a range of mental health disorders whose onset is often preceded by a prolonged experience of stress.

Taken together, this is an interesting manuscript that provides some conceptual advance in understanding the role of the hypothalamus, but has limitations in terms of the causal conclusions and the applicability to individuals with stress disorders.

Thank you for your assessment of our work. We hope that you will agree that we are now more transparent about the limitations in the updated version of the manuscript and that it still provides an interesting conceptual advance to the field.

References

- Aarsland, D., Pålhagen, S., Ballard, C. G., Ehrt, U., & Svenningsson, P. (2011). Depression in Parkinson disease—Epidemiology, mechanisms and management. *Nature Reviews. Neurology*, *8*(1), 35–47. <https://doi.org/10.1038/nrneuro.2011.189>
- Agid, O., Kohn, Y., & Lerer, B. (2000). Environmental stress and psychiatric illness. *Biomedicine & Pharmacotherapy*, *54*(3), 135–141. [https://doi.org/10.1016/S0753-3322\(00\)89046-0](https://doi.org/10.1016/S0753-3322(00)89046-0)
- Arnsten, A. F. T. (2009). Stress signalling pathways that impair prefrontal cortex structure and function. *Nature Reviews. Neuroscience*, *10*(6), 410–422. <https://doi.org/10.1038/nrn2648>
- Averbeck, B. B., & Costa, V. D. (2017). Motivational neural circuits underlying reinforcement learning. *Nature Neuroscience*, *20*(4), 505–512. <https://doi.org/10.1038/nn.4506>
- Bao, A.-M., Meynen, G., & Swaab, D. F. (2008). The stress system in depression and neurodegeneration: Focus on the human hypothalamus. *Brain Research Reviews*, *57*(2), 531–553. <https://doi.org/10.1016/j.brainresrev.2007.04.005>
- Baroncini, M., Jissendi, P., Balland, E., Besson, P., Pruvo, J.-P., Francke, J.-P., Dewailly, D., Blond, S., & Prevot, V. (2012). MRI atlas of the human hypothalamus. *NeuroImage*, *59*(1), 168–180. <https://doi.org/10.1016/j.neuroimage.2011.07.013>
- Betts, M. J., Cardenas-Blanco, A., Kanowski, M., Jessen, F., & Düzel, E. (2017). In vivo MRI assessment of the human locus coeruleus along its rostrocaudal extent in young and older adults. *NeuroImage*, *163*, 150–159. <https://doi.org/10.1016/j.neuroimage.2017.09.042>
- Billot, B., Bocchetta, M., Todd, E., Dalca, A. V., Rohrer, J. D., & Iglesias, J. E. (2020). Automated segmentation of the hypothalamus and associated subunits in brain MRI. *NeuroImage*, *223*, 117287. <https://doi.org/10.1016/j.neuroimage.2020.117287>
- Bocchetta, M., Gordon, E., Manning, E., Barnes, J., Cash, D. M., Espak, M., Thomas, D. L., Modat, M., Rossor, M. N., Warren, J. D., Ourselin, S., Frisoni, G. B., & Rohrer, J. D. (2015). Detailed volumetric analysis of the hypothalamus in behavioral variant frontotemporal dementia. *Journal of Neurology*, *262*(12), 2635. <https://doi.org/10.1007/s00415-015-7885-2>
- Bongioanni, A., Folloni, D., Verhagen, L., Sallet, J., Klein-Flügge, M. C., & Rushworth, M. F. S. (2021). Activation and disruption of a neural mechanism for novel choice in monkeys. *Nature*, 1–5. <https://doi.org/10.1038/s41586-020-03115-5>
- Brown, R. E., Basheer, R., McKenna, J. T., Strecker, R. E., & McCarley, R. W. (2012). Control of Sleep and Wakefulness. *Physiological Reviews*, *92*(3), 1087–1187. <https://doi.org/10.1152/physrev.00032.2011>
- de Kloet, E. R., Joëls, M., & Holsboer, F. (2005). Stress and the brain: From adaptation to disease. *Nature Reviews. Neuroscience*, *6*(6), 463–475. <https://doi.org/10.1038/nrn1683>

- Edlow, B. L., Takahashi, E., Wu, O., Benner, T., Dai, G., Bu, L., Grant, P. E., Greer, D. M., Greenberg, S. M., Kinney, H. C., & Folkerth, R. D. (2012). Neuroanatomic connectivity of the human ascending arousal system critical to consciousness and its disorders. *Journal of Neuropathology and Experimental Neurology*, *71*(6), 531–546.
<https://doi.org/10.1097/NEN.0b013e3182588293>
- Faull, O. K., Jenkinson, M., Ezra, M., & Pattinson, K. T. (2016). Conditioned respiratory threat in the subdivisions of the human periaqueductal gray. *eLife*, *5*.
<https://doi.org/10.7554/eLife.12047>
- Folloni, D., Sallet, J., Khrapitchev, A. A., Sibson, N., Verhagen, L., & Mars, R. B. (2019). Dichotomous organization of amygdala/temporal-prefrontal bundles in both humans and monkeys. *eLife*, *8*, e47175. <https://doi.org/10.7554/eLife.47175>
- Glasser, M. F., Coalson, T. S., Robinson, E. C., Hacker, C. D., Harwell, J., Yacoub, E., Ugurbil, K., Andersson, J., Beckmann, C. F., Jenkinson, M., Smith, S. M., & Van Essen, D. C. (2016). A multi-modal parcellation of human cerebral cortex. *Nature*, *536*(7615), 171–178.
<https://doi.org/10.1038/nature18933>
- Haber, S. N., Lehman, J., Maffei, C., & Yendiki, A. (2023). The Rostral Zona Incerta: A Subcortical Integrative Hub and Potential Deep Brain Stimulation Target for Obsessive-Compulsive Disorder. *Biological Psychiatry*, *0*(0). <https://doi.org/10.1016/j.biopsych.2023.01.006>
- Hamani, C., Mayberg, H., Stone, S., Laxton, A., Haber, S., & Lozano, A. M. (2011). The subcallosal cingulate gyrus in the context of major depression. *Biological Psychiatry*, *69*(4), 301–308.
<https://doi.org/10.1016/j.biopsych.2010.09.034>
- Haynes, W. I. A., & Haber, S. N. (2013). The Organization of Prefrontal-Subthalamic Inputs in Primates Provides an Anatomical Substrate for Both Functional Specificity and Integration: Implications for Basal Ganglia Models and Deep Brain Stimulation. *The Journal of Neuroscience*, *33*(11), 4804–4814. <https://doi.org/10.1523/JNEUROSCI.4674-12.2013>
- Herman, J. P., Flak, J., & Jankord, R. (2008). Chronic stress plasticity in the hypothalamic paraventricular nucleus. *Progress in Brain Research*, *170*, 353–364.
[https://doi.org/10.1016/S0079-6123\(08\)00429-9](https://doi.org/10.1016/S0079-6123(08)00429-9)
- Hermans, E. J., Henckens, M. J. A. G., Joëls, M., & Fernández, G. (2014). Dynamic adaptation of large-scale brain networks in response to acute stressors. *Trends in Neurosciences*, *37*(6), 304–314. <https://doi.org/10.1016/j.tins.2014.03.006>
- Hermans, E. J., van Marle, H. J. F., Ossewaarde, L., Henckens, M. J. A. G., Qin, S., van Kesteren, M. T. R., Schoots, V. C., Cousijn, H., Rijpkema, M., Oostenveld, R., & Fernández, G. (2011). Stress-related noradrenergic activity prompts large-scale neural network reconfiguration. *Science (New York, N.Y.)*, *334*(6059), 1151–1153.
<https://doi.org/10.1126/science.1209603>

- Huys, Q. J. M., Maia, T. V., & Paulus, M. P. (2016). Computational Psychiatry: From Mechanistic Insights to the Development of New Treatments. *Biological Psychiatry. Cognitive Neuroscience and Neuroimaging*, *1*(5), 382–385.
<https://doi.org/10.1016/j.bpsc.2016.08.001>
- Keuken, M. C., Bazin, P.-L., Crown, L., Hootsmans, J., Laufer, A., Müller-Axt, C., Sier, R., van der Putten, E. J., Schäfer, A., Turner, R., & Forstmann, B. U. (2014). Quantifying inter-individual anatomical variability in the subcortex using 7 T structural MRI. *NeuroImage*, *94*, 40–46. <https://doi.org/10.1016/j.neuroimage.2014.03.032>
- Klein-Flügge, M. C., Jensen, D. E. A., Takagi, Y., Priestley, L., Verhagen, L., Smith, S. M., & Rushworth, M. F. S. (2022). Relationship between nuclei-specific amygdala connectivity and mental health dimensions in humans. *Nature Human Behaviour*.
<https://doi.org/10.1038/s41562-022-01434-3>
- Kullmann, S., Heni, M., Linder, K., Zipfel, S., Häring, H.-U., Veit, R., Fritsche, A., & Preissl, H. (2014). Resting-state functional connectivity of the human hypothalamus. *Human Brain Mapping*, *35*(12), 6088–6096. <https://doi.org/10.1002/hbm.22607>
- Lebow, M. A., & Chen, A. (2016). Overshadowed by the amygdala: The bed nucleus of the stria terminalis emerges as key to psychiatric disorders. *Molecular Psychiatry*, *21*(4), 450–463.
<https://doi.org/10.1038/mp.2016.1>
- Lechan, R. M., & Toni, R. (2000). Functional Anatomy of the Hypothalamus and Pituitary. In *Endotext*. MDText.com, Inc. <http://www.ncbi.nlm.nih.gov/pubmed/25905349>
- Lemaire, J.-J., Frew, A. J., McArthur, D., Gorgulho, A. A., Alger, J. R., Salomon, N., Chen, C., Behnke, E. J., & De Salles, A. A. F. (2011). White matter connectivity of human hypothalamus. *Brain Research*, *1371*, 43–64.
<https://doi.org/10.1016/j.brainres.2010.11.072>
- Levone, B. R., Cryan, J. F., & O’Leary, O. F. (2014). Role of adult hippocampal neurogenesis in stress resilience. *Neurobiology of Stress*, *1*, 147–155.
<https://doi.org/10.1016/j.ynstr.2014.11.003>
- Mai, J. K., Majtanik, M., & Paxinos, G. (2015). *Atlas of the Human Brain*. Academic Press.
- Mai, J., Majtanik, M., & Paxinos, G. (2016). *Atlas of the Human Brain: Vol. 4th edition*. Academic press. <https://www.elsevier.com/books/atlas-of-the-human-brain/mai/978-0-12-802800-1>
- Makris, N., Swaab, D. F., van der Kouwe, A., Abbs, B., Boriel, D., Handa, R. J., Tobet, S., & Goldstein, J. M. (2013a). Volumetric parcellation methodology of the human hypothalamus in neuroimaging: Normative data and sex differences. *NeuroImage*, *69*, 1–10. <https://doi.org/10.1016/j.neuroimage.2012.12.008>
- Makris, N., Swaab, D. F., van der Kouwe, A., Abbs, B., Boriel, D., Handa, R. J., Tobet, S., & Goldstein, J. M. (2013b). Volumetric parcellation methodology of the human

- hypothalamus in neuroimaging: Normative data and sex differences. *NeuroImage*, *69*, 1–10. <https://doi.org/10.1016/j.neuroimage.2012.12.008>
- McEwen, B. S., Bowles, N. P., Gray, J. D., Hill, M. N., Hunter, R. G., Karatsoreos, I. N., & Nasca, C. (2015). Mechanisms of stress in the brain. *Nature Neuroscience*, *18*(10), Article 10. <https://doi.org/10.1038/nn.4086>
- McEwen, B. S., & Gianaros, P. J. (2010). Central role of the brain in stress and adaptation: Links to socioeconomic status, health, and disease. *Annals of the New York Academy of Sciences*, *1186*, 190–222. <https://doi.org/10.1111/j.1749-6632.2009.05331.x>
- McEwen, B. S., Nasca, C., & Gray, J. D. (2016). Stress Effects on Neuronal Structure: Hippocampus, Amygdala, and Prefrontal Cortex. *Neuropsychopharmacology*, *41*(1), Article 1. <https://doi.org/10.1038/npp.2015.171>
- Misaki, M., Tsuchiyagaito, A., Guinjoan, S. M., Rohan, M. L., & Paulus, M. P. (2023). *Trait repetitive negative thinking in depression is associated with functional connectivity in negative thinking state, not resting state* (p. 2023.03.23.533932). bioRxiv. <https://doi.org/10.1101/2023.03.23.533932>
- Myers, B., McKlveen, J. M., & Herman, J. P. (2012). Neural Regulation of the Stress Response: The Many Faces of Feedback. *Cellular and Molecular Neurobiology*. <https://doi.org/10.1007/s10571-012-9801-y>
- Neudorfer, C., Germann, J., Elias, G. J. B., Gramer, R., Boutet, A., & Lozano, A. M. (2020a). A high-resolution in vivo magnetic resonance imaging atlas of the human hypothalamic region. *Scientific Data*, *7*(1), Article 1. <https://doi.org/10.1038/s41597-020-00644-6>
- Neudorfer, C., Germann, J., Elias, G. J. B., Gramer, R., Boutet, A., & Lozano, A. M. (2020b). A high-resolution in vivo magnetic resonance imaging atlas of the human hypothalamic region. *Scientific Data*, *7*(1), 305. <https://doi.org/10.1038/s41597-020-00644-6>
- Nieuwenhuys, R., Voogd, J., & van Huijzen, C. (Eds.). (2008). *The Human Central Nervous System, Diencephalon: Hypothalamus*. Springer. https://doi.org/10.1007/978-3-540-34686-9_10
- Ogawa, A., Osada, T., Tanaka, M., Kamagata, K., Aoki, S., & Konishi, S. (2020). Connectivity-based localization of human hypothalamic nuclei in functional images of standard voxel size. *NeuroImage*, *221*, 117205. <https://doi.org/10.1016/j.neuroimage.2020.117205>
- Oishi, Y., Xu, Q., Wang, L., Zhang, B.-J., Takahashi, K., Takata, Y., Luo, Y.-J., Cherasse, Y., Schiffmann, S. N., d'Exaerde, A. de K., Urade, Y., Qu, W.-M., Huang, Z.-L., & Lazarus, M. (2017). Slow-wave sleep is controlled by a subset of nucleus accumbens core neurons in mice. *Nature Communications*, *8*(1), 1–12. <https://doi.org/10.1038/s41467-017-00781-4>
- Osada, T., Suzuki, R., Ogawa, A., Tanaka, M., Hori, M., Aoki, S., Tamura, Y., Watada, H., Kawamori, R., & Konishi, S. (2017). Functional subdivisions of the hypothalamus using areal parcellation and their signal changes related to glucose metabolism. *NeuroImage*, *162*, 1–12. <https://doi.org/10.1016/j.neuroimage.2017.08.056>

- Raise-Abdullahi, P., Meamar, M., Vafaei, A. A., Alizadeh, M., Dadkhah, M., Shafia, S., Ghalandari-Shamami, M., Naderian, R., Afshin Samaei, S., & Rashidy-Pour, A. (2023). Hypothalamus and Post-Traumatic Stress Disorder: A Review. *Brain Sciences*, *13*(7), 1010. <https://doi.org/10.3390/brainsci13071010>
- Roosendaal, B., McEwen, B. S., & Chattarji, S. (2009). Stress, memory and the amygdala. *Nature Reviews. Neuroscience*, *10*(6), 423–433. <https://doi.org/10.1038/nrn2651>
- Schmidt, M. V., Sterlemann, V., & Müller, M. B. (2008). Chronic stress and individual vulnerability. *Annals of the New York Academy of Sciences*, *1148*, 174–183. <https://doi.org/10.1196/annals.1410.017>
- Schönknecht, P., Anwander, A., Petzold, F., Schindler, S., Knösche, T. R., Möller, H. E., Hegerl, U., Turner, R., & Geyer, S. (2013). Diffusion imaging-based subdivision of the human hypothalamus: A magnetic resonance study with clinical implications. *European Archives of Psychiatry and Clinical Neuroscience*, *263*(6), 497–508. <https://doi.org/10.1007/s00406-012-0389-5>
- Schonknecht, P., Anwander, A., Petzold, F., Schindler, S., Knosche, T. R., Moller, H. E., Hegerl, U., Turner, R., & Geyer, S. (2013). Diffusion imaging-based subdivision of the human hypothalamus: A magnetic resonance study with clinical implications. *European Archives of Psychiatry and Clinical Neuroscience*, *263*(6), 497–508. <https://doi.org/10.1007/s00406-012-0389-5>
- Schwabe, L., Hermans, E. J., Joëls, M., & Roosendaal, B. (2022). Mechanisms of memory under stress. *Neuron*, *110*(9), 1450–1467. <https://doi.org/10.1016/j.neuron.2022.02.020>
- Smith, S. M., Hyvärinen, A., Varoquaux, G., Miller, K. L., & Beckmann, C. F. (2014). Group-PCA for very large fMRI datasets. *Neuroimage*, *101*, 738–749. <https://doi.org/10.1016/j.neuroimage.2014.07.051>
- Smith, S. M., Nichols, T. E., Vidaurre, D., Winkler, A. M., Behrens, T. E. J., Glasser, M. F., Ugurbil, K., Barch, D. M., Van Essen, D. C., & Miller, K. L. (2015). A positive-negative mode of population covariation links brain connectivity, demographics and behavior. *Nature Neuroscience*, *18*(11), 1565–1567. <https://doi.org/10.1038/nn.4125>
- Tian, Y., Margulies, D. S., Breakspear, M., & Zalesky, A. (2020). Topographic organization of the human subcortex unveiled with functional connectivity gradients. *Nature Neuroscience*, *23*(11), Article 11. <https://doi.org/10.1038/s41593-020-00711-6>
- Ulrich-Lai, Y. M., & Herman, J. P. (2009). Neural regulation of endocrine and autonomic stress responses. *Nature Reviews. Neuroscience*, *10*(6), 397–409. <https://doi.org/10.1038/nrn2647>
- Verhagen, L., Gallea, C., Folloni, D., Constans, C., Jensen, D. E., Ahnine, H., Roumazeilles, L., Santin, M., Ahmed, B., Lehericy, S., Klein-Flügge, M. C., Krug, K., Mars, R. B., Rushworth, M. F., Pouget, P., Aubry, J.-F., & Sallet, J. (2019). Offline impact of transcranial focused

ultrasound on cortical activation in primates. *eLife*, 8.

<https://doi.org/10.7554/eLife.40541>

Zhang, G.-W., Shen, L., Tao, C., Jung, A.-H., Peng, B., Li, Z., Zhang, L. I., & Tao, H. W. (2021).

Medial preoptic area antagonistically mediates stress-induced anxiety and parental behavior. *Nature Neuroscience*, 24(4), Article 4. <https://doi.org/10.1038/s41593-020-00784-3>

Reviewer #1 (Remarks to the Author):

Review

Generally, I want to thank and congratulate the authors for another improvement of their compelling study that is characterized by well reflected steps of analyses. The only concern that, however, can be quickly resolved to my opinion is on Q1/A1. All other answers leave no further questions from my side but clarify the respective concerns - still, see some re-comments in case the initial questions were unclear which of course was not intended.

A1. I think the request for the hippocampal subregions has been formidably handled on one hand, and this effort is truly appreciated. On the other hand I do not fully agree with the result presentation and discussion of this:

A1.1. I think the HIP results should not be presented completely negative as is now the case: First, three connections were strong enough to push aside others. So, there are contributions, but only from a minority of connections (3 versus 8 of the amygdala). The statement "Added only noise" seems contradicting this observation, although this seems to relate to the prediction power of all connections. I would prefer a presentation in that there is a dichotomic situation: some HIP connections are carriers of information, and some are not at all ('noise').

A1.2. Discussion: The statement "The hippocampus shows sign of degeneration following stress (Levone et al., 2014; McEwen et al., 2016) and is therefore an important subcortical region not included a priori here." – is not quite correct in the year 2023. The hippocampus, structurally, is deficient in many psychiatric diseases, in subregionally different ways, but none of the longitudinal studies could convincingly prove that prolonged stress leads to 'degeneration'. The neurotoxic direction of the discussion might not be so helpful here, and more data speak to an early programming of HIP structure by early life adversity, with lasting effects.

A1.3. More importantly, and seeing the many amygdala connections contributing, the authors might to more carefully consider that the predicted (subclinical) phenotype was built from the subjective domain, and dominated here by affect & fear rather than true chronic stress ('burnout') items. Also, the predicted score was not an intermediate, validated stress marker such as the cortisol response (that has been demonstrated to be predicted by HIP structure and function in many publications). Thus, one key argument that the HIP is not contributing much here could be the non-HPA-axis type of the phenotype, whereas the amygdala is prone to coding fearful emotions.

A2: Perfect. No additions from my side.

A3: Same.

A4: I see the line of argumentation and the methodological decisions regarding the ROIs of the regional selected is not under critique. Yet, please understand that given for example the subiculum of the hippocampus as directly involved in the HPA axis regulation, it is difficult to follow why the hippocampus (with automated subregional segmentations) were not included from scratch. – In this line, disturbances of the HPA axis are affecting monoaminergic neurotransmission downstream, particularly when chronic stress play in. The accumbens, surely extremely important for the reward system, is also introduced with rather indirect argumentation (Parkinson's etc.). Among these competing regions, the hippocampus is not inferior in terms of the knowledge base around stress - take only the GR mediated hippocampal HPA axis feedback loop. But I'm satisfied with the hippocampal results being shown overall.

A5 is very convincing and insightful, the effort is appreciated. No additional questions.

Q6-Q10: clear.

Q11: clear and very interesting, indeed.

Q12: clear.

Q13: clear.

Q14: clear and indeed complex. Thanks for explaining this in full detail. It is helpful that these FC details are now in the supplemental data.

Q15: clear.

Q16: clear. I think the fact that BMI does not alter the results is another hint that the main clinical phenotype goal is not so much HPA-related (where BMI is).

Q17-Q18: clear.

Q19: The sentence "Importantly, from the work presented here, we cannot conclude that predictions of stress can only be achieved when based on hypothalamus functional connectivity." is too weak. No one would assume that only HTH FC relates to stress. See also work by Kühnel et al. that used whole brain FC to predict true acute stress patterns.

Q20: Clear now. It was or had not been clear that all connections contain HTH subregions.

Q21-Q28: clear.

Q29: See Q1.

Q30: See papers by Kühnel et al. on task-fMRI predicting acute subjective stress levels in a psychosocial stress test. See Kiem et al. for HIP functional connectivity predicting dex-CRH-test based cortisol values. See Veer et al. who also published resting state fMRI / cortisol correlations.

Q31-32: Done.

Reviewer #2 (Remarks to the Author):

The manuscript seems much improved by revision and I think it is now suitable for publication.

Reviewer #3 (Remarks to the Author):

The authors have completed a careful and thoughtful response to this and the other reviewers' concerns. Given that my primary issue was the limited ability to draw conclusions about clinical populations and given that the authors fully acknowledge this point, I have no other concerns.

REVIEWERS' COMMENTS

Reviewer #1 (Remarks to the Author):

Review

Generally, I want to thank and congratulate the authors for another improvement of their compelling study that is characterized by well reflected steps of analyses. The only concern that, however, can be quickly resolved to my opinion is on Q1/A1. All other answers leave no further questions from my side but clarify the respective concerns - still, see some re-comments in case the initial questions were unclear which of course was not intended.

Thanks for your thorough comments both in the previous and this round of revisions. They have really helped us, and we respond to the remaining points, in particular Q1/A1 below.

A1. I think the request for the hippocampal subregions has been formidably handled on one hand, and this effort is truly appreciated. On the other hand I do not fully agree with the result presentation and discussion of this:

A1.1. I think the HIP results should not be presented completely negative as is now the case: First, three connections were strong enough to push aside others. So, there are contributions, but only from a minority of connections (3 versus 8 of the amygdala). The statement “Added only noise” seems contradicting this observation, although this seems to relate to the prediction power of all connections. I would prefer a presentation in that there is a dichotomic situation: some HIP connections are carriers of information, and some are not at all (‘noise’).

We agree we could be more nuanced and have now changed the description of the hippocampus results in the Supplementary Information as suggested:

“Similarly, when edges were included iteratively, adding the hippocampus to the set of ROIs did **not further improve the predictions**: the peak was not superior (updated peak prediction at 18 connections including hippocampus-to-hypothalamus: $r=0.222$ versus original peak prediction at 22 connections without hippocampus included: $r=0.272$). **Nevertheless**, among the strongest 20 predictors identified in the training set, we identified three edges between hypothalamus and hippocampus (VM to HClhead (5th), PV to HCm1head (11th), and VMa to HCtail (19th)). **This suggests some hippocampus connections do indeed carry information relevant to stress**. Supplementary Figure 8 shows the results of these additional analyses. “

A1.2. Discussion: The statement “The hippocampus shows sign of degeneration following stress (Levone et al., 2014; McEwen et al., 2016) and is therefore an important subcortical region not included a priori here.” – is not quite correct in the year 2023. The hippocampus, structurally, is deficient in many psychiatric diseases, in subregionally different ways, but none of the longitudinal studies could convincingly prove that prolonged stress leads to ‘degeneration’. The neurotoxic direction of the discussion might not be so helpful here, and more data speak to an early programming of HIP structure by early life adversity, with lasting effects.

Thank you - we are not experts in the hippocampal literature and appreciate this feedback. We have changed this sentence to reflect this updated view:

“The hippocampus is affected by stress, in particular early life adversity, which can have lasting effects on its structure and function (McLaughlin et al., 2019; Wang et al., 2022). It is therefore an important subcortical region not included a priori here.”

A1.3. More importantly, and seeing the many amygdala connections contributing, the authors might to more carefully consider that the predicted (subclinical) phenotype was built from the subjective domain, and dominated here by affect & fear rather than true chronic stress (‘burnout’) items. Also, the predicted score was not an intermediate, validated stress marker such as the cortisol response (that has been demonstrated to be predicted by HIP structure and function in many publications). Thus, one key argument that the HIP is not contributing much here could be the non-HPA-axis type of the phenotype, whereas the amygdala is prone to coding fearful emotions.

Thank you, we have added a few sentences to the discussion as suggested by the reviewer:

Discussion:

While these new analyses do not question the importance of the hippocampus in stress predictions *per se*, they do suggest functional connectivity between specific hippocampus-subdivisions to hypothalamus-nuclei does not carry additional predictive variance for stress. We note, however, that the marker of stress used in this study was derived from subjective questionnaire items such as perceived stress and fear. As part of the HCP dataset, we did not have access to a true chronic stress item (e.g., burnout) or a validated stress marker such as the cortisol response. Coupling between the hippocampus and regions other than the hypothalamus, or as a function of objective stress markers, should be considered in future work.

Q19: The sentence “Importantly, from the work presented here, we cannot conclude that predictions of stress can only be achieved when based on hypothalamus functional connectivity.” is too weak. No one would assume that only HTH FC relates to stress. See also work by Kühnel et al. that used whole brain FC to predict true acute stress patterns.

We have reworded this sentence to say:

“Importantly, from the work presented here, we can only conclude that predictions of stress can be achieved when based on hypothalamus functional connectivity. We cannot draw conclusions about the contributions of other brain hubs not investigated here. Previous work has shown more widespread whole-brain as well as specific amygdala/hippocampus contributions in the context of acute stress (Kiem et al., 2013; Kühnel et al., 2022; Veer et al., 2011, 2012).”

Q30: See papers by Kühnel et al. on task-fMRI predicting acute subjective stress levels in a psychosocial stress test. See Kiem et al. for HIP functional connectivity predicting dex-CRH-test based cortisol values. See Veer et al. who also published resting state fMRI / cortisol correlations.

Thanks, these have now been included. Thanks again so much for your helpful and careful reading of our manuscript which we believe has really helped to improved it.

References:

- Kiem, S. A., Andrade, K. C., Spoormaker, V. I., Holsboer, F., Czisch, M., & Sämann, P. G. (2013). Resting state functional MRI connectivity predicts hypothalamus-pituitary-axis status in healthy males. *Psychoneuroendocrinology*, *38*(8), 1338–1348. <https://doi.org/10.1016/j.psyneuen.2012.11.021>
- Kühnel, A., Czisch, M., Sämann, P. G., BeCOME Working Group, Binder, E. B., & Kroemer, N. B. (2022). Spatiotemporal Dynamics of Stress-Induced Network Reconfigurations Reflect Negative Affectivity. *Biological Psychiatry*, *92*(2), 158–169. <https://doi.org/10.1016/j.biopsych.2022.01.008>
- McLaughlin, K. A., Weissman, D., & Bitrán, D. (2019). Childhood Adversity and Neural Development: A Systematic Review. *Annual Review of Developmental Psychology*, *1*, 277–312. <https://doi.org/10.1146/annurev-devpsych-121318-084950>
- Veer, I. M., Oei, N. Y. L., Spinhoven, P., van Buchem, M. A., Elzinga, B. M., & Rombouts, S. A. R. B. (2011). Beyond acute social stress: Increased functional connectivity between amygdala and cortical midline structures. *NeuroImage*, *57*(4), 1534–1541. <https://doi.org/10.1016/j.neuroimage.2011.05.074>
- Veer, I. M., Oei, N. Y. L., Spinhoven, P., van Buchem, M. A., Elzinga, B. M., & Rombouts, S. A. R. B. (2012). Endogenous cortisol is associated with functional connectivity between the amygdala and medial prefrontal cortex. *Psychoneuroendocrinology*, *37*(7), 1039–1047. <https://doi.org/10.1016/j.psyneuen.2011.12.001>
- Wang, H., van Leeuwen, J. M. C., de Voogd, L. D., Verkes, R.-J., Roozendaal, B., Fernández, G., & Hermans, E. J. (2022). Mild early-life stress exaggerates the impact of acute stress on corticolimbic resting-state functional connectivity. *The European Journal of Neuroscience*, *55*(9–10), 2122–2141. <https://doi.org/10.1111/ejn.15538>